# A Notch-dependent transcriptional mechanism controls expression of temporal patterning factors in *Drosophila* medulla

**Alokananda Ray, Xin Li***

Department of Cell and Developmental Biology, University of Illinois at Urbana-Champaign, Urbana, United States

**Abstract** Temporal patterning is an important mechanism for generating a great diversity of neuron subtypes from a seemingly homogenous progenitor pool in both vertebrates and invertebrates. *Drosophila* neuroblasts are temporally patterned by sequentially expressed Temporal Transcription Factors (TTFs). These TTFs are proposed to form a transcriptional cascade based on mutant phenotypes, although direct transcriptional regulation between TTFs has not been verified in most cases. Furthermore, it is not known how the temporal transitions are coupled with the generation of the appropriate number of neurons at each stage. We use neuroblasts of the *Drosophila* optic lobe medulla to address these questions and show that the expression of TTFs Sloppy-paired 1/2 (Slp1/2) is directly regulated at the transcriptional level by two other TTFs and the cell-cycle dependent Notch signaling through two *cis*-regulatory elements. We also show that supplying constitutively active Notch can rescue the delayed transition into the Slp stage in cell cycle arrested neuroblasts. Our findings reveal a novel Notch-pathway dependent mechanism through which the cell cycle progression regulates the timing of a temporal transition within a TTF transcriptional cascade.

**\*For correspondence:**
lixin@illinois.edu

**Competing interest:** The authors declare that no competing interests exist.

## Editor's evaluation

Neural stem cells express cascades of transcription factors that are important for generating the diversity of neurons in the brain of flies and mammals. Yet nothing is known about whether the transcription factor cascades are built from direct gene regulation, e.g. factor A binding to enhancers in gene B to activate its expression. Here, Ray and Li show that one temporal factor, Slp1/2, is regulated transcriptionally via two molecularly defined enhancers that directly bind two other transcription factors in the cascade as well as integrating Notch signaling. This is a major step forward for the field and provides a model for subsequent studies on other temporal transcription factor cascades.

## Introduction

The generation of a great diversity of neurons from a small pool of neural progenitors is critical for constructing functional nervous systems. This is partially achieved by integration of temporal patterning and spatial patterning of neural progenitors (reviewed in *Lin and Lee, 2012*; *Allan and Thor, 2015*; *Holguera and Desplan, 2018*; *Sagner and Briscoe, 2019*). Temporal patterning of neural progenitors refers to the generation of differently fated progeny in a birth-order-dependent manner, and this is observed in both invertebrates and vertebrates (reviewed in *Pearson and Doe, 2004*; *Oberst et al., 2019*; *Maurange, 2020*). The central nervous system in *Drosophila melanogaster* has been an excellent model to study temporal patterning. Neural progenitors called neuroblasts (NBs) have been

shown to sequentially express series of temporal transcription factors (TTFs), which are each required to specify subsets of neuron fates (reviewed in *Doe, 2017*; *Miyares and Lee, 2019*). For example, neuroblasts in the embryonic ventral nerve cord (VNC) are temporally patterned by a TTF cascade Hunchback (Hb), Kruppel (Kr), Nubbin/Pdm2 (Pdm), Castor (Cas), and Grainy head (Grh) (*Brody and Odenwald, 2000*; *Isshiki et al., 2001*; *Grosskortenhaus et al., 2006*; *Baumgardt et al., 2009*), while *Drosophila* optic lobe medulla neuroblasts utilize a different TTF cascade composed of Homothorax (Hth), SoxNeuro (SoxN), Doublesex-Mab related 99B (Dmrt99B), Odd paired (Opa), Eyeless (Ey), Earmuff (Erm), Homeobrain (Hbn), Sloppy-paired (Slp1,Slp2), Scarecrow (Scro), Dichaete (D), BarH1/2, Tailless (Tll), and Glial cell missing (Gcm) ( *Suzuki et al., 2013*; *Zhu et al., 2022*; *Konstantinides et al., 2022*; *Li et al., 2013*; *Tang et al., 2022*). In these TTF cascades, cross-regulations were identified among TTF genes based on loss- and gain-of-function phenotypes, and they were proposed to form transcriptional cascades (*Isshiki et al., 2001*; *Grosskortenhaus et al., 2006*; *Baumgardt et al., 2009*; *Li et al., 2013*; *Maurange et al., 2008*) that can in theory self-propagate (*Averbukh et al., 2018*). However, with a few exceptions (*Hirono et al., 2012*), the *cis*-regulatory elements of these TTF genes haven't been characterized. Thus, it is not known whether these cross-regulations are indeed direct transcriptional regulations. Moreover, in the embryonic TTF cascade, although mis-expression of one TTF is sufficient to activate the next TTF and repress the 'next plus one' TTF, loss of one TTF often does not block temporal progression, but only causes the corresponding fates to be skipped (*Isshiki et al., 2001*). In the medulla TTF cascade, one TTF is required but not sufficient to promote the TTF cascade progression (*Li et al., 2013*). These studies suggest that additional mechanisms besides cross-regulations among TTFs might function to regulate TTF progression.

As the neural progenitors go through the TTF cascade, they need to generate a certain number of postmitotic progeny at each temporal stage. Therefore, it has been questioned whether cell cycle progression is required for the TTF cascade progression. In the embryonic VNC TTF cascade, cytokinesis is required for the first transition (Hb to Kr) and acts by promoting the nuclear export of seven-up (*svp*) mRNA encoding a switching factor, but all later temporal transitions progress normally in G2-arrested neuroblasts (*Isshiki et al., 2001*; *Grosskortenhaus et al., 2005*; *Mettler et al., 2006*). In larval VNC NBs, timely transition from the Imp /Castor/Chinmo to Syncrip/Broad stage also requires cell cycle progression (*van den Ameele and Brand, 2019*). These results suggest that the requirement of cell-cycle progression varies depending on the specific temporal transition. In the vertebrate cortex, the temporal transition from generating deep layer neurons to generating upper layer neurons is not affected by blocking cell cycle progression when constitutively active Notch signaling is provided to maintain the un-differentiated status of cortical neural progenitors (*Okamoto et al., 2016*). However, it is possible that Notch signaling, which is normally dependent on the asymmetric cell division of neural progenitors, may have rescued the possible phenotype caused by the arrest of cell cycle progression.

Notch signaling plays pleiotropic roles in neurogenesis and neural development in both vertebrates and invertebrates (reviewed in *Moore and Alexandre, 2020*). Upon binding to a ligand from a neighboring cell, the Notch receptor is cleaved to release the Notch intracellular domain (NICD), which then enters the nucleus and associates with the DNA-binding protein CSL (CBF1/RBPjκ/Su(H)/ Lag-1). The NICD-CSL then recruits the transcriptional coactivator Mastermind (MAM) to activate transcription of Notch target genes, mainly the bHLH transcriptional repressors of the HES (hairy and enhancer of split) /HEY families (reviewed in *Fortini, 2009*; *Kopan and Ilagan, 2009*; *Bray and Furriols, 2001*). *Drosophila* NBs divide asymmetrically multiple times and at each division generate a self-renewed neuroblast and a more differentiated progeny, that is either an intermediate progenitor [Ganglion Mother Cell (GMC) for type I NBs, or Intermediate Neural Progenitor (INP) for type II NBs], or a postmitotic progeny (type 0 NBs) (reviewed in *Walsh and Doe, 2017*). During the asymmetric division of *Drosophila* neuroblasts, Numb, a negative regulator of Notch signaling, is asymmetrically localized to the intermediate progenitor, resulting in unidirectional Delta-Notch signaling from the intermediate progenitor to the neuroblast (*Knoblich et al., 1995*; *Rhyu et al., 1994*; *Roegiers and Jan, 2004*; *Fuerstenberg et al., 1998*). Ectopic activation of Notch signaling in the progeny causes them to revert into neuroblasts and over-proliferate, while inhibition of Notch signaling in type II NBs eliminates the whole lineage (*Wang et al., 2006*; *Bowman et al., 2008*; *Weng et al., 2010*; *Zacharioudaki et al., 2012*). In contrast, in most type I NB lineages, loss of N signaling does not affect neuroblast maintenance, and it was shown that E(spl) complex proteins and Deadpan act redundantly to maintain the un-differentiated state (stemness) of neuroblasts by repressing differentiation genes

(*Zacharioudaki et al., 2012*; *Almeida and Bray, 2005*; *Magadi et al., 2020*). Notch signaling is also involved in controlling the daughter cell proliferation modes specifically the type I to type 0 switch in embryonic VNC NBs (*Ulvklo et al., 2012*; *Bivik et al., 2016*). Work in the vertebrate central nervous system suggests that during the asymmetric division of neural progenitors, the more differentiated progeny expresses Delta and activates Notch signaling in the sister cell to maintain the neural progenitor fate (reviewed in *Moore and Alexandre, 2020*). In addition, Notch signaling shows cell-cycle-dependent activation and the expression of the Notch target Hes genes oscillates during the cell cycle in neural progenitors (reviewed in *Kageyama et al., 2009*). Although Notch signaling has been shown to play many important roles in neural lineage development, whether Notch signaling also has a role in temporal transitions between TTFs has not been investigated.

Here, we use *Drosophila* medulla neuroblasts to address these questions. In the larval optic lobe, a neurogenesis wave spreads in a medial to lateral direction to sequentially convert neuroepithelial cells into medulla neuroblasts. As a result, medulla neuroblasts of different ages are orderly aligned on the lateral to medial spatial axis with the youngest on the lateral side (*Figure 1A*). Medulla neuroblasts divide asymmetrically multiple times to self-renew and to generate a series of GMCs. At the same time, they transit through the TTF sequence (*Figure 1A*). Among the TTFs, Slp1 and Slp2 are two homologous fork-head family transcription factors expressed in the same temporal pattern in medulla neuroblasts (*Figure 1B–B'''*). We have chosen the Ey to Slp1/2 transition as a focal point because in the earliest studies that reported temporal patterning of medulla neuroblasts, Slp was the first medulla TTF whose regulatory relationship with the previous TTF Ey was clearly understood from genetic studies: Slp expression is activated by Ey and Slp in turn represses Ey expression (*Suzuki et al., 2013*; *Li et al., 2013*). In this work, we identified two *cis*-regulatory enhancer elements that regulate the temporal expression pattern of Slp1/2, and showed that the previous TTF Ey, another TTF Scro, and the Notch signaling pathway directly regulate the transcription of *slp* through these enhancers. Slp1/2 expression is delayed when N signaling is lost or when cell cycle progression is blocked which also causes loss of N signaling. Furthermore, we show that supplying transcriptionally active Notch can rescue the timely Ey to Slp transition in cell-cycle arrested neuroblasts. Thus, our work demonstrates that cell cycle-dependent Notch signaling cooperates with TTFs to promote the progression of the TTF transcriptional cascade.

## Results

### The expression of Slp1 and Slp2 in medulla neuroblasts is regulated at the transcriptional level by the coordinated function of two enhancers

The observed distribution of temporal patterning transcription factors may be regulated at either the transcriptional or post-transcriptional level. The protein expression pattern of TTFs is not necessarily the same as that of their mRNA transcripts. A case in point is the expression pattern of the SoxN protein recently reported to function alongside Dmrt99B and Hth to specify identities of early-born neurons in the medulla; while mRNA transcripts of the *soxN* gene are widely distributed in neuroblasts, the SoxN protein is only expressed in the youngest neuroblasts, suggesting that post-transcriptional mechanisms are at work to confine the actual domain of SoxN protein expression (*Zhu et al., 2022*). To examine whether the expression of Slp1/2 in medulla neuroblasts is regulated at transcription, we observed the distribution of *slp1* and *slp2* mRNA transcripts by fluorescence in situ-hybridization (*Figure 1C–D''''*). The observed patterns of *slp1* and *slp2* mRNA expression closely parallel the expression of Slp1 and Slp2 proteins (*Figure 1B–D''''*), and are also corroborated by scRNA-seq data (*Zhu et al., 2022*), indicating that the expression of *slp1* and *slp2* is directly regulated at transcription.

We then sought to identify transcriptional enhancers that enabled the expression of the *slp1 and slp2* genes in the characteristic pattern observed in medulla neuroblasts. We examined patterns of GFP expression driven by putative enhancer elements (*Pfeiffer et al., 2008*; *Jenett et al., 2012*; *Pfeiffer et al., 2010*) (available from the Janelia FlyLight image database https://flweb.janelia.org/cgi-bin/flew.cgi), in *Drosophila* third instar larval brains. We identified the *GMR35H02 Gal4* line as containing a candidate enhancer driving Slp1/2 expression in medulla neuroblasts (*Figure 2A*). We then verified the expression of *UAS-GFP* driven by *GMR35H02-Gal4* and found that, indeed the GFP expression in medulla neuroblasts was initiated at the same time as endogenous Slp1 and Slp2 (*Figure 2—figure supplement 1A-A''''*).

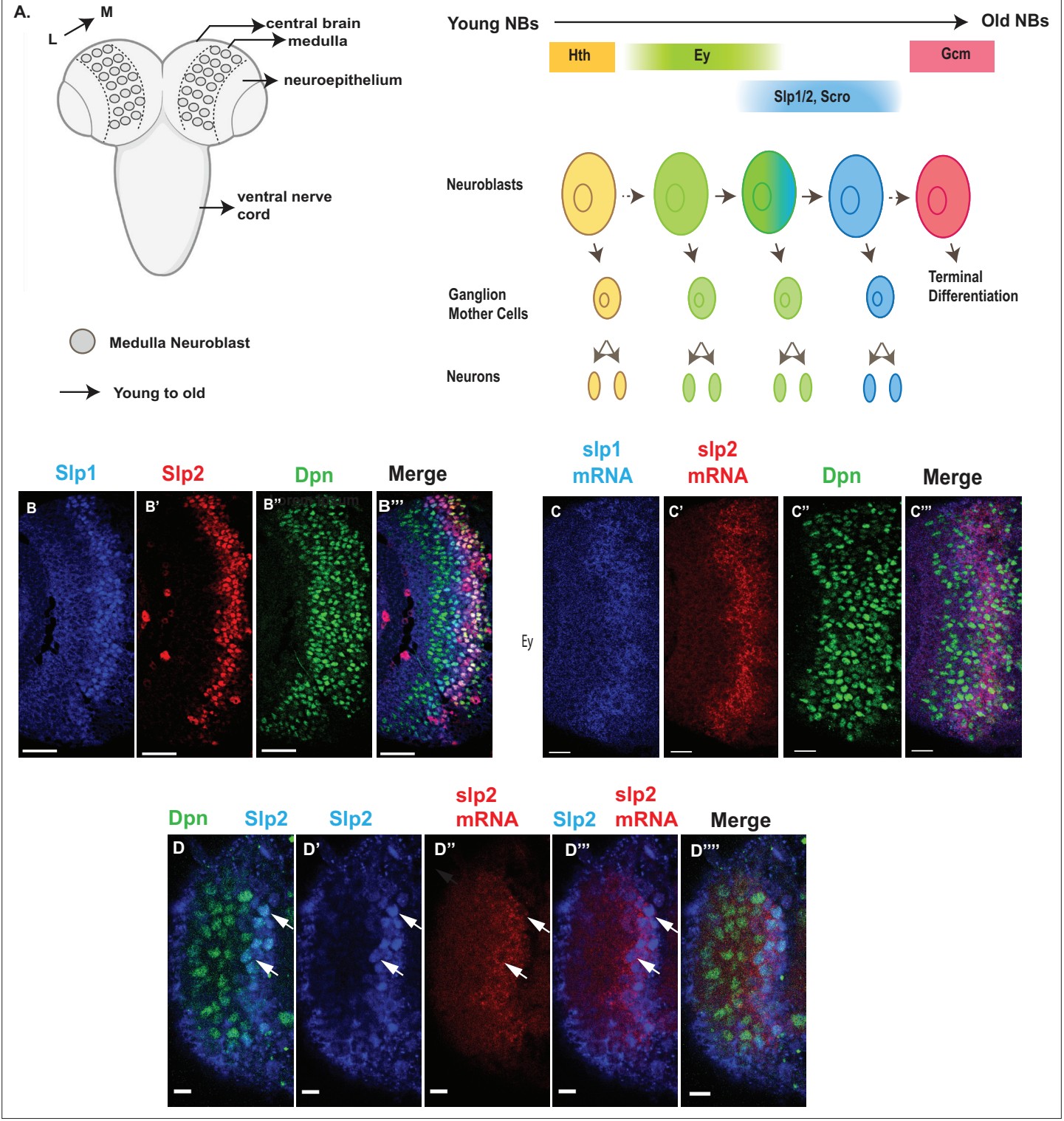

**Figure 1.** Expression of *slp1* and *slp2* genes in medulla neuroblasts is controlled at transcription. (**A**) Schematic of *Drosophila* brain at the third instar larval stage highlighting the location of the optic lobe medulla. Medulla NBs (shown as circles), located on the surface of the optic lobe, are transformed from neuroepithelial cells (NE) as a neurogenesis wave spreads in a medial to lateral direction. As a result, medulla NBs of different ages are aligned on the lateral (**L**) to medial (**M**) spatial axis. Schematic to the right shows part of the temporal patterning program of medulla NBs with a focus on the Ey to Slp1/2 transition. NBs undergo asymmetric divisions to self-renew and to produce intermediate progenitors called Ganglion Mother Cells that each divide once to produce two neurons. In the Ey stage, NBs undergo a few divisions while gradually activating Slp expression resulting in a significant overlap between Ey and Slp expression in NBs. After Slp level reaches a certain threshold, Ey expression is down-regulated and eventually deactivated

*Figure 1 continued on next page*

Figure 1 continued

completely. This process repeats as subsequent temporal patterning factors are activated and earlier factors down-regulated over several neuroblast division cycles. After several temporal stages that are not shown here indicated by the dashed arrow, neuroblasts express TTF Gcm and exit the cell cycle. The larval brain graphic was created using BioRender. (**B-B'''**) Expression patterns of endogenous Slp1 and Slp2 proteins in medulla neuroblasts identified by their expression of Deadpan (Dpn), a neuroblast marker. (**C-C'''**) Expression patterns of Slp1 and Slp2 mRNAs in Dpn expressing medulla neuroblasts closely parallels the corresponding protein expression patterns. (**D-D''''**) Detection of *slp2* mRNA and Slp2 protein in the same brain shows spatial co-localization of *slp2* transcripts and Slp2 protein in the same neuroblasts. Two distinct neuroblasts indicated by arrowheads shown for emphasis. These cells express the neuroblast marker Dpn (**D, D''''**) and Slp2 in the nucleus and the *slp2* mRNA is localized to the cytoplasm. Scale bar for panels (**B-B'''**) and (**C-C'''**): 20 µm. Scale bar for (**D-D''''**) 6 µm.

Subsequently, we looked for conserved sequences within the GMR35H02 fragment to narrow down the sequence element containing the potential active enhancer of Slp1/2 expression. Previous cross-taxa comparative studies have indicated that in many cases, circuits regulating critical patterning processes are conserved across vast evolutionary distances (*Brody et al., 2007*; *Awgulewitsch and Jacobs, 1992*; *Malicki et al., 1992*; *Wray et al., 2003*). Therefore, we reasoned that enhancers of genes essential for temporal patterning might be conserved at least across various related *Drosophila* species. We compared using Evoprinter (*Odenwald et al., 2005*) the sequence within *GMR35H02* DNA element across twelve *Drosophila* species. This comparison revealed three broadly conserved segments within *GMR35H02* of sizes 2.1 kbp (R2.2), 973 bp (M1), and 1.5 kbp (L1.5) (*Figure 2A* shows their relative positions within the *GMR35H02* sequence). For the first iteration of enhancer bashing experiments, we cloned these three sequences into pJR12 vector that expresses an eGFP reporter gene from a basal *hsp70* promoter (*Rister et al., 2015*) and made transgenic flies (See primers in *Supplementary file 1*). All DNA constructs containing the enhancer fragments were inserted at the same genomic landing site by φC31 transgenesis, to minimize variation due to genomic position differences and to ensure direct comparability between different reporters. The transgenic line containing the 1.5 kbp enhancer fragment named SlpL1.5 expressed GFP in the same pattern observed for *GMR35H02-Gal4* (*Figure 2—figure supplement 1B-B''*), indicating that the active enhancer of *slp1* and *slp2* genes in *GMR35H02* is located within this smaller 1584 bp sequence. We then searched for smaller conserved DNA segments within this 1584 bp sequence by Evoprinter and identified three even smaller conserved sequences of sizes 220 bp, 592 bp, and 123 bp, respectively, that we named Slpf1 (220 bp), Slpf2 (592 bp), and Slpf3(123 bp)(*Figure 2A*). As earlier, we cloned these three smaller DNA elements into the pJR12 vector and generated transgenic flies expressing the GFP reporters from an identical chromosomal location. Transgenic flies containing the 220 bp fragment (Slpf1) expressed GFP very similar to the *GMR35H02-Gal4* line we had earlier observed, while the other two did not show any expression (*Figure 2B–B'''*; *Figure 2—figure supplement 1C-E''*), indicating that we had narrowed down the active enhancer within the *GMR35H02* sequence to a 220 bp element.

However, the deletion of this element by CRISPR-Cas9 failed to eliminate the expression of Slp1 and Slp2 in neuroblasts (*Figure 2—figure supplement 2A-A'''*), though it noticeably reduced the expression domain of Slp1 (*Figure 2—figure supplement 2C*). Given that the expression of developmentally regulated genes is often controlled by multiple enhancers (*Frankel et al., 2010*; *Fujioka and Jaynes, 2012*) and that enhancer redundancy is important for achieving robust developmental outcomes (*Frankel et al., 2010*; *Ghiasvand et al., 2011*; *Lagha et al., 2012*), this indicated that other enhancers might act redundantly with this 220 bp enhancer to regulate Slp1/2 expression in medulla neuroblasts. To find other enhancers of *slp* expression, we scanned through the REDfly database (*Gallo et al., 2006*; *Rivera et al., 2019*) – a curated repository of enhancers reported for *Drosophila melanogaster* genes. An earlier study had conducted an exhaustive analysis of a 30 kbp genomic region surrounding the *slp* transcription units to understand better the regulatory networks underlying *slp* expression during embryonic segmentation and to identify *cis*-regulatory elements driving *slp1 /2* expressions at parasegment boundary formation (*Fujioka and Jaynes, 2012*). In this study, segments of regulatory DNA within 30 kbp of *slp1* and *slp2* transcription start sites, averaging 2kbp in size, were incorporated in a *lacZ* reporter coupled to an *even-skipped* promoter, and patterns of lacZ expression were then observed in transgenic flies (*Figure 2A*, *Table 1*). We obtained transgenic lines used in this study (kind gifts from Dr. Miki Fujioka and Dr. James B. Jaynes) and then tested by antibody staining whether the expression of *lacZ* driven by these enhancers colocalized with endogenous Slp1 and Slp2 proteins in the medulla (*Figure 2—figure supplement 3*). Through this screen

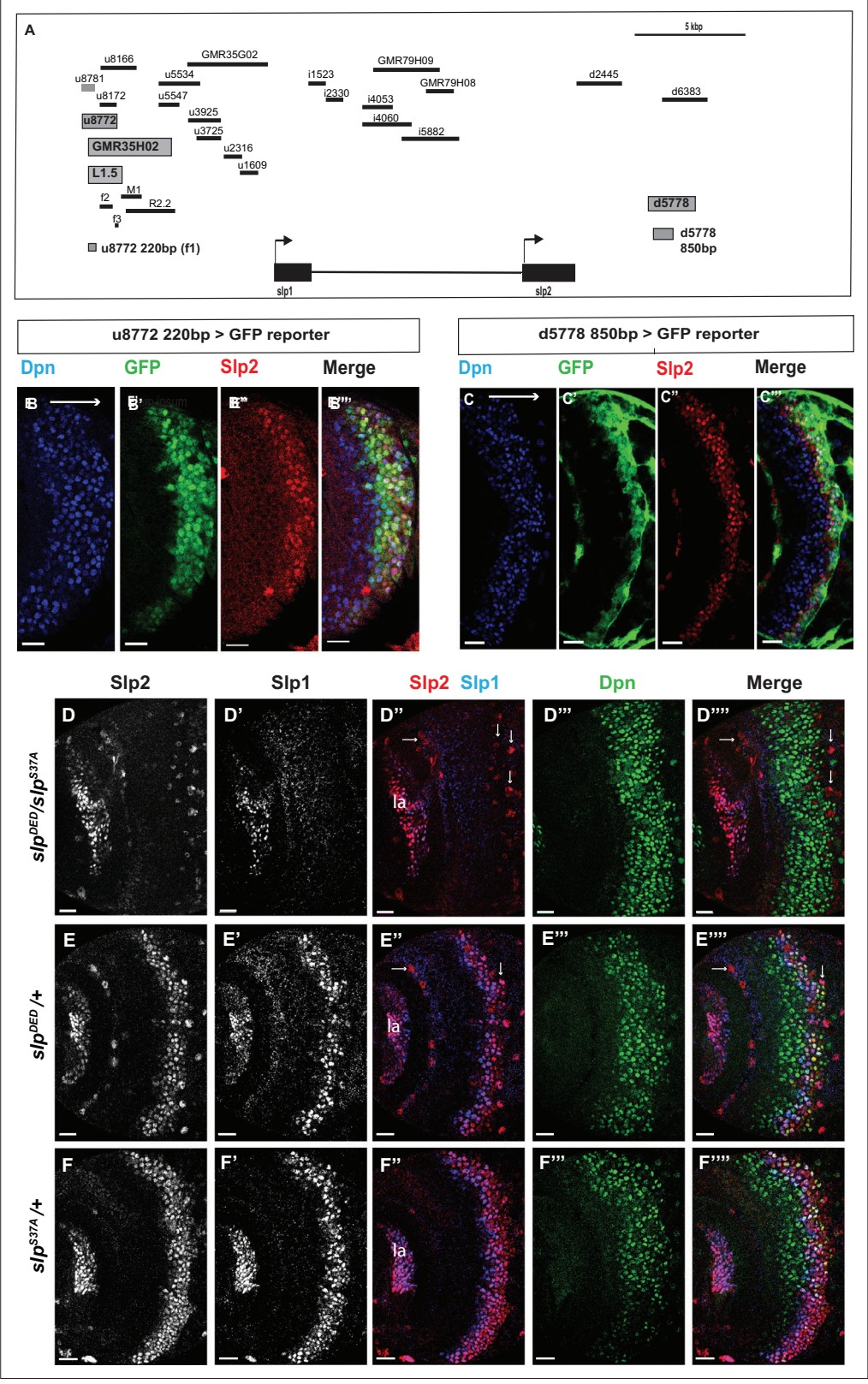

**Figure 2.** Transcription of *slp1* and *slp2* in medulla neuroblasts is regulated by two enhancers of lengths 220 bp and 850 bp, respectively. (**A**) A schematic of the enhancer screening. *slp1* and *slp2* genes and transcriptional enhancers identified as regulating their expression are shown as black and grey boxes, respectively. Positions of slp1 coding locus (2 L:3,825,675.3,827,099 [+]), slp2 coding locus (2 L:3,836,840.3,839,185 [+]),

*Figure 2 continued on next page*

*Figure 2 continued*

GMR35H02 (2 L:3,817,265.3,821,014) and the REDfly enhancers u8772 (2 L:3,816,967.3,818,532) and d5778 (2 L:3,842,530.3,844,660) are shown relative to one another. A 220 bp enhancer located within the genomic segment covered by GMR35H02 and an 850 base pair enhancer element within the REDFly enhancer d5778 were identified as potential cis-regulatory elements of *slp1* and *slp2* genes. The distance between the start of GMR35H02 and the end of d5778 is around 27.394 kbp. Other fragments that were screened /cloned are shown as black bars with names on top. (**B-B′′′**) A 220 bp enhancer element located within the REDfly enhancer u8772 activates GFP expression in medulla neuroblasts in the same pattern as endogenous Slp1 and Slp2 in reporter assays. (**C-C′′′**) Reporter GFP expression driven by an 850 bp enhancer segment located within the REDfly enhancer d5778 also closely coincides with the expressions of endogenous Slp1 and Slp2, although it is initiated at a slightly later temporal stage than the 220 bp enhancer. (**D-D′′′**) A CRISPR-Cas9 edited chromosome 2 with both enhancers deleted (indicated as $Slp^{DED}$) when placed over the $Slp^{S37A}$ chromosome results in loss of Slp1 and Slp2 expression in medulla neuroblasts in affected flies (n=11). The effect is confined to neuroblasts, as Slp2 expression is retained in both lamina (la) and in surface glial cells (arrows), and Slp1 is also seen in laminar cells. This also confirms that coding sequences of *slp1* and *slp2* genes are unaffected by the CRISPR-Cas9 editing procedure. Control experiments where $Slp^{DED}$ was placed against a wild-type chromosome 2 (indicated by '+') (**E-E′′′**) (n=9) and where the $Slp^{S37A}$ chromosome was placed against a wild-type chromosome 2 (**F-F′′′**) (n=11) show normal expression of Slp1 and Slp2 in medulla neuroblasts. Scale bars 20 μm.

The online version of this article includes the following source data and figure supplement(s) for figure 2:

**Figure supplement 1.** The identification of the 220 bp enhancer by promoter bashing.

**Figure supplement 2.** The two enhancers act partially redundantly.

**Figure supplement 2—source data 1.** Quantification and comparison of Dpn expression in wild type control, u8772 enhancer deletion, and d5778 enhancer deletion brains.

**Figure supplement 2—source data 2.** Quantification and comparison of Slp1 expression in wild type control, u8772 enhancer deletion, and d5778 enhancer deletion brains.

**Figure supplement 2—source data 3.** Quantification and comparison of Slp2 expression in wild type control, u8772 enhancer deletion, and d5778 enhancer deletion brains.

**Figure supplement 2—source data 4.** Expression level quantification of Dpn, Slp1 and Slp2 in wild type control, u8772 enhancer deletion, and d5778 enhancer deletion brains.

**Figure supplement 3.** Screen of REDfly enhancers identifies d5778 as an enhancer of slp1/2 transcription in medulla neuroblasts.

**Figure supplement 4.** The d5778 850 bp enhancer drives reporter expression predominantly in neuroblasts.

we identified a second regulatory segment corresponding to the REDfly enhancer d5778 (*Figure 2A*) that drove GFP expression in medulla neuroblasts within the endogenous Slp1/2 expression domain (*Figure 2—figure supplement 3Q-Q″*). Since the overlapping d6383 did not express GFP in medulla neuroblasts (*Figure 2—figure supplement 3R-R″*), we narrowed down the active enhancer's location to an 850 bp segment within the d5778 enhancer (*Figure 2C–C′′′*). Reporter expression driven by this second enhancer (hereafter referred to as d5778 850 bp enhancer) is initiated at a slightly later time compared to the first identified enhancer, and it is in older Slp expressing neuroblasts and also in glial cells at the junction of the medulla and the central brain. To confirm that the d5778 850 bp driven reporter is expressed in medulla neuroblasts, we used *dpn*-Gal4 to drive *UAS-GFP-RNAi* in neuroblasts, and this eliminated GFP expression in medulla neuroblasts. In contrast, GFP expression in neuroblasts is not affected when we use *repo*-Gal4 to drive *UAS-GFP-RNAi* in glia (*Figure 2—figure supplement 4*). Our analysis of the REDfly enhancers that could potentially regulate expression of Slp1/2 in the medulla also revealed that the 220 bp enhancer segment identified earlier is contained within another REDfly enhancer named u8772 (*Figure 2A*). Thus, we refer to the first identified enhancer hereafter as u8772 220 bp. As in the case of the u8772 220 bp enhancer deletion, sole deletion of the d5778 enhancer by CRISPR-Cas9 did not abolish Slp1/2 expression (*Figure 2—figure supplement 2B-B′′′,C*), suggesting that these two enhancers function redundantly and may make additive contributions to attaining the observed levels of Slp1 and Slp2 expression.

To test whether a combination of these two enhancers is required to regulate Slp1 and Slp2 expression in medulla neuroblasts, we used CRISPR-Cas9 to delete the second enhancer in the strain that had the first enhancer deleted. Since the fly strain containing deletions of two enhancers was homozygous lethal, we placed the double enhancer deletion line ($slp^{DED}$) over a deficiency line ($slp^{S37A}$)

**Table 1.** Fragments screened or cloned for identifying enhancers of *slp1/2* transcription in medulla neuroblasts.

Table summarizing names and chromosomal locations of enhancer fragments screened or cloned. Those with names beginning with GMR are Janelia FlyLight Gal4 lines. Enhancer fragments with names beginning in u, i, or d are enhancers from the study (*Fujioka and Jaynes, 2012*).

| Screened fragments | Start base | End base | Expression in medulla neuroblasts |
|---|---|---|---|
| GMR35H02 | 3817265 | 3821014 | Yes |
| GMR79H09 | 3830094 | 3833128 | No |
| GMR35G02 | 3821714 | 3825360 | No |
| GMR79H08 | 3832497 | 3833703 | No |
| u8772 | 3816967 | 3818532 | Yes |
| u8781 | 3816967 | 3817608 | Yes |
| u8172 | 3817605 | 3818532 | No |
| u8166 | 3817605 | 3819171 | Mainly in neurons |
| u5534 | 3820168 | 3822240 | unspecific broad expression |
| u5547 | 3820168 | 3821009 | weak expression |
| u3925 | 3821751 | 3823158 | No |
| u3725 | 3822001 | 3823158 | No |
| u2316 | 3823356 | 3824032 | No |
| u1609 | 3824033 | 3824739 | No |
| i1523 | 3827196 | 3827972 | No |
| i2330 | 3827970 | 3828686 | No |
| i4053 | 3829629 | 3830998 | No |
| i4060 | 3829629 | 3831819 | No |
| i5882 | 3831440 | 3833928 | No |
| d2445 | 3839285 | 3841364 | No |
| d5778 | 3842530 | 3844660 | Yes |
| d6383 | 3843114 | 3845187 | No |
| Cloned fragments | Start base | End base | Expression in medulla neuroblasts |
| 35H02-L1.5 | 3817261 | 3818808 | Yes |
| 35H02-M1.0 | 3818785 | 3819757 | No |
| 35H02-R2.2 | 3818872 | 3821014 | No |
| slpf1-220bp (u8772 220 bp) | 3817298 | 3817517 | Yes |
| slpf2-592bp | 3817782 | 3818373 | No |
| slpf3-123bp | 3818686 | 3818808 | No |
| d5778 850 bp | 3842770 | 3843619 | Yes |

that lacks the *slp1* transcription unit and all intervening sequences up to 16 bp downstream of the *slp2* transcriptional start site (*Sato and Tomlinson, 2007*), and examined Slp1/2 expression in larval brains. In such brains, both Slp1 and Slp2 expression are lost specifically in medulla neuroblasts and their progenies, but their expression in lamina neurons and glia are not affected (*Figure 2D–D''''*). As controls, heterozygous *slp*^S37A or *slp*^DED brains show wild-type Slp1 and Slp 2 expression patterns (*Figure 2E–F''''*). This result confirms that these two enhancers acting together are necessary and specific enhancers for Slp1 and Slp2 expressions in medulla neuroblasts and progeny.

## Bio-informatics analysis of enhancers identified binding sites for transcription factors Ey, Slp1, Scro/Vnd and Su(H)

We analyzed the sequences of the u8772 220 bp and the d5778 850 bp enhancers using a web-based tool that uses the MEME suite (*Bailey et al., 2015*) application FIMO (*Grant et al., 2011*) for predicting potential transcription factor binding sites (Dr. Bart Deplancke and Michael Frochaux, personal communication). This tool can be accessed at https://biss.epfl.ch. This initial analysis predicted the occurrence of putative binding sites for the previous TTF Ey, the CSL transcription factor Su(H), Slp1 itself, and an Nk2-class homeodomain transcription factor Ventral nervous system defective (Vnd) (*Figure 3*, *Table 2*, *Supplementary file 3*) in our enhancers. The previous TTF Ey was shown to be required but not sufficient for activating Slp genes' expression in medulla neuroblasts (*Li et al., 2013*), but it was unclear whether this regulation is direct. Vnd is not expressed in medulla neuroblasts according to our single-cell RNA-sequencing data (*Zhu et al., 2022*), but a related homologous NK-2 homeobox transcription factor Scarecrow (*Zaffran et al., 2000*) (Scro) is expressed around the same time as Slp1 (*Zhu et al., 2022*). Also, the loss of Scro leads to significantly reduced Slp expression level (*Zhu et al., 2022*). The CSL transcription factor Su(H) is the *Drosophila* homolog of RBP-J (*Bailey and Posakony, 1995*; *Furukawa et al., 1995*) and the critical DNA-binding component of the active Notch transcriptional complex. The Notch signaling pathway plays an overarching role as a master regulator of development and differentiation in neuroblasts, but it has not been shown to regulate the expression of temporal patterning genes in neural progenitors. Therefore, we next tested whether Slp1/2 expression is regulated by Notch signaling.

## The Notch pathway regulates Slp1 and Slp2 expression in medulla neuroblasts

We observed the expression of Slp1 and Slp2 in RNAi knockdown clones of key Notch pathway components induced by *ay-Gal4* (*actin >FRT-y⁺-STOP-FRT-Gal4*, in which *actin* promoter drives Gal4 expression only after a STOP cassette is excised by the action of heat shock activated Flippase *Ito et al., 1997*). Since *slp1* and *slp2* are two homologous genes located in close proximity in the genome (*Nüsslein-Volhard et al., 1984*) and are expressed in similar patterns with Slp1 expression slightly preceding Slp2 (*Figure 1B–B‴*), in all subsequent figures we have shown data for Slp2 assuming that Slp1 expression follows the same pattern. On inducing RNAi knockdowns of the critical Notch pathway components Notch and Su(H) – members of the tripartite core of the active Notch transcriptional complex (*Bray, 2006*), it is seen that Slp2 expression is delayed within the RNAi knockdown clones marked by GFP expression (*Figure 4A–A‴ and B–B‴* respectively) compared to wild-type neuroblasts of the same age (i.e. in the same vertical 'stripe'). Since the lateral to medial spatial axis is a proxy of the neuroblast age, we counted the number of neuroblast when Slp2 is first turned on. In control regions, Slp2 is turned on at the 4th- 5th neuroblast from the lateral edge, while in N-RNAi clones (n=12), Slp2 is turned on at the 10th-15th neuroblast. Similarly, Slp2 expression initiates between the 7th and 10th neuroblasts inside Su(H)-RNAi clones (n=11). A similar effect is seen when a dominant-negative variant of the third core component of the Notch transcriptional complex Mastermind (Mam) is expressed under the control of *ay-*Gal4 (*Figure 4C–C‴*). Within *mamDN* expressing clones (n=6), Slp2 is initiated between the 7th and 10th neuroblast as with *Su(H) RNAi* clones. These data suggest that timing of Slp1/2 expression in medulla neuroblasts is regulated by the Notch pathway.

To identify Notch's cognate ligand that activates Notch signaling in medulla neuroblasts, we examined expression of known Notch ligands- Delta (*Campos-Ortega, 1988*), Serrate (*Fleming et al., 1990*; *Rebay et al., 1991*) and Weary (*Kim et al., 2010*) in medulla neuroblasts (data not shown). Of these only *UAS-GFP* driven by *delta-Gal4* was seen to be expressed in medulla neuroblasts and in their progeny GMCs. We then examined whether loss of Dl recapitulates the effects we observe on Slp1/2 expression with loss of function of core Notch transcription complex components, as would be expected if Dl is required for activating Notch signaling in these cells. In neuroblasts within Dl-RNAi expression clones, Slp2 expression is initiated at the 6th-10th neuroblasts from the medulla lateral edge (*Figure 4D–D‴*) (n=5), consistent with our observations of Slp2 expression in mutants of other Notch pathway components discussed above.

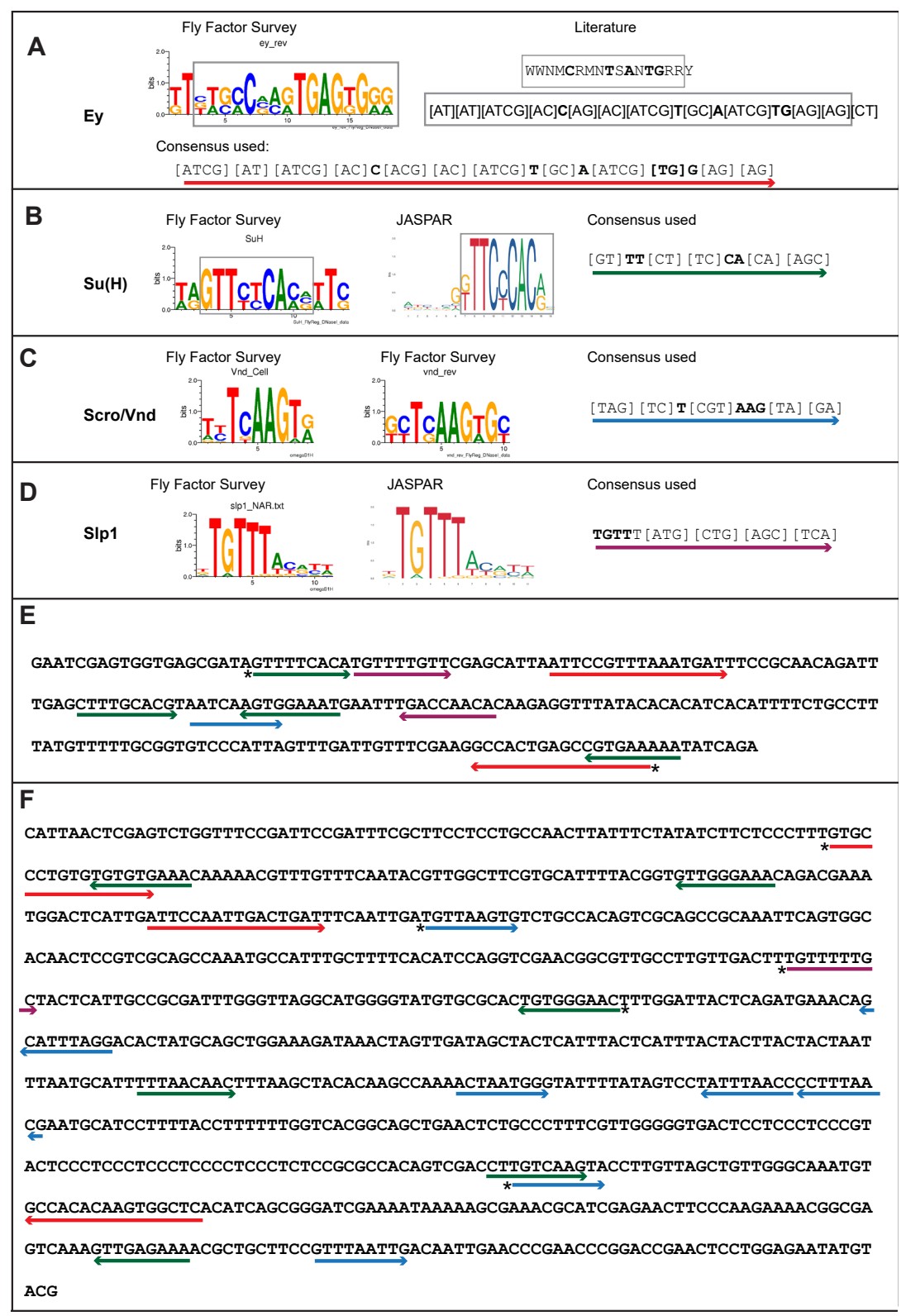

**Figure 3.** Predicted binding sites for Ey, Su(H), Scro, and Slp1 in the two enhancers. (**A–D**) Position weight matrix of for Ey (**A**), Su(H) (**B**), Scro /Vnd (**C**) and Slp1 (**D**) from Fly Factor Survey, JASPAR and literature are considered together to make a consensus sequence. Arrows of different colors are underlying these consensus binding sites. Critical invariant bases are in bold. Grey boxes indicate the corresponding positions of the Position weight matrix from two sources. (**E,F**) Predicted binding sites for Ey, Su(H), Scro, and Slp1 in the 220 bp enhancer (**E**) and 850 bp enhancer (**F**). Arrows are color

*Figure 3 continued on next page*

*Figure 3 continued*

coded as in (**A–D**). Right-pointing arrow indicates the binding site is on direct strand, while left-pointing arrow indicated the binding site is on reverse strand. Star* indicates that the binding site is also predicted by the FIMO pipeline.

The online version of this article includes the following figure supplement(s) for figure 3:

**Figure supplement 1.** NK-2 family transcription factors with known binding consensus motifs were used to identify potential Scro binding sites in the d5778 850 bp enhancer.

## Genetic interactions between Notch, Ey, and Scro in the regulation of Slp expression

Since Ey is required for Slp expression, the delay of Slp expression with loss of N signaling might be due to a delay in earlier temporal transitions resulting in late Ey onset, or a specific delay in the Ey to Slp transition, or both. To test this, we examined the expression of Ey with loss of N pathway components. Ey is normally turned on in the 2nd-3rd neuroblast. In *N-RNAi* clones, Ey is turned on in the 5th-8th neuroblast (n=6) (***Figure 5A1-A8***), suggesting that Notch signaling is required for the timely transition into the Ey stage. However, this does not preclude an additional direct role for Notch signaling in the Ey to Slp transition. To examine this, we counted the number of Ey +neuroblast in which the transition to Slp stage occurs. We observed that in wild-type neuroblasts, Slp2 starts to be expressed at a weak level in the 2nd-3rd Ey +neuroblast, and the level becomes stronger one or two neuroblasts later and then remains strong for several neuroblasts, thus the transition to the Slp stage occurs in the 3rd –4th Ey +neuroblast, and after the transition, Ey level starts to decrease because of repression by Slp. In *N-RNAi* clones, the transition to the Slp stage occurs in the 6th-9th Ey +neuroblast (n=6) (***Figure 5A1-A8***). In *Su(H)* mutant clones, the transition occurs after the 6th-8th Ey +neuroblasts (n=5)(***Figure 5B–C"***). To make sure the delayed Slp transition is not mediated through Ey with loss of Su(H), we supplied Ey using a *dpn*-Gal4 and UAS-Ey in *Su(H)* mutant clones. In such clones, Ey is over-expressed, but the transition to Slp stage is still delayed to the 6th-7th Ey +neuroblast (n=6) (***Figure 5D–E"***). These results suggest that Notch signaling also facilitates the Ey to Slp transition in addition to an earlier indirect role through Ey.

We next examined if the expression of earlier TTFs or later TTFs is also affected by loss of N signaling. One of the first TTFs, SoxN, is not significantly affected in *Su(H)-RNAi* clones (***Figure 5— figure supplement 1A-A'''***). Opa, the TTF upstream of Ey, is normally expressed in two stripes. In *Su(H)-RNAi* clones, the first Opa stripe appears expanded, while the second stripe is delayed (***Figure 5—figure supplement 1B-B'''***). Furthermore, the expression of later factors D and BarH1 are severely delayed or mostly lost when we knocked down Su(H) (***Figure 5G–H'***). Thus N signaling may affect multiple temporal transitions.

Since Notch signaling is active throughout the neuroblast life, and may facilitate multiple temporal transitions, we reasoned that Notch signaling may require the previous TTF for specificity. Therefore, we tested if supplying active Notch signaling can rescue Slp expression when Ey is knocked down. Supplying NICD using a *dpn >FRT-stop-FRT3-FRT-NICD* transgene (which drives the expression of NICD from the *dpn* enhancer in presence of heat shock activated Flippase; see Materials and methods) was not sufficient to rescue Slp expression in *ey-RNAi* clones (***Figure 5I–I"***). This result is consistent with our hypothesis that Notch signing, as a general signaling pathway active in all neuroblasts, requires the previous TTF Ey for its specificity to facilitate Slp expression and promote the transition. Furthermore, overexpression of Scro was also not sufficient to rescue Slp expression in *ey-RNAi* clones (***Figure 5J–J"***). Taken together, these results suggest that the previous TTF Ey is required to initiate Slp expression, while Notch signaling and Scro require Ey to further activate Slp expression and facilitate the timely transition to the Slp stage.

## Notch pathway and Ey regulate the activity of *slp* enhancers

Since Su(H) and Ey are required for regulate Slp expression, we next tested whether these regulations are mediated through the identified enhancers. We tested whether knockdown of each factor would affect GFP reporter expression driven by these enhancers. We reasoned that if any of the factors we had identified binding sites for bound to the enhancers and activated Slp1/2 transcription, GFP reporter expression driven by these enhancers would be lost in the loss of function clones of these transcription factors. We induced RNAi knockdown clones for Ey and Su(H) using *ay-Gal4* in transgenic

**Table 2.** List of predicted binding sites and mutagenesis strategy.

Sequences of 21 bp fragments including the underlined binding sites are shown. The critical invariant bases are in bold. For sites on the reverse strand (-), the reserve strand sequence is shown. For deletions, all 21bps are deleted. Mutated bases are shown as lower-case letters. * indicates that the binding site is predicted by the FIMO pipeline.

| Site Name | Strand | Original Sequence | Mutate to | Constructs (Size in bp) |
|---|---|---|---|---|
| 220-Ey-Site1 | + | TAATTCCGTTTAAATGATTTC | Deleted | 220-Ey-1d (199 bp) |
| 220-Ey-Site2* | - | TTTTCACGGCTCAGTGGCCTT | TTTTCACGGCTatGTGGCCTT | 220-Ey-2m (220 bp) |
| 220-SuH-Site1* | + | GGTGAGCGATAGTTTTCACAT | GGTGAGCGActaggagacgAT | 220-SuH-1d has the first site deleted. 220-SuH-1–4 m (220 bp) has all four sites mutated. 220-SuH-1m4m (220 bp) has site 1 and site 4 mutated. |
| 220-SuH-Site2 | + | TTTGAGCTTTGCACGTAATCA | TTTGAGCTTaGactGTAATCA | |
| 220-SuH-Site3 | - | AAATTCATTTCCACTTGATTA | AAATcCAcTTCCACTTGATTA | |
| 220-SuH-Site4 | - | CTGATATTTTCACGGCTCAG | CTGActcgcTTtACGGCTCAG | |
| 220-Scro-Site1 | + | CACGTAATCAAGTGGAAATGA | CACGTAATCgtGTGGAAATGA | 220-Scro-1m (220 bp) |
| 220-Slp1-Site1 | + | TTCACATGTTTTGTTCGAGCA | TTCACATGTaccaagCGAGCA | 220-Slp1-1m2m has both sites mutated (220 bp) |
| 220-Slp1-Site2 | - | CTCTTGTGTTGGTCAAATTCA | CTCTTGgagTcGatcAATTCA | |
| 850-Ey-Site1* | + | TTGTGCCCTGTGTGTGTGAAA | aTGTGgCtTGTGTGTGTGAAA | 850-Ey-1m2d3d has site 1 mutated, and sites 2 and 3 deleted (808 bp). |
| 850-Ey-Site2 | + | TGATTCCAATTGACTGATTTC | Deleted | |
| 850-Ey-Site3 | - | ATGTGAGCCACTTGTGTGGCA | Deleted | |
| 850-SuH-Site1 | - | TTTTTGTTTCACACACACAGG | TaTcTGTcgCACACACACAGG | 850-SuH-1m2m3m6m has sites 1, 2, 3 and 6 mutated (850 bp). 850-SuH-1–6 m has sites 1–6 mutated (850 bp). |
| 850-SuH-Site2 | - | CGTCTGTTTCCCAACACCGTA | CGTCTGTaTCCgtACACCGTA | |
| 850-SuH-Site3* | - | TCCAAAGTTCCCACAGTGCGC | TCCAAAGaTCCgACAGTGCGC | |
| 850-SuH-Site4 | + | GCATTTTTAACAACTTTAAGC | GCATTTcTAAgtACTTTAAGC | |
| 850-SuH-Site5 | + | GTCGACCTTGTCAAGTACCTT | GTCGACCaTGTCAAGTACCTT | |
| 850-SuH-Site6 | - | AGCAGCGTTTTCTCAACTTTG | AGCAGCGTTaTCTgAACTTTG | |

*Table 2 continued on next page*

*Table 2 continued*

| Site Name | Strand | Original Sequence | Mutate to | Constructs (Size in bp) |
|---|---|---|---|---|
| 850-Scro-Site1* | + | TTGATGTTAAGTGTCTGCCAC | Deleted | 850-Scro-1-7d has all 7 sites deleted (724 bp) |
| 850-Scro-Site2 | - | CATAGTGTCCTAAATGCTGTT | Deleted | |
| 850-Scro-Site3 | + | CAAGCCAAAACTAATGGGTAT | Deleted | |
| 850-Scro-Site4,5 | - | TCGTTAAAGG GGTTAAATAGG | Deleted | |
| 850-Scro-Site6* | + | TTGTCAAGTACCTTGTTAGCT | Deleted | |
| 850-Scro-Site7 | + | CGTTTAATTGACAATTGAACC | Deleted | |
| 850-Slp1-Site1* | + | TGACTTTGTTTTTGCTACTCA | Deleted | 850-Slp1-1d (829 bp) |

flies expressing wild type GFP reporters driven by these two enhancers, and then compared the expression of Slp2 and GFP in wild-type neuroblasts and in neuroblasts where each of these TFs had been individually knocked down.

For both enhancers, GFP reporter expression and Slp2 expression were lost in *ey-RNAi* clones (*Figure 6A–B'''*), confirming an earlier observation that Ey is required for initiating Slp1/2 expression. It also strengthened the evidence that these two were indeed bona-fide enhancers activating Slp1/2 expression. Similarly, GFP reporter expression was lost or delayed in *su(H)-RNAi* clones for both enhancers (*Figure 6C–D'''*), consistent with the changes in Slp2 expression. Since Slp binding sites were also identified in the *slp* enhancers, we next tested the effect of loss of Slp1/2 function on the expression of our GFP reporters. To this end, we induced *slp1/2* loss of function (LOF) mutant clones in flies carrying the *GFP reporter and UbiRFP FRT40A*. In *Figure 6E–F' slp* LOF clones are visible as dark regions devoid of RFP whereas wild type neuroblasts are marked by RFP expression from the Ubiquitin promoter. Reporter expression driven by both enhancers within *slp* LOF neuroblasts is normal as in wild-type neuroblasts. Though this suggests that Slp1/2 are not required to initiate their own expression, it does not preclude a possible role of Slp1/2 in sustaining their own expression in older Slp1/2 stage neuroblasts or progeny, but the perdurance of the GFP reporter prevents us from testing this possible role. In summary, our genetic experiments confirmed that identified enhancers of *slp1/2* respond to activation by Ey and Su(H).

## Mutations of potential binding sites for Ey, Su(H) and Scro in *slp1/2* enhancers impair expression of the reporters

To further test whether the regulators we identified regulate Slp1/2 expression directly through binding to the enhancers, we set out to mutate all possible binding sites for each of the regulators-Ey, Su(H), Scro and Slp1, on each enhancer. We identified all potential matches to consensus binding motifs for these factors using the DNA Pattern Find Tool within the Sequence Manipulation Suite (*Stothard, 2000*; *Figure 3*, *Table 2*) and using the FIMO based web-application to predict transcription factor binding sites mentioned earlier (accessible at https://biss.epfl.ch) (please see Materials and methods for details). We adopted this combined approach instead of solely mutating FIMO predicted binding sites for a TF since mutation of FIMO predicted binding sites alone did not always result in GFP reporter expression patterns that were consistent with our observations from genetic experiments. For instance, in the u8772 220 bp enhancer, deletion of the Su(H) binding site that was predicted using FIMO (220-Su(H)-site1 in *Table 2*) did not result in a loss of reporter GFP expression (*Figure 7—figure supplement 1*). This contradicted our result from genetic experiments with the wild type u8772 220 bp GFP reporter, where GFP expression from the reporter is lost or delayed in *su(H)-RNAi* knockdown clones (*Figure 6C–C'''*). We suspected this lack of correspondence between

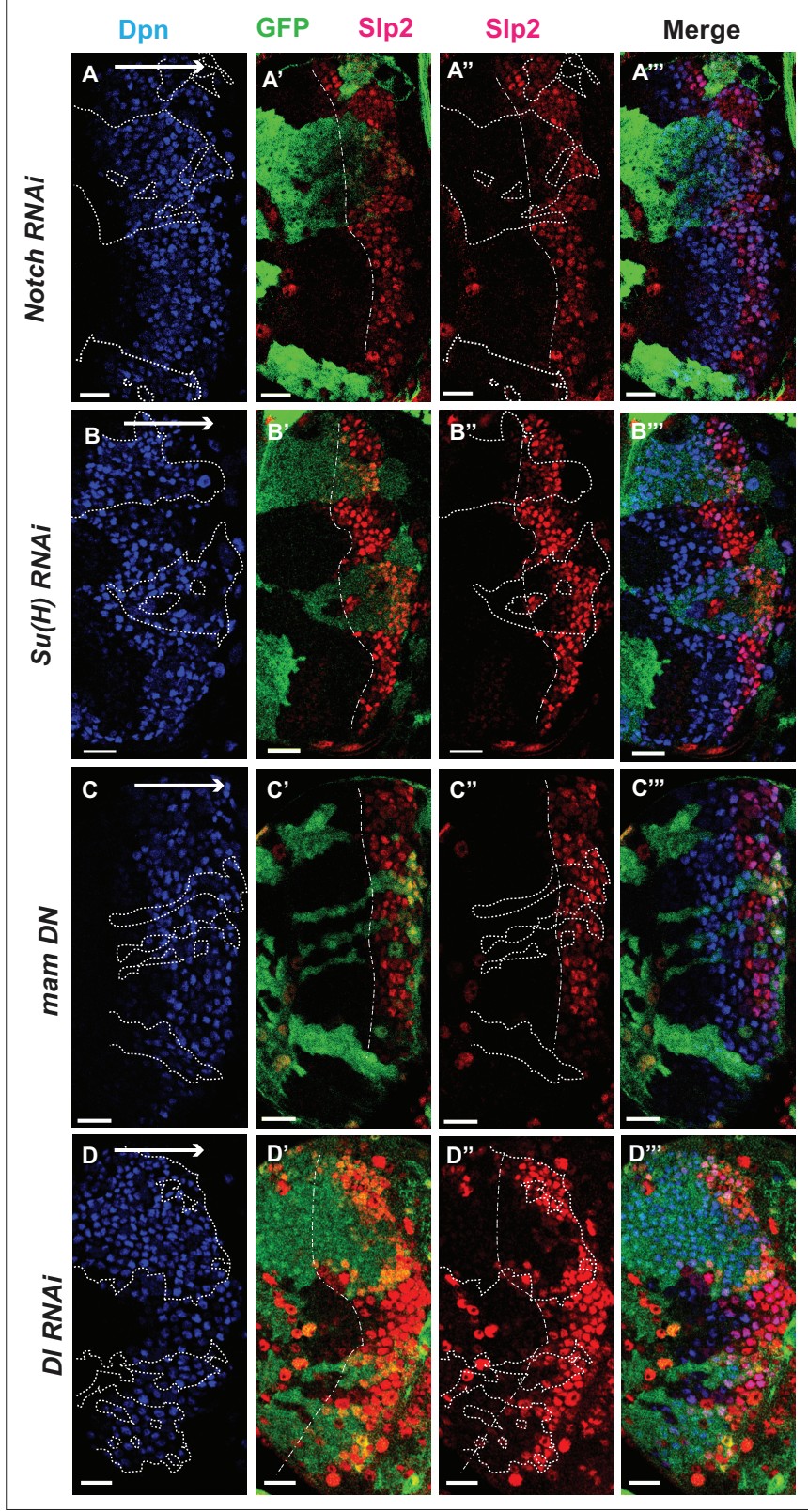

**Figure 4.** The Notch pathway regulates expression of *slp* genes in medulla neuroblasts. White arrow to the right indicates the neuroblast age from young to old. All clones are generated with *ay-Gal4 UAS-GFP* and clone boundaries are shown in dotted lines. The dash-dotted lines indicate where Slp2 should be turned on, and the curvature is due to the curvature of the Dpn initiation. (**A**) RNAi knockdown of Notch in GFP marked clones leads

*Figure 4 continued on next page*

*Figure 4 continued*

to a delay in expression of Slp2 compared to contemporaneous wild type neuroblasts outside the GFP marked clones. While in wild-type brains Slp2 expression is seen in the 4th-5th neuroblast from the lateral edge of the medulla, inside *N-RNAi* clones Slp2 expression is first noted in the 10th –15th neuroblast (n=12 clones). (**B**) In *Su(H) RNAi* clones marked by GFP, Slp2 expression begins at the 7th-10th neuroblast (n=11 clones). (**C**) In clones expressing the dominant negative mutant variant of Mastermind, Slp2 expression is also delayed and start at the 7th-10th neuroblasts (n=6 clones) (**D**) Knockdown of Dl, a Notch ligand, caused a similar delay in Slp2 expression, with Slp2 expression being initiated at the 6th to 10th NB (n=5 clones). Clonal data from at least three brains were observed for each mutant to infer delay in Slp2 expression initiation. Scale bar 20 μm.

phenotypes from genetic experiments and those observed from our transgenic GFP reporters could be explained by our failure to identify all potential transcription factor binding sites using FIMO alone.

The "DNA Pattern Find" pattern recognition tool accepts transcription factor (TF) binding consensus sequences and DNA sequences to search within for TF binding sites as inputs, and returns the locations of sites matching the consensus within the provided sequence. Consensus DNA sequences recognized by transcription factors were obtained for Ey, Su(H) and Slp1 from Fly Factor Survey, JASPAR databases and from published literature (for Ey) (*Tanaka-Matakatsu et al., 2015*; *Figure 3*). We noticed that predicted binding sites identified by FIMO such as the 220-Ey-Site2 and 850-Ey-Site1 often had minor mismatches to the consensus binding sequence (for example, the 220-Ey-Site2 had two mismatches to the consensus). Therefore, we also considered sites with a maximum of two mismatches on non-critical bases to our consensus sequences as potential binding sites for a particular transcription factor. For Scro, no consensus sequences could be found, hence we searched d5778 850 bp enhancer using available binding consensus sequences for a closely related NK-2 transcription factor Vnd and 19 other related NK-2 transcription factors (*Figure 3—figure supplement 1*) and selected seven most commonly occurring sequence motifs for these factors as putative Scro binding sites. Given the small size of the u8772 220 bp enhancer we identified just the binding site predicted for Vnd within this enhancer as the probable Scro binding site to avoid mutating a significant fraction of its sequence length. We then designed strategies to either delete or mutate the identified binding sites for each TF based on their proximity; when two sites for any two or more TFs overlapped, we used mutations that would not disrupt binding of the other factor. We also tried to keep the lengths of the mutated enhancer variants as close as possible to the wild-type enhancers. We then checked our mutant sequences through the FIMO and DNA Pattern Find to select mutation sequence designs that minimized disruptions of other predicted binding sites. *Table 2* lists the original TF binding sites identified and the sequences to which they were mutated. Custom-designed gene blocks (*Supplementary file 4*) containing the mutated enhancer sequences were cloned upstream of the GFP reporter in pJR12 vector as before, and transgenic flies expressing these reporter constructs were then made using the same landing site in the genome as the wild-type reporters. We next compared the expression patterns and intensities of GFP reporters with mutated TF binding sites against GFP reporter expression driven by the 'wild type' version of that same enhancer (*Figure 7A–A''''*, *Figure 8A–A''''*). To compare GFP reporter intensities between wild-type and mutated variants of an enhancer, identical microscope illumination, and imaging settings were used for all variants of the same enhancer.

For the d5778 850 bp reporter, combined deletion and mutation of three predicted Ey binding sites abolished the reporter expression, while deletion of seven possible Scro binding sites reduced the reporter expression greatly (*Figure 8B–C'''' and E*). Deletion of the putative Slp1 binding site did not significantly change the GFP reporter expression (*Figure 8D–D''''*).

For the u8772 220 bp enhancer, individual deletion of one Ey binding site (220-Ey-Site1) or mutation of another Ey binding site (220-Ey-Site2) nearly eliminated GFP expression (*Figure 7B–C'''' and H*). There are four predicted Su(H) binding sites, and among them 220-SuH-Site1 and 4 are perfect matches to the consensus, but only site1 was predicted by FIMO. Site 2 and 3 each has two mismatches to the consensus on non-critical bases. Mutation of all four predicted Su(H) sites in the u8772 220 bp enhancer led to a complete loss of GFP expression from the reporter (*Figure 7D–D'''' and H*). We also generated another reporter mutating only the two Su(H) sites that were perfect matches to the consensus, and this caused a dramatic reduction of the reporter expression (*Figure 7E–E'''' and H*). Mutation of the Scro/Vnd site in the u8772 220 bp enhancer greatly reduced the level of GFP expression (*Figure 7F–F'''' and H*), Mutation of Slp1 binding sites significantly reduced the

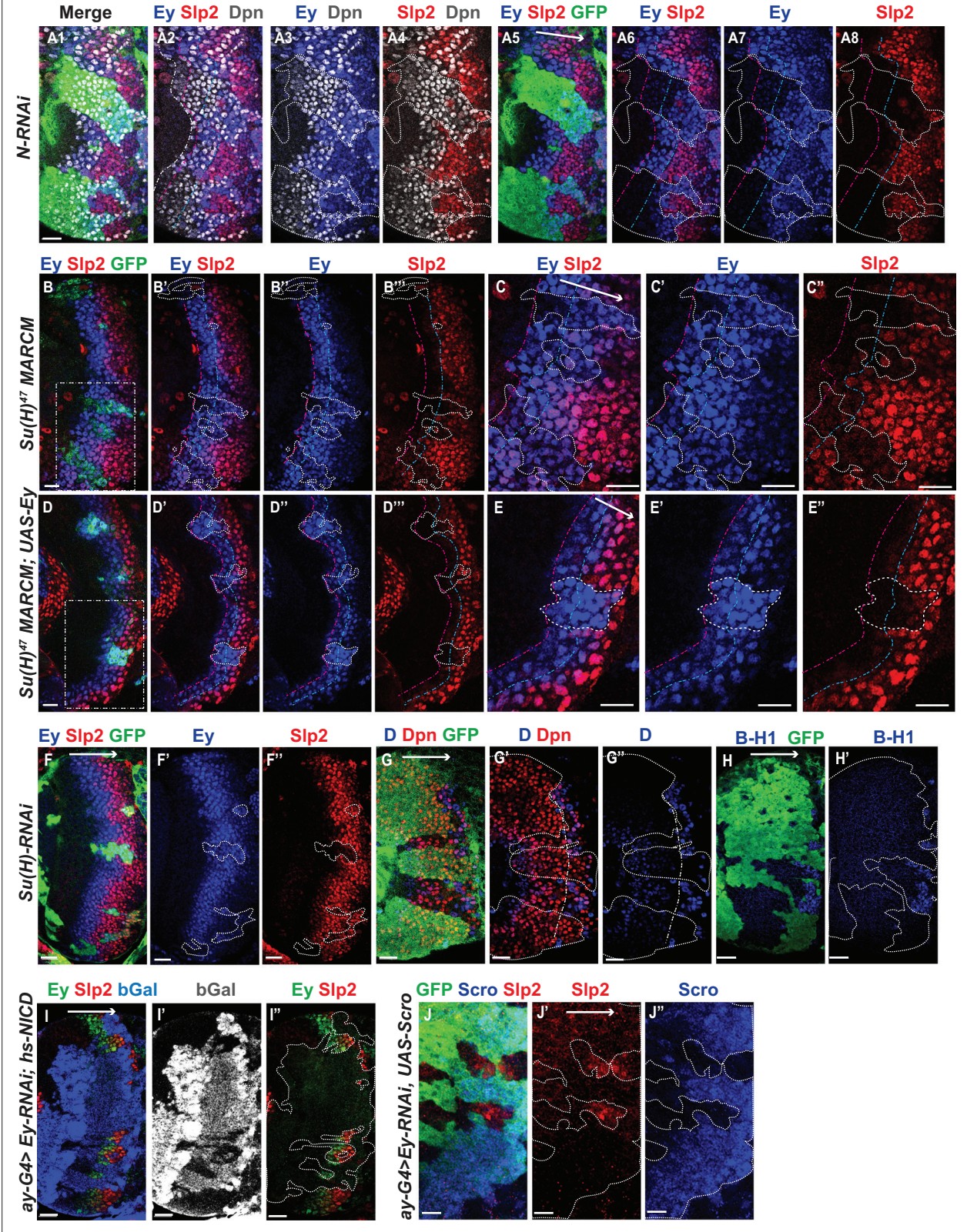

**Figure 5.** Genetic interactions between Notch, Ey and Scro in the regulation of Slp expression. In all panels, dotted lines outline clone margins. Dash-dotted lines mark the expected transition front for Dpn or TTFs (white for Dpn, pink for Ey, cyan for Slp2, and yellow for D). The curvature of the dash-dotted lines is determined by the curvature of Dpn or the previous TTF. (**A1–A8**) In N-RNAi clones marked by GFP in green, Ey is turned on in the 5th-8th NB (6.50±1.05, n=6), and the transition to Slp stage occurs in the 6th –9th Ey +NB (7.67±1.21, n=6). In control regions (not marked by GFP),

*Figure 5 continued on next page*

*Figure 5 continued*

Ey is turned on in the 2nd –3rd NB (2.50±0.55, n=6), and the transition to Slp stage occurs on the 3rd –4th Ey +NB (3.83±0.41, n=6). The delay of Ey expression and the further delay of Slp transition are significant by t-test (p=4.8 × 10$^{-5}$ and p=0.0003, respectively). (**B-B'''**) In *Su(H)* mutant clones marked by GFP, Ey expression is delayed and the transition to Slp stage is further delayed: Slp2 is still barely detectable or very weak at the 6th –8th Ey +NBs (6.80±0.84, n=5). In control regions, the transition to Slp2 stage occurs in the 4th Ey +NBs (n=5). The further delay of Ey to Slp transition is significant by t-test (p=0.002). (**C-C"**) Magnified view of the rectangle area containing three clones shown in B. (**D-D'''**) In *Su(H)* mutant clones over-expressing Ey, the transition to Slp2 stage is still delayed to the 6th-7th Ey +NBs (6.40±0.55, n=5). In control regions, the transition occurs in the 3rd-4th Ey +NBs (3.80±0.45, n=5), The further delay of Ey to Slp transition is significant by t-test (p=4.50 × 10$^{-5}$). (**E-E"**) Magnified view of the rectangle area containing one clone shown in D. (**F-H'**) The expression of Ey and Slp2 (**F-F"**), Dichaete and Dpn (**G-G"**), and BarH-1 (**H,H'**) in *Su(H)*-RNAi clones marked by GFP. (**I-I"**) Supplying active NICD using *hsFlp; dpn >FRT-STOP-FRT-NICD* (heat shocked for 12 min 3 days before dissection) is not sufficient to activate Slp2 expression in Ey-RNAi clones marked by bGal. (**J-J"**) Supplying Scro is not sufficient to activate Slp2 expression in Ey-RNAi clones marked by GFP. Scale bars 20 µm.

The online version of this article includes the following figure supplement(s) for figure 5:

**Figure supplement 1.** Notch signaling is also involved in early temporal transitions.

level of GFP expression, but preserved the pattern of GFP expression as the wild type GFP reporter (*Figure 7G–G''' and H*).

Six potential Su(H) binding sites were identified in the d5778 850 bp enhancer (sites 850-SuH-Site 1 through 6 in *Table 2*). The DNA Pattern Finder tool predicted three Su(H) binding sites that were perfect matches to consensus Su(H) binding sites (*Figure 3*). These were 850-SuH-Site2, 3 and 6. Among them, Site 3 was also predicted by the FIMO-based analysis. 850-SuH-Site1 has one mismatch, while sites 4 and 5 have two mismatches to the Su(H) binding consensus on non-critical bases. We created two different Su(H) site variants. The 850-SuH-1m2m3m6m construct mutated sites 1,2,3 and 6, and the 850-SuH-1–6 m construct had all potential Su(H) binding sites mutated. For both variants, GFP expression is greatly intensified in glial cells, making it difficult to assess the neuroblast expression (*Figure 9—figure supplement 1*). Therefore, we used *repoGal4* to knock down GFP expression in glial cells using *UAS-GFP-RNAi*, and found that the neuroblast GFP expression was actually increased compared to the wild-type reporter for both variants (*Figure 9B–C''' and D*). These data suggest that Notch signaling does not activate this enhancer directly. This is likely the reason why the d5778 850 bp enhancer drives a delayed expression pattern than the u8772 220 bp enhancer and the endogenous Slp.

Taken together, our data suggest that Notch signaling may activate *slp1/2* expression by acting on the u8772 220 bp enhancer alone. Although loss of all Su(H) binding sites in the 220 bp enhancer caused a complete loss of reporter expression, endogenous Slp expression is only delayed when Notch or Su(H) is knocked down. This can be explained by our finding that the d5778 850 bp enhancer directs a delayed expression of Slp and this expression is not dependent on Notch signaling.

## DamID-sequencing confirms binding of Ey and Su(H) to identified enhancers of Slp1/2

DamID-sequencing can be used to profile genome-wide binding patterns for transcription factors and chromatin proteins (*Steensel and Henikoff, 2000*; *Southall et al., 2013*). We used a *SoxN*-Gal4 expressed in all medulla neuroblasts (*Zhu et al., 2022*) to drive the expression of DamSu(H) or DamEy fusion proteins under *UAS* control and inferred the genome-wide binding patterns of Ey and Su(H) through DamID-sequencing. For each transcription factor two biological replicates were sequenced. Reproducibility of peaks of transcription factor binding was determined using the Irreproducibility Discovery Rate (I.D.R.) program to comply with recommended standards of reproducibility specified for genomic data consortia such as ENCODE (*Landt et al., 2012*) and all further analyses were done based on I.D.R. reproducible data. Reproducible peaks of Ey and Su(H) binding were observed in both replicates in the genomic region corresponding to the identified enhancers of Slp1/2 transcription, although reproducible Su(H) peaks were observed only in a smaller region toward the 3' end of the d5778 enhancer, in contrast to the u8772 220 bp enhancer where Su(H) peaks are distributed prominently throughout the enhancer body (*Figure 10*). We also observed peaks of Ey binding in the genomic region of *scro* gene (*Figure 10—figure supplement 1A*), consistent with the observation that Scro expression is initiated in medulla neuroblasts at about the same time as Slp1/2.

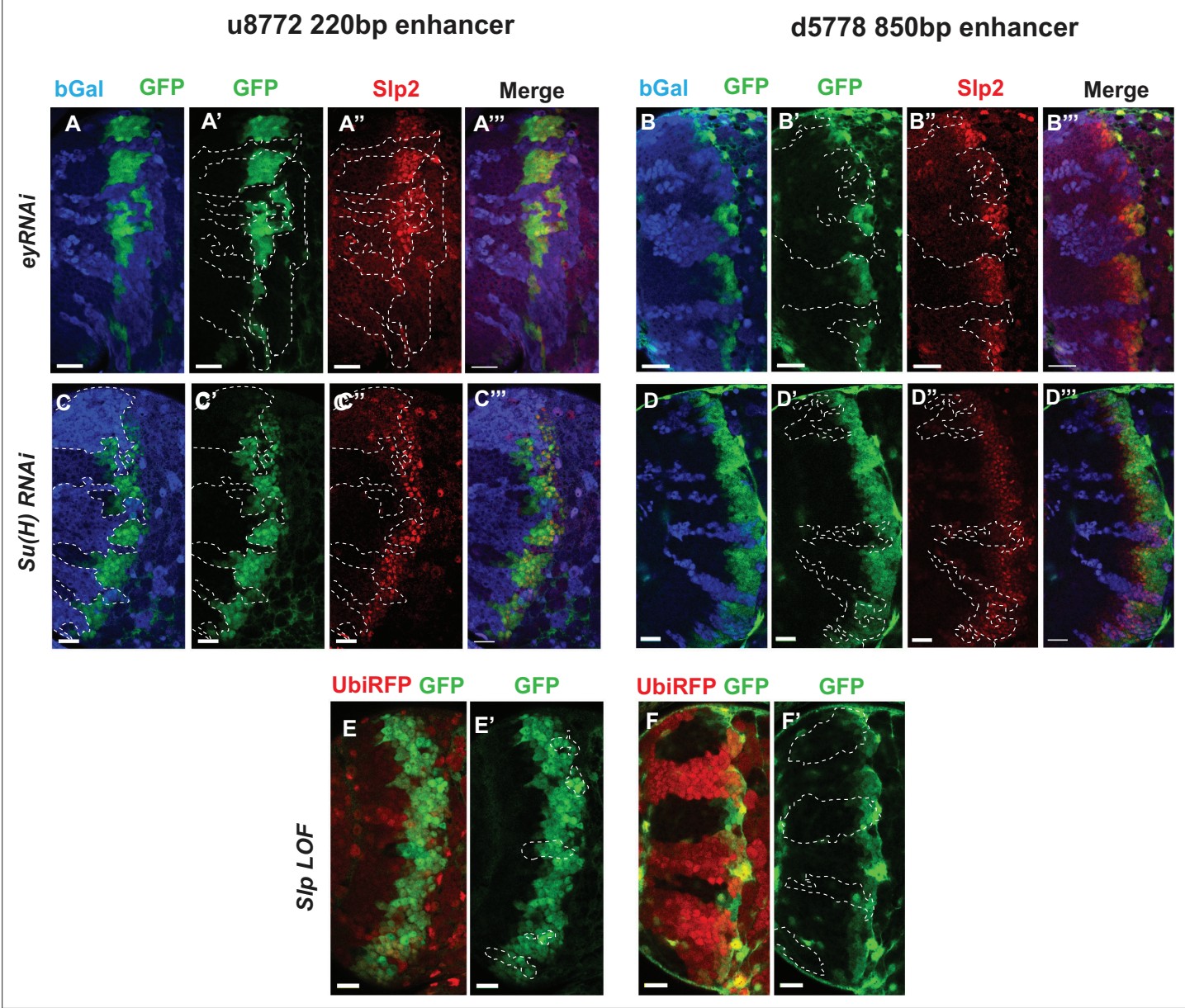

**Figure 6.** Eyeless and the Notch pathway act through the 220 bp and 850 bp enhancers to regulate expression of *slp* genes. (Panels A-A''' through D-D''') RNAi clones in which Ey or Su(H) is knocked down are marked by *Beta-galactosidase* (*bGal*) in blue, while regions unmarked by *bGal* are populated by wild type neuroblasts. (**A-A'''**) The expression of a GFP reporter driven by the 220 bp enhancer and endogenous Slp2 in *ey RNAi* clones and wild type neuroblasts. Expressions of both the GFP reporter and endogenous Slp2 are similarly abolished in *ey* knockdown clones (representative data shown, n=5 brains). (**B-B'''**) The expression of a GFP reporter driven by the d5778 850 bp enhancer and endogenous Slp2 in *ey RNAi* knockdown clones and in wild type neuroblasts. Knockdown of Ey leads to loss of both the GFP reporter expression and the endogenous Slp2 protein (representative data shown, n=15 brains). (**C-C'''**) The expression of the GFP reporter driven by 220 bp enhancer and endogenous Slp2 in *Su(H) RNAi* clones and in wild type neuroblasts (representative data shown, n=11 brains). (**D-D'''**) The expression of the GFP reporter driven by 850 bp enhancer and of endogenous Slp2 in *Su(H) RNAi* clones and in wild-type neuroblasts (representative data shown, n=14 brains). In case of both enhancers the expression of the GFP reporter is lost or delayed. (**E, E'**) The expression of the 220 bp GFP reporter in *slp1* and *slp2* loss-of-function mitotic clones ('dark' regions) and in wild-type neuroblasts (marked by RFP) (representative data shown, n=24 brains). (**F,F'**) Expression of the 850 bp enhancer GFP reporter in *slp* LOF clones and wild-type neuroblasts similar to (**E,E'**) (representative data shown, n=16 brains). Slp1 binding sites were identified in both enhancer elements which indicate that Slp1 may auto-regulate its expression. However, GFP reporter expression driven by these two enhancers is not significantly affected within *slp* LOF neuroblasts, suggesting that Slp1/2 are not required to initiate their own expression in neuroblasts. Representative images are provided. Scale bar 20 μm.

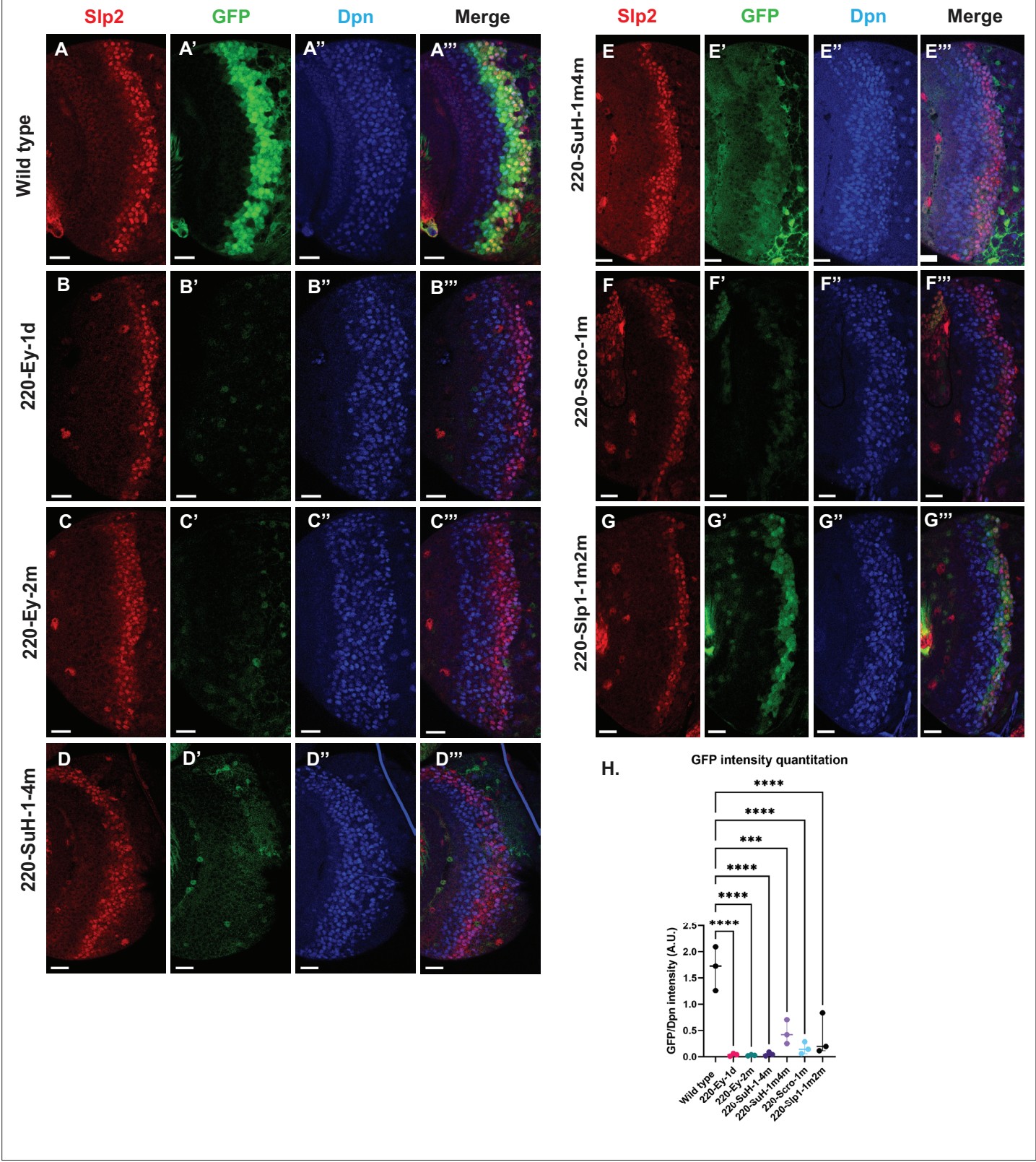

**Figure 7.** Mutations of predicted binding sites for transcription factors Ey, Su(H) and Scro in the u8772 220 bp enhancer establish their requirement for activating *slp1/slp2* mRNA transcription in medulla neuroblasts. Expression of GFP reporter driven by the wild-type 220 bp enhancer (**A-A'''**) was compared against GFP reporters driven by mutated versions of the 220 bp enhancer in which predicted binding sites for specific transcription factors had been mutated: (**B-B'''**) with a predicted site for Ey binding (220-Ey-site1 in *Table 2*) deleted, (**C-C'''**) with a different predicted site for Ey binding

*Figure 7 continued on next page*

Figure 7 continued

(site 220-Ey-Site2 in **Table 2**) mutated, (**D-D'''**) with four predicted Su(H) binding sites mutated (sites 220-SuH-Site1 to 4), (**E-E'''**) with two Su(H) binding sites mutated (sites 220-SuH-Site1 and 220-SuH-Site4 in **Table 2**), (**F-F'''**) with a predicted site for Scro binding mutated (220-Scro-Site1 in **Table 2**), (**G-G'''**) with two predicted Slp1 binding sites mutated (sites 220-Slp1-Site1 and 220-Slp1-Site2). Loss of binding sites for Ey, Su(H) and Scro in the 220 bp enhancer led to a loss or reduction of GFP reporter expression. Scale bar: 20 μm. (**H**) Intensity comparisons of mutated and wild type GFP reporters of the u8772 220 bp enhancer. The decrease in intensity of GFP expression for Ey, Scro, Su(H) and Slp1 binding site mutated reporters relative to wild type are all statistically significant. Ordinary one way ANOVA was performed to determine statistical significance between the wild type and the mutated enhancer variants. The differences in GFP expressions were statistically significant for all enhancer variants relative to the wild-type enhancer. Adjusted p values from Dunnett's multiple comparisons test between wild type and mutants are as follows: for both Ey site mutants, the 220-Scro-1m mutant, and the 220-Slp-1m2m, p values were <0.0001. For the 220-SuH-1–4 m, p value was also <0.0001, but for the 220-SuH-1m4m the p value was 0.0001. Three different sets of observations were used for analyses (n=3). The graph shows the distribution of individual observations about the median and within 95% confidence intervals.

The online version of this article includes the following source data and figure supplement(s) for figure 7:

**Source data 1.** Quantification and comparison of reporter expression level driven by wild type and mutant forms of u8772 220bp enhancer.

**Figure supplement 1.** Deletion of one Su(H) site identified by FIMO in the u8772 220 bp enhancer GFP reporter does not abolish GFP expression.

Additionally, our experiments also revealed that a list of genes were bound reproducibly both by Ey and by Su(H) (**Supplementary file 5**). Using DAVID, we conducted a functional annotation clustering analysis of this gene list. Genes related to the Notch pathway and participating in the cell division cycle were strikingly enriched within this list of co-bound genes (**Supplementary file 6**), thus raising the possibility that in medulla neuroblasts, TTFs such as Ey can modulate the responsiveness of Notch target genes in the medulla to mitogenic Notch signaling as has been seen in Type II neuroblasts (**Farnsworth et al., 2015**) (note that Ey can function as both transcriptional activator and repressor). Another set of genes were bound reproducibly only by Ey (**Supplementary file 5**). For example, only Ey I.D.R peaks were observed in the genomic region of the *hmx* gene (**Figure 10—figure supplement 1B**), encoding a transcription factor expressed in a type of Notch-off neurons born from the Ey stage (**Konstantinides et al., 2022**; **Kurmangaliyev et al., 2020**). DAVID analysis of genes bound uniquely by Ey showed an enriched representation of genes involved in axon guidance and maintenance, and interestingly, of genes that participate within or interact with the Hippo pathway and Ras pathway (**Supplementary file 6**). It remains to be studied whether Ey regulates the expression of these genes. Finally, a third set of genes were bound reproducibly only by Su(H) (**Supplementary file 5**). For example, an I.D.R peak only for Su(H) was found in the *dll* genomic region (**Figure 10—figure supplement 1C**). Since Notch-on neurons born from the Tll stage neuroblasts express Dll (**Konstantinides et al., 2022**), it will be interesting to test whether Notch signaling activates Dll expression directly. DAVID analysis of the gene list bound uniquely by Su(H) shows a representation of genes participating in the EGFR/MAPK pathway such as p38 MAPK and in BMP signaling (**Supplementary file 6**). These preliminary data indicate that the Notch pathway may interact with other signaling pathways such as MAPK and BMP in medulla neuroblasts and present a promising opportunity for future studies.

Taken together our DamID data support evidence from our genetics studies and suggests that Ey and Su(H)/N pathway activate Slp1/2 transcription by direct binding to Slp1/2 enhancers.

## Notch signaling is dependent on cell-cycle progression

Although it has been well-established that GMCs generated by the asymmetric division of neuroblasts signal to their sister neuroblasts to provide the Notch signaling (**Knoblich et al., 1995**; **Rhyu et al., 1994**), it has not been demonstrated whether Notch signaling is lost in cell-cycle arrested medulla neuroblasts since they fail to generate GMCs. We used a regional Gal4, *vsx-Gal4* that is expressed in the center domain of the medulla crescent starting in the neuroepithelium (**Erclik et al., 2017**), to drive *UAS-DCR2* and a RNAi against Proliferating Cell Nuclear Antigen (PCNA), which is essential for DNA replication and S phase progression (**Moldovan et al., 2006**). Blocking cell cycle progression caused premature transformation of neuroepithelial cells to neuroblasts, as has been previously reported (**Zhou and Luo, 2013**). These neuroblasts express the normal neuroblast marker Deadpan (Dpn) (**Figure 11A–A'''**), but produce few progenies, and some neuroblasts are present in the deep layers which would normally be occupied by neuronal progenies (**Figure 11D**). In the region where PCNA is knocked down, a large fraction of neuroblasts don't express E(spl)mγGFP, a Notch signaling

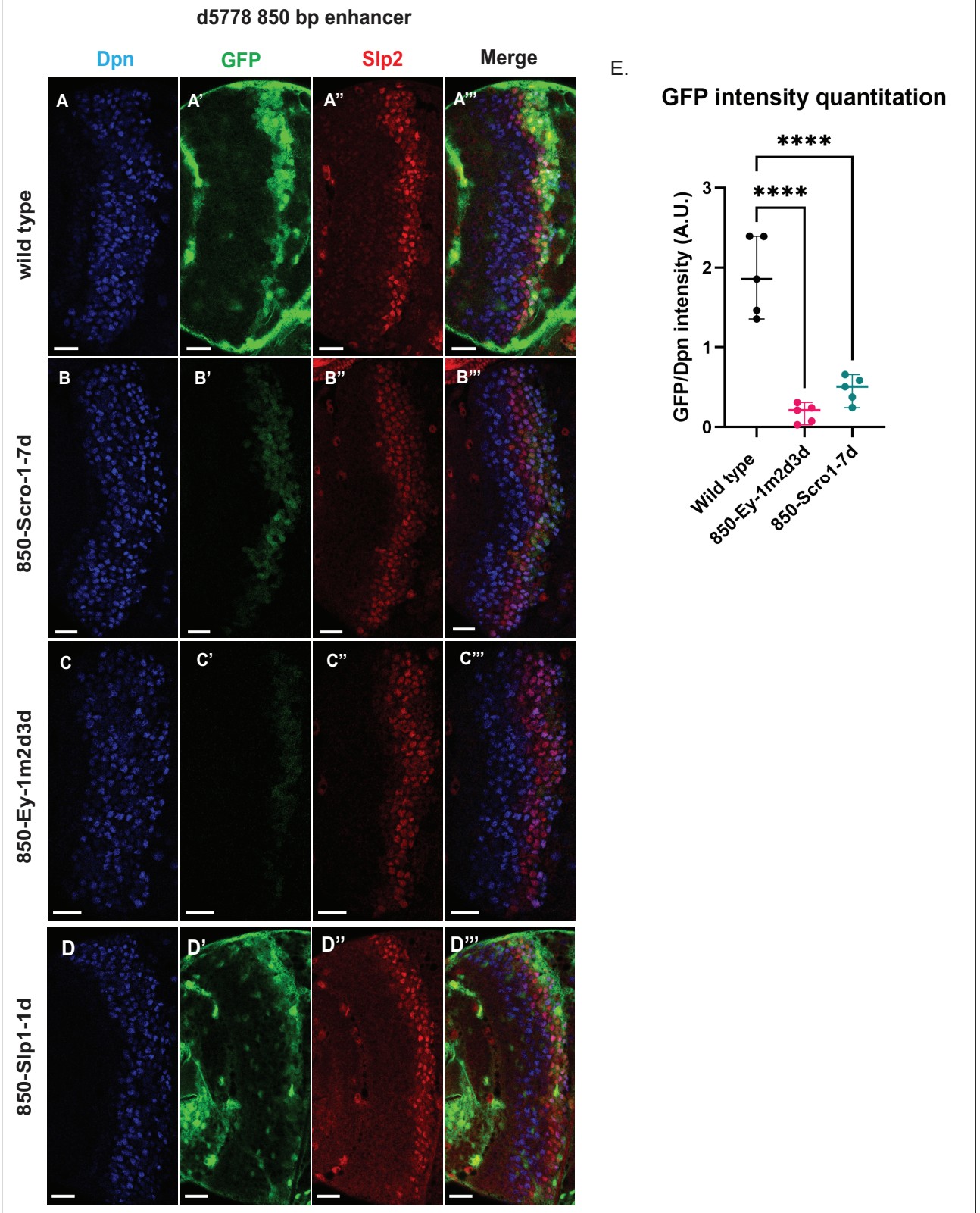

**Figure 8.** Mutations of putative binding sites for Ey and Scro in the d5778 850 bp enhancer show their requirement for activating Slp1/Slp2 gene expressions in medulla neuroblasts. A parallel comparison was made for GFP reporter expression driven by the wild-type 850 bp enhancer (**A-A'''**) to the GFP reporter expression driven by versions of the 850 bp enhancer with binding sites for specific transcription factors Scro, Ey, and Slp1 mutated (**B-B'''**, **C-C'''**, and **D-D'''** respectively). For the Ey site mutant 850-Ey-1m2d3d shown in (**B-B''''**) three potential binding sites for Ey were mutated

*Figure 8 continued on next page*

*Figure 8 continued*

(850-Ey-Site1 in *Table 2* was mutated as indicated, 850-Ey-Site2 and 850-Ey-Site3 were deleted). For the Scro sites mutant 850-Scro-1-7d, seven of the potential Scro binding sites (850-Scro-Site1 through 850-Scro-Site7 in *Table 2*) were deleted. Loss of binding sites for Ey obliterated GFP reporter expression in neuroblasts, while loss of potential Scro binding sites resulted in a noticeable reduction of GFP expression from the d5778 850 bp reporter. (**E**) Quantitation of d5778 850 bp reporter intensities of the wild type and the factor binding site mutated variants shows that the reduction of GFP intensity compared to wild type for Ey binding site and Scro binding site mutated reporters is statistically significant. Scale bars: 20 µm. Ordinary one way ANOVA was performed to compare differences in GFP expression between the wild type d5778 850 bp enhancer and the transcription factor binding site mutants. Adjusted p values from Dunnett's multiple comparisons test between the wild type d5778 850 bp enhancer and both the Ey and Scro sites mutants were less than 0.0001. Five different sets of observations were used for analyses (n=5). The graph shows the distribution of individual observations about the median and within 95% confidence intervals.

The online version of this article includes the following source data for figure 8:

**Source data 1.** Quantification and comparison of reporter expression level driven by wild type and mutant forms of d5778 850bp enhancer.

reporter (*Campos-Ortega, 1988*), while all neuroblasts in the control regions express E(spl)mγGFP (*Figure 11A–A'''*). Thus, Notch signaling is largely lost in cell-cycle arrested neuroblasts.

## The role of cell cycle progression in the medulla TTF cascade

Next, we tested whether blocking cell-cycle progression also affected the medulla TTF cascade progression. In *vsx-Gal4 >PCNA* RNAi brains, neuroblasts in the affected region have reduced expression of Ey, and loss of Slp1/2, but they keep expressing SoxN (*Figure 11B–D''*). This suggests that an earlier temporal transition step is blocked which is required for Ey expression, and this prevents us from examining whether the Ey to Slp transition also requires cell cycle progression. To circumvent this problem, we used the *ay-Gal4* to drive *UAS-DCR2* and *PCNA-RNAi*. By adjusting heat shock timing to induce clones, we were able to obtain a condition where Ey expression is minimally affected, and at this condition we observed that Slp2 expression is still severely delayed (*Figure 11E–F''*). In addition to the timing of clone induction, the *ay-Gal4* is also likely to be weaker than *Vsx-Gal4* at inducing RNAi, and the cell cycle is likely only slowed down rather than arrested when using *ay-Gal4*, because there are still some progenies produced in such clones. Furthermore, Slp1/2 expression is also significantly delayed when we knock down String /Cdc25 required for the G2 to M phase transition (*Edgar and O'Farrell, 1989*), or overexpress Dacapo (Dap, a Cyclin-dependent kinase inhibitor in the CIP/KIP family) (*Lane et al., 1996*) using *ay-Gal4*, while Ey is slightly delayed (*Figure 11—figure supplement 1A-D'''*). Thus, cell cycle progression is also required for the precise timing of Ey to Slp1/2 transition.

## Notch signaling can rescue Slp expression delay caused by cell cycle defects

Finally, we tested if supplying Notch signaling to cell-cycle arrested or delayed neuroblasts can rescue the timing of the Ey to Slp transition. We supplied active Notch in neuroblasts using the *dpn >FRT-stop-FRT3-FRT-NICD* transgene, while simultaneously inducing knockdown of PCNA using *ay-Gal4* under the same heat shock condition that leaves Ey expression largely unaffected. Supplying NICD was sufficient to rescue Slp2 expression in *PCNA-RNAi* clones (*Figure 11G–H''*). This suggests that for the Ey to Slp transition, the presence of active Notch signaling can substitute for the requirement of the cell-cycle progression. In wild type condition, supplying active NICD did not significantly affect the timing of the Ey to Slp transition, which is still around 3rd-4th Ey +neuroblast (*Figure 11I–J'''*), possibly because that wild type cycling neuroblasts already have active Notch signaling.

## Discussion

### Evidence of direct transcriptional activation in the medulla TTF cascade

*Drosophila* neuroblasts are temporally patterned by sequentially expressed TTFs. Although the expression pattern and mutant phenotypes suggest that TTFs form a transcriptional cascade, direct transcriptional regulation between TTFs has not been demonstrated in most cases. This work has characterized two enhancers of the *slp* genes that enable the expression of Slp1 and Slp2 in medulla neuroblasts. The u8772 220 bp enhancer is activated at an earlier stage relative to the d5778 850 bp enhancer. In these two enhancers, we identified sites for the previous TTF -Ey and Scro-a TTF expressed at around the same time as Slp1. Deleting either enhancer alone did not eliminate

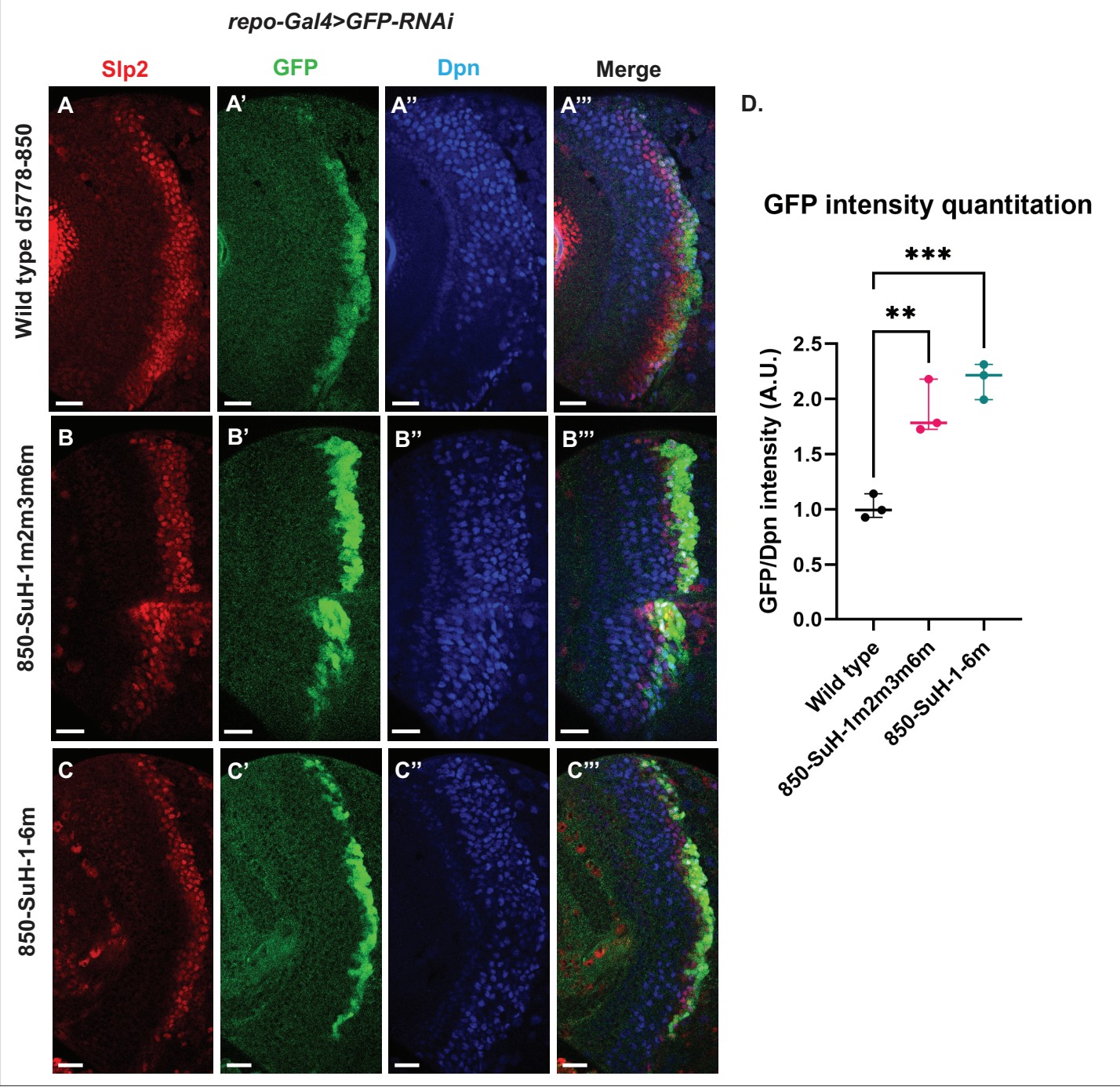

**Figure 9.** Mutation of potential Su(H) binding sites in the d5778 850 bp enhancer does not decrease GFP reporter expression in neuroblasts. To measure GFP intensities in medulla neuroblasts unimpeded by GFP fluorescence from glial cells, *UAS-GFP-RNAi; d5778 850 bp >GFP* wild type reporter or *UAS-GFP-RNAi; d5778 850 Su(H) binding site mutant GFP* reporters were mated with a *UAS-Dcr2; Repo-Gal4* expressing line, and phenotypes in the resulting progeny were observed. Mutation of four Su(H) binding sites in the 850-SuH-1m2m3m6m mutant (**B-B'''**) and all six possible Su(H) binding sites in the 850-SuH-1–6 m mutant (**C-C'''**) showed no loss of GFP expression relative to the wild type d5778 850 bp GFP reporter (**A-A'''**). Instead, GFP expression was upregulated in Su(H) site mutants of the d5778 850 bp reporter. This indicates that Su(H) does not activate *slp1/slp2* expression through the d5778 850 bp enhancer directly. Scale bars: 20 μm. (**D**) Three sets of observations were used for statistical analyses (n=3). Ordinary one-way ANOVA was used to compare relative GFP intensities between Su(H) site mutants and the wild type d5778 850 bp enhancer. Adjusted p values by Dunnett's multiple comparisons test between the wild type d5778 850 bp reporter and the Su(H) site mutant reporters are as follows: for the 850-SuH-1m2m3m6m reporter p=0.0019, and the 850-SuH-1–6 m reporter is p=0.0005. Difference of means of each sample relative to wild type are statistically significant

*Figure 9 continued on next page*

*Figure 9 continued*

as p<0.005, below the significance threshold of p<0.05. The graph shows the distribution of individual observations about the median and within 95% confidence intervals.

The online version of this article includes the following source data and figure supplement(s) for figure 9:

**Source data 1.** Quantification and comparison of reporter expression level driven by wild type and Su(H)-site-mutated forms of d5778 850bp enhancer after glial expression is removed by RNAi.

**Figure supplement 1.** Mutation of Su(H) binding sites in the d5778 850 bp enhancer causes intensified GFP expression in glia.

the expression of endogenous Slp1 and Slp2, suggesting that they act partially redundantly with one another. Deletion of both enhancers completely eliminates Slp1 and Slp2 expression in medulla neuroblasts but does not affect their expression in lamina neurons or glia, confirming the specificity and necessity of these two enhancers. Using GFP reporter assays, we have shown that mutation of Ey binding sites in these enhancers abolishes reporter expression similar to genetic experiments where we observe a loss of GFP reporter within *ey* RNAi clones. Our results are also consistent with previous studies that showed a complete loss of endogenous Slp1/2 expression in *UAS-ey-RNAi* expressing neuroblasts (*Li et al., 2013*). We also confirmed the in vivo binding of Ey to the identified enhancers of Slp by Dam-ID sequencing. The expression of the TTF Scro is initiated simultaneously as Slp1/2,

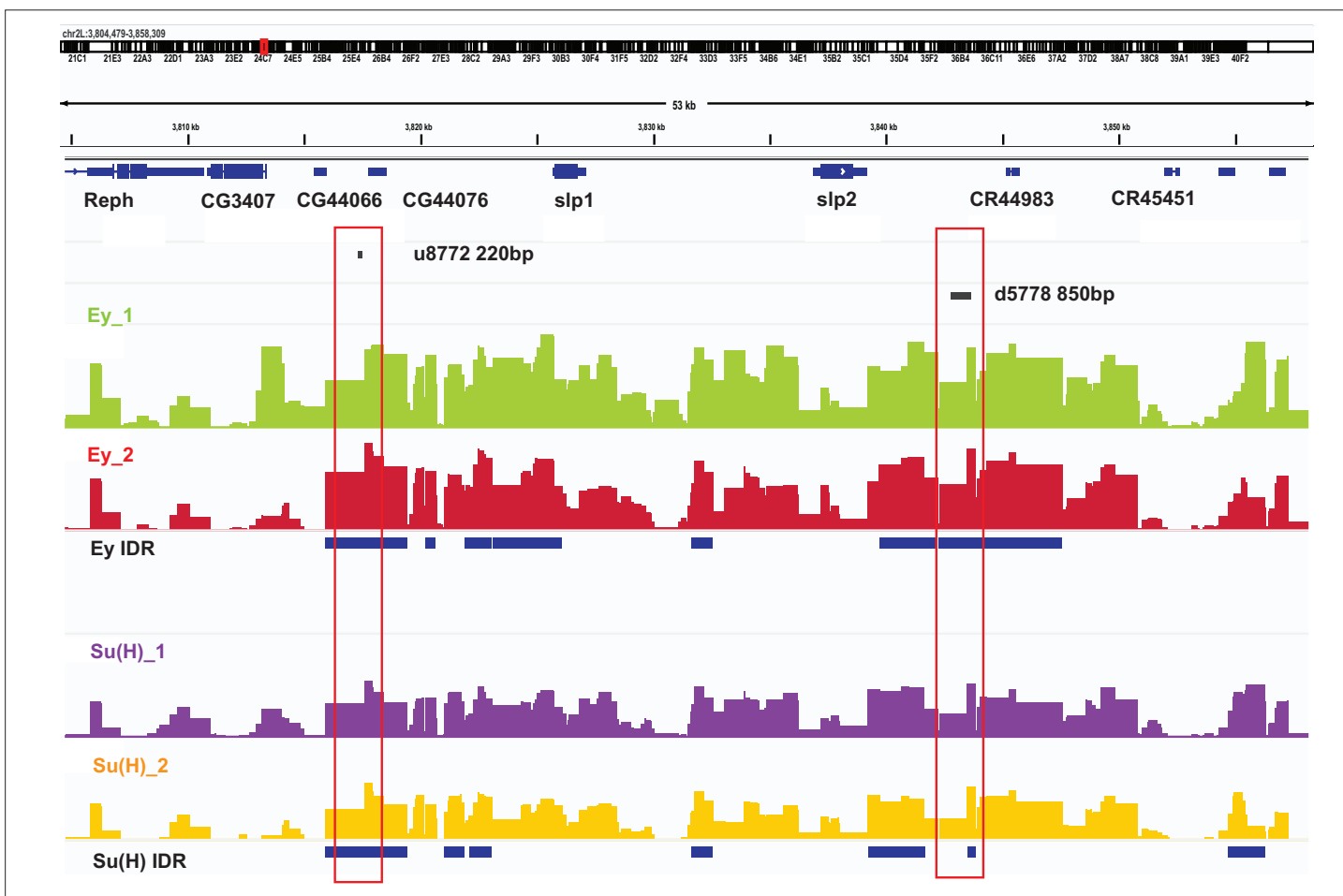

**Figure 10.** DamID-seq demonstrates reproducible binding of Ey and Su(H) to the *slp1/2* enhancers in vivo. Prominent reproducible peaks of Ey and Su(H) binding are seen in neighborhood of the *slp1* and *slp2* gene loci including at the genomic locations of identified 220 bp and 850 bp enhancers in both replicates of Ey-Dam and Su(H)-Dam experiments. Peaks at the u8772 220 bp and the d5778 850 bp enhancers have passed IDR <0.05 cut-off supporting their reproducibility.

The online version of this article includes the following figure supplement(s) for figure 10:

**Figure supplement 1.** Examples of DamID-seq profiles on selected genes.

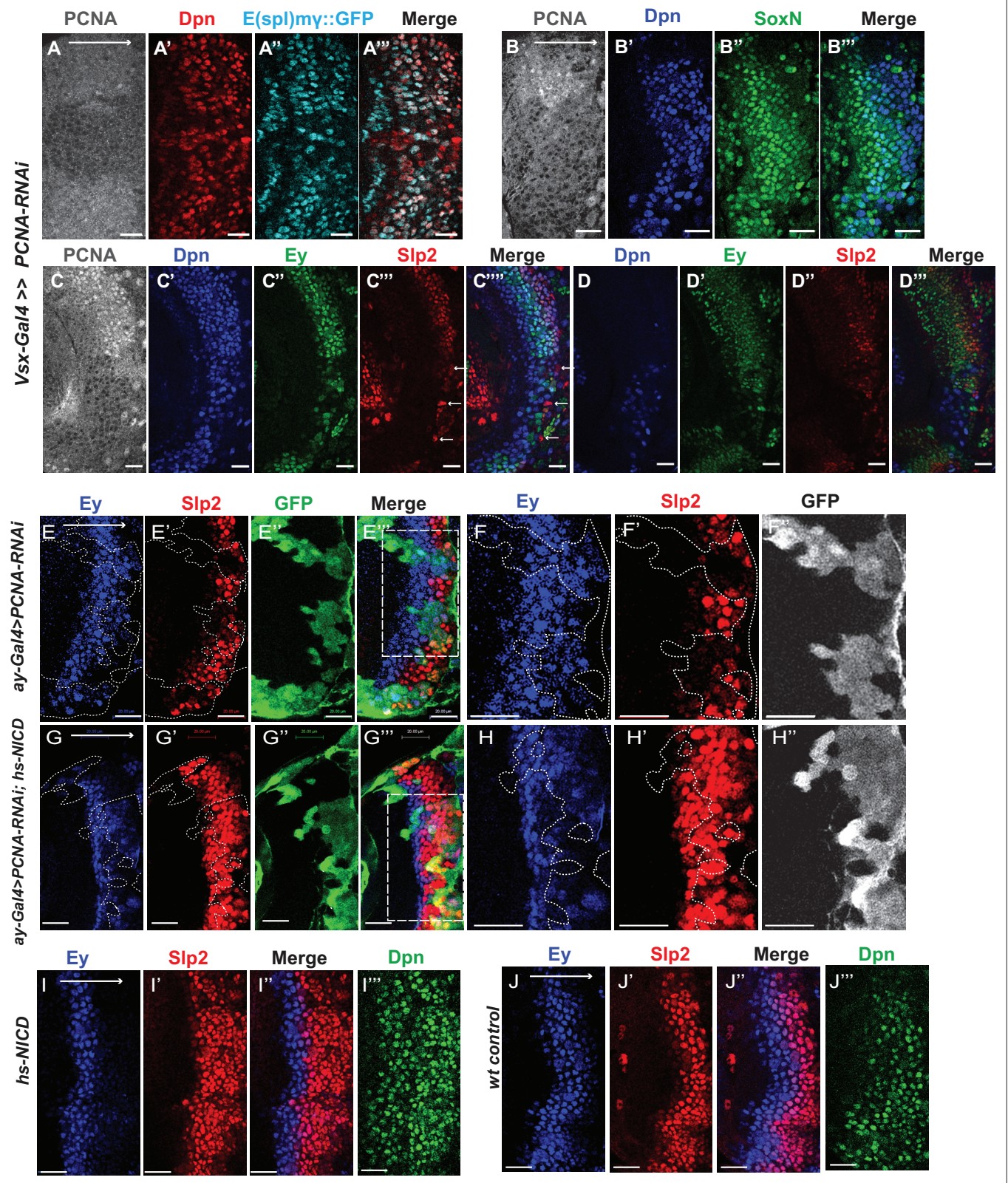

**Figure 11.** Cell cycle progression is required for the precise timing of Slp expression and this is mediated in part through Notch signaling. (**A-A'''**) The expression of PCNA, Dpn (red), and E(spl)myGFP (cyan) in *vsx-Gal4* driving *UAS-DCR2* and *UAS-PCNA-RNAi* (VDRC51253) optic lobes (n=5). The affected region is indicated by loss of PCNA staining. (**B-B''**) The expression of Dpn and SoxN in *vsx-Gal4* driving *UAS-DCR2* and *UAS-PCNA-RNAi* optic lobes (n=5). (**C-D'''**) The expression of Dpn, Ey, and Slp2 in *vsx-Gal4* driving *UAS-DCR2* and *UAS-PCNA-RNAi* optic lobes (n=9) showing the surface

*Figure 11 continued on next page*

*Figure 11 continued*

neuroblast layer view (**C-C''''**) and the deep progeny layer view (**D-D'''**) of the same brain. (**C-C''''**) In affected region marked by loss of PCNA staining, Ey expression is reduced and Slp2 expression is lost in Dpn +neuroblasts. Some glia express Slp2 but not Dpn are indicated by white arrows. (**D-D'''**) Dpn expressing neuroblasts are observed in deep layers in the affected region, but they don't express Ey or Slp2. (**E-E'''**) Larvae of genotype *ywhsFLP; ay-Gal4 UAS-GFP /UAS-PCNA-RNAi; UAS-DCR2/+* were heat shocked at 37°C for 8 min 70 hours before dissection at 3rd instar larval stage. Ey expression (blue) is minimally affected in clones marked by GFP (green), while Slp2 (red) expression is delayed (n=17 clones). (**F-F''**) Magnified view of the boxed region in E'''. (**G-G'''**) Larvae of genotype *ywhsFLP; ay-Gal4 UA-SGFP /UAS-PCNA-RNAi; UAS-DCR2/dpn >FRT-STOP-FRT-NICD* were heat shocked at 37 °C for 8 min 70 hr before dissection at 3rd instar larval stage. Ey expression (blue) is minimally affected in clones marked by GFP (green), while Slp2 (red) expression initiation is rescued (n=10 clones). (**H-H''**) Magnified view of the boxed region in G'''. (**I-J'''**) The expression of Ey, Slp2, and Dpn in *ywhsFLP; dpn >FRT-STOP-FRT-NICD* (**I-I'''**) and *yw* control (**J-J'''**) optic lobes (n=5 each) (heat shocked 37°C for 8 min 70 hr before dissection). Scale bars 20 μm.

The online version of this article includes the following figure supplement(s) for figure 11:

**Figure supplement 1.** Cell cycle progression is required for the temporal cascade progression.

and it has been shown that loss of Scro significantly reduces Slp expression level. Mutation of most probable Scro binding sites on the u8772 220 bp or d5778 850 bp enhancers led to a dramatic reduction of GFP reporter expression. Thus, the combined effect of mutating Scro binding sites on both enhancers recapitulates the observed impact of Scro knock-down on endogenous Slp1/2 expression, which is reduced expression of Slp1/2 in neuroblasts expressing *UAS scro-RNAi* and a consequent loss of neural fates specified by Slp1/2 in their progeny (*Zhu et al., 2022*). It is interesting to note that our observation of multiple enhancers regulating Slp1/2 expression (*Figure 12*) is consistent with regulation of Slp1/2 in other developmental contexts. Previous studies have noted the presence of multiple enhancers of Slp1/2 expression in the vicinity of the *slp1* and *slp2* coding loci. Many of these regulatory DNA segments function as stripe enhancers enabling Slp1/2 to function as pair-rule genes during embryonic segmentation (*Fujioka and Jaynes, 2012*). Although these enhancers share some overlapping functions and domains of activation, a full complement of stripe enhancers is required for maintaining parasegment boundaries and wingless expression (*Cadigan et al., 1994*; *Fujioka and Jaynes, 2012*).

## Notch signaling regulates temporal transitions in medulla neuroblasts

It was previously demonstrated that although Ey is necessary for activating Slp1/2 expression it is not sufficient (*Li et al., 2013*). There is always a time delay after the start of Ey expression to the start of Slp expression to ensure the sufficient duration of the Ey window. How is the timing controlled? From our analyses of the *slp1/2* enhancer sequences we found several binding sites for the CSL transcription factor Su(H), most prominently known as a central component of the ternary Notch transcription complex and the primary DNA binding component. To confirm the involvement of the Notch pathway in regulating Slp1/2, we observed the effects of knocking down key Notch pathway components on endogenous Slp1/2 expression. In all cases we observed a delay in the expression of Ey and a further delay in the transition to the Slp1/2 stage in neuroblasts expressing the RNAi knockdowns. Mutating Su(H) binding sites in the u8772 220bpenhancer led to a loss or reduction of GFP reporter expression in neuroblasts. However, mutating Su(H) binding sites in the d5778 850 bp enhancer did not decrease the reporter expression. These results suggest that Notch signaling directly regulates Slp expression through the u8772 220 bp enhancer, but not the d5778 850 bp enhancer, and this is consistent with the delayed expression driven by the d5778 850 bp enhancer. However, Ey still plays a more critical role in activating Slp1/2 expression than the Notch pathway, since Slp1/2 are still expressed albeit later in the absence of Su(H) and other Notch components, and Notch signaling requires Ey to speed up the Ey to Slp transition. As with Ey, we confirmed Su(H) binding to the u872 220 bp enhancer using DamID-seq. Thus, our work provided strong evidence that N signaling, a general signaling pathway involved in neuroblast development, regulates the timing of activation of a TTF gene directly (*Figure 12*). In addition, our results also raised the interesting hypothesis that Notch signaling might be involved in facilitating all temporal transitions, because the turning on of Opa and Ey is also delayed, and we observed a further and further delay in turning on of later TTFs. Whether Notch signaling regulates other TTF expression directly or indirectly still awaits further investigation.

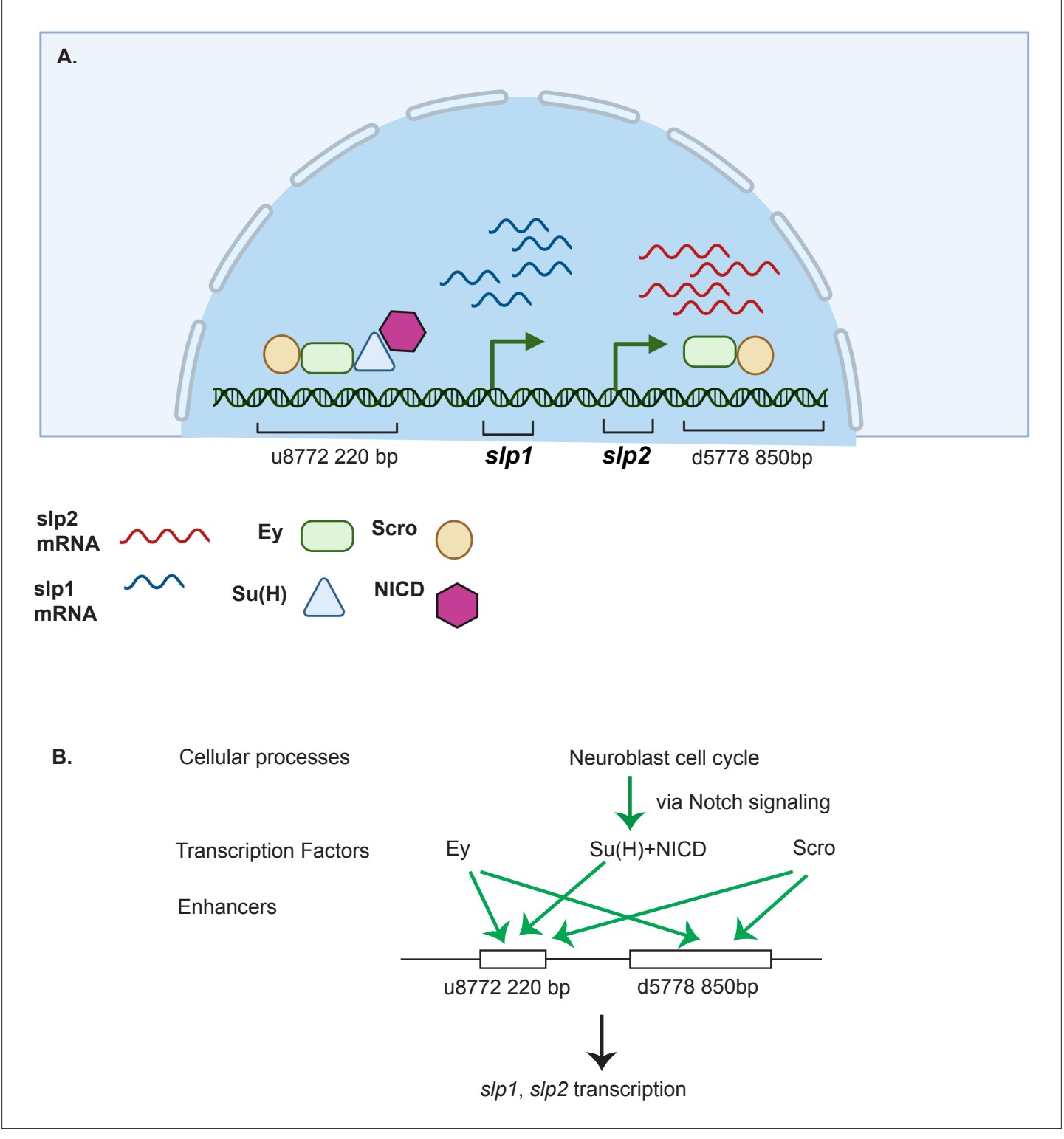

**Figure 12.** A coordinated action of temporal patterning factors Eyeless (Ey) and Scarecrow (Scro) and the Notch signaling pathway regulates the expressions of Slp1 and Slp2 in medulla neuroblasts. (**A**) A schematic summarizing the main findings of this study. In medulla neuroblasts, transcription of the *slp1* and *slp2* genes is regulated by other temporal patterning factors Ey and Scro and by the Notch pathway. Ey and Scro both activate *slp1/2* transcription by binding to the identified enhancers u8772 220 bp and d5778 850 bp. Su(H) activates slp1/2 transcription by binding to the u8772 220 bp enhancer. (**B**) The cell cycle in neuroblasts is required for continued activation of Notch signaling in medulla neuroblasts. We show that the cell cycle influences expression of *slp1* and *slp2* genes via its effects on the Notch signaling pathway; Slp1/2 expression is lost in cell cycle arrested neuroblasts

*Figure 12 continued on next page*

*Figure 12 continued*

due to a failure of Notch signaling. Restoration of Notch signaling rescues Slp1/2 expression in cell cycle arrested neuroblasts. Cellular processes like the neuroblast cell cycle cooperate with temporal patterning and Notch signaling to attain precise developmental outcomes. Graphic A created using BioRender.

## Cooperative regulation of target genes including *slp* by the TTF Ey and Notch signaling

What might explain the delay in Slp1/2 expression in the absence of Notch signaling? Recent developments in single-molecule Fluorescence In Situ Hybridization (smFISH) technology and live imaging techniques using the MS2-MCP system have enabled studying the transcription process in molecular detail. Imaging transcription driven by Notch responsive enhancers in native contexts has shown this process to be inherently 'bursty', i.e., episodes of transcription (enhancer 'On' state) are punctuated with gaps in activity (enhancer in 'Off' state) (*Falo-Sanjuan et al., 2019*; *Lee et al., 2019*). The dosage of NICD modulates the duration of the 'On' phase in one context studied by live imaging (*Falo-Sanjuan et al., 2019*; *Lee et al., 2019*). Additionally, binding of tissue-specific regional factors to these Notch responsive enhancers may prime these enhancers and help synchronize transcription and sustain a steady transcriptional output upon Notch binding to enhancers; this helps integrate important positional cues and the perception of context (*Falo-Sanjuan et al., 2019*). Applying these insights to our system, we suggest that Ey may act by priming the Notch-responsive enhancer of *slp* providing crucial contextual information, and this is required for Notch to further activate Slp1/2 transcription, and speed up the transition (*Figure 12*).

## The role of cell cycle progression and Notch signaling in the medulla TTF cascade

Notch target genes and Dpn are transcriptional repressors that act partially redundantly to maintain neuroblast identity. In type II NBs, Dpn depends on Notch signaling, and loss of Dpn causes premature differentiation (*Bowman et al., 2008*; *San Juan et al., 2012*). However, in type I NBs, Dpn is not lost when Notch signaling is lost, and Notch signaling seems dispensable for the self-renewing abilities of NBs (*Zacharioudaki et al., 2012*; *Almeida and Bray, 2005*; *Magadi et al., 2020*). In the medulla neuroblasts, we also observed that in *Su(H)* mutant clones, the clone size and neuroblast proliferation are not significantly affected. On the other hand, we observed that N signaling is dependent on cell-cycle progression, and the Notch target gene is lost when cell cycle progression is blocked.

In the medulla, blocking cell cycle progression in neuroepithelial cells prematurely transforms them into neuroblasts, and these neuroblasts seem to be arrested or severely delayed in the TTF cascade. When we arrested or slowed down the cell cycle later in neuroblasts to preserve Ey expression, we observed Slp expression is still delayed. Therefore, cell cycle progression also has a role in the Ey to Slp transition. Further, we showed that supplying Notch signaling is sufficient to rescue the delay in the Ey to Slp transition caused by cell cycle defect. Thus at the Ey to Slp transition, the cell cycle effect is mediated through the direct regulation of Slp transcription by Notch signaling. Taken together, our results suggest that in Ey stage neuroblasts, Ey is required to initiate Slp expression but not sufficient to activate it to a strong level right away, and after each asymmetric division, activation of Notch signaling in the neuroblast enhances Slp expression, until Slp expression reaches a certain level to repress Ey expression and make the transition (*Figure 12*). This can be part of a mechanism to coordinate the TTF temporal transition with the cell cycle progression to generate the appropriate number of neural progenies at a given temporal stage.

## Materials and methods
### Fly stocks and genetics

Flies were reared on yeast food at 25 °C unless otherwise stated.

### Enhancer identification

Flies carrying the *GMR35H02-Gal4* insertion (BDSC 49923) were crossed with transgenic flies expressing *UAS-GFPnls* (BDSC 4776). GFP driven by the *GMR35H02-Gal4* was then compared to

endogenous Slp1 and Slp2 expression. The procedure for making transgenic constructs and strains is described separately.

## Clonal experiments using *ay-Gal4*

The following RNAi lines or *UAS* lines were used for RNAi knockdown experiments or over-expression experiments: *UAS-ey-RNAi* (BDSC 32486), *UAS-scro-RNAi* (BDSC 33890), *UAS-N-RNAi* (BDSC 7078), *UAS-Su(H)-RNAi* (VDRC 103597), *UAS-Dl-RNAi* (VDRC 32788), *UAS-PCNA-RNAi* (VDRC 51253), *UAS-stg-RNAi* (VDRC 17760), *UAS-Dap* (BDSC 83338), *UAS-Scro* (FlyORF: F000666).

*ay-Gal4* (*actin* >FRT-y$^+$-STOP-FRT-Gal4, in which *actin* promoter drives Gal4 expression after a STOP cassette is excised by the action of heat shock activated Flippase) was used to drive RNAi lines or *UAS* lines. Flies of genotype *ywhsFlp; ay-Gal4 >UAS GFP; UAS-Dcr2/Tm6B* were crossed to the RNAi lines. Larvae were heat shocked at 37 °C for 10 min 48 hr after egg laying and then raised at 29 °C, until brains of third instar larvae were dissected and stained.

For observing the effect of *ey-RNAi* on the GFP reporter expression (**Figure 6**) flies of genotype *ywhsFlp; ay-Gal4 >UAS lacZ; ey-RNAi* (made by combining *ay-Gal4 >UAS* lacZ on chromosome II (BDSC 4410) with *UAS-ey-RNAi* on chromosome III) were crossed to flies with genotype *ywhsFlp; Sp/CyO; u8772 220bp >GFP* (enhancer1) or *ywhsFlp; Sp/CyO; d5778 850bp >GFP* (enhancer2).

For studying effects of *Su(H)-RNAi* on GFP reporter expression, flies of genotype *ywhsFlp; ay-Gal4 >lacZ; Tm2/Tm6B* were crossed to flies of genotype *UAS-Dcr2; Su(H)-RNAi; u8772 220>GFP* (enhancer 1) or *UAS-Dcr2; Su(H)-RNAi; d5778 870>GFP* (enhancer 2).

Heat shock protocols for RNAi experiments driven by *ay-Gal4* in the presence of GFP reporters were the same as described above for *ay-Gal4* driven RNAi by themselves (i.e. without GFP reporters).

## *Su(H)* MARCM experiment with *UAS-Ey*

Flies of genotype *ywhsFlp UAS-GFP; FRT40A tubGal80; dpn-Gal4* were crossed to flies of genotype *yw; FRT40A Su(H)$^{47}$ /CyO; UAS-Ey/+*. The progeny larvae were heat-shocked at 37 °C for 45 min 48 hr after egg laying and then raised at 25 °C, until brains of third instar larvae were dissected and stained. Anti-Ey staining is used to distinguish brains carrying UAS-Ey from those that do not.

## Expression of NICD in *ey-RNAi* knockdown clones

Flies of genotype *ywhsFlp; Sp/CyO; UAS-eyRNAi* were crossed to flies of genotype *ywhsFlp; ay-Gal4 UAS-lacZ/CyO; dpn >hsNICD/Tm6B*. Progeny larvae were heat-shocked at 37 °C for 12 min 72 hr after egg-laying. Third instar larval brains were dissected 48 hr after heat shock treatment.

## Expression of Scro in *ey-RNAi* knockdown clones

Flies of genotype *ywhsFlp; ay-Gal4 >UAS GFP; UAS-eyRNAi /Tm6B* were crossed to flies of genotype *UAS-Scro/Tm3*. Progeny larvae were heat-shocked at 37 °C for 9 min 48 hr after egg-laying. Third instar larval brains were dissected 72 hr after heat shock treatment.

## Cell cycle arrest experiments

For some experiments examining the effect of cell cycle arrest on Slp1/2 expression and on the expression of the Notch reporter E(spl)mγGFP, *Vsx-Gal4* was used to drive the expression of *UAS-PCNA-RNAi* and *UAS-Dcr2* in neuroblasts. Flies of genotype *Vsx-G4; E(spl)mγGFP;UAS-Dcr2/Tm6B* were crossed to those with genotype *ywhsFlp; UAS-PCNA-RNAi*. Larvae were shifted to 29 °C 48 hr after egg-laying. Brains of third instar larvae were observed. For other experiments, flies of genotype *ywhsFlp; UAS-PCNA-RNAi* were crossed to flies of genotype *ywhsFlp; ay-Gal4 >UAS GFP; UAS-Dcr2/ Tm6B*. Larvae with genotype *ywhsFLP; ay-Gal4 UAS-GFP /UAS-PCNA-RNAi; UAS-DCR2/+* were heat shocked for 8 min at 37 °C 50 hr after egg-laying (70 hr before they develop into climbing third instar larvae) and were then dissected. For rescuing Notch expression in cell cycle arrested neuroblasts flies of genotype *ywhsFlp; ay-Gal4 >UAS GFP; UAS-Dcr2/Tm6B* were crossed to those with genotype *ywhsFlp; UAS-PCNA-RNAi; dpn >hsNICD*. Larvae of genotype *ywhsFLP; ay-Gal4 UAS-GFP / UAS-PCNA-RNAi; UAS-DCR2/dpn >hsNICD* were then heat shocked for 8 min at 37 °C 50 hr after egg laying (70 hr before the third instar stage) as before and then dissected.

### Negative labeling of *slp* loss-of-function clones

*Ubi-RFP FRT40A/CyO* flies (BDSC 34500) were crossed with flies of genotype *ywhsFlp;Sp/CyO; u8772 220bp >GFP* (enhancer1) and *ywhsFlp;Sp/CyO; d5778 870bp >GFP* (enhancer2) to create strains with genotypes *ywhsFlp; Ubi-RFP FRT40A; u8772 220bp >GFP* and *ywhsFlp; Ubi-RFP FRT40A; d5778 870bp >GFP* respectively. Flies of genotype *ywhsFlp; slp^{S37A}/Sm6-Tm6B* flies (a kind gift from Dr. Andrew Tomlinson) were then crossed to *ywhsFlp; Ubi-RFP FRT40A; u8772 220bp >GFP* and *ywhsFlp; Ubi-RFP FRT40A; d5778 870bp >GFP* and larvae were heat shocked for 45 min 48 hr after egg laying. Third instar larvae were then dissected. Clones carrying two copies of this deficiency are seen as dark regions amidst Ubi >RFP marked wild-type neuroblasts.

## Bioinformatic identification of transcription factor binding sites (TFBS)

Initially, to identify a list of possible candidate TFs with binding sites within our enhancers, we analyzed our enhancer sequences using the MEME suite tools (*Bailey et al., 2015*), TOMTOM (*Gupta et al., 2007*), and especially using an analytic tool developed at EPFL (https://biss.epfl.ch) that uses FIMO (*Grant et al., 2011*) to identify motifs on a sequence (Michael Frochaux, Dr.Bart Deplancke personal communication). However, mutating sites identified by this initial round of analysis did not always result in GFP reporter expression patterns that were consistent with our observations from the genetic experiments as explained earlier. To remedy this, we employed a slightly different strategy to identify all potential binding sites of Ey, Su(H), Slp1, and Scro. We searched for consensus motifs of transcription factor binding sites for Ey, Su(H), Slp1 in JASPAR (*Sandelin et al., 2004*; *Fornes et al., 2020*) and Fly Factor Survey (*Zhu et al., 2011*). We then searched for potential matches to these consensus sequences in our enhancers using the DNA Pattern Find program in Sequence Manipulation Tool (*Stothard, 2000*). Since consensus motifs for binding sites of Scro were not reported in either JASPAR or Fly Factor Survey, we picked 20 related transcription factors of the NK-domain TF family with known binding motifs as templates for finding possible Scro binding sequences (NK2 TFs used are listed in *Figure 3—figure supplement 1*). We looked for potential matches to these motifs within our enhancers using the Pattern Recognition Tool of the Sequence Manipulation Suite. Sequences that were most shared between the 20 NK domain containing TFs were scored as the most likely motifs for Scro binding in the d5778 870 bp enhancer. Given the smaller size of the u8772 220 bp enhancer, only motifs corresponding to Vnd binding sites were mutated since Scro has been suggested to share the greatest homology to Vnd of all NK domain containing TFs (*Zaffran et al., 2000*).

The consensus sequences we used for identifying binding sites for each of our candidate regulators Ey, Su(H), and Scro in the DNA Pattern Recognition program are detailed in *Figure 3*. Using this approach, we found additional potential binding sites for Ey and Su(H) than what were predicted using FIMO. We noticed that predicted binding sites identified by FIMO such as the 220-Ey-Site2, 850-Ey-Site1 and 850-Scro-Site 1 and 6, often had minor mismatches to the consensus binding sequence used as input (for example, the 220-Ey-Site2 had two mismatches to the consensus). Therefore, we also considered sites with a maximum of two mismatches to our consensus sequences for the DNA Pattern Find program as potential binding sites for a particular transcription factor. We detected such potential binding sites that deviated by at most two base pairs from the consensus search sequence using less stringent pattern search conditions in the DNA Pattern Find program.

To test this binding site search method's validity, we cloned custom gene blocks of enhancer segments containing mutated binding sites for each transcription factor into the pJR12 vector. We subsequently made transgenic flies expressing these 'mutant' reporters as described elsewhere in Methods. All transgenes were inserted at the same chromosomal location as the wild-type GFP reporters. Reporter GFP expressions driven by these mutation-carrying enhancers were consistent with evidence from genetic experiments where wild-type GFP reporter expressions were observed in Ey, Su(H), or Scro knockdown or loss of function clones, thus demonstrating the reliability of this approach.

## Plasmids constructs and the making of transgenic fly stocks

Primer sequences for all cloning are provided in a supplementary file (*Supplementary file 2*).

For making all constructs DNA was amplified from the template using Expand High-Fidelity Polymerase (Roche) unless stated otherwise and all constructs were verified by sequencing. All fly embryo

injections for making transgenic flies were carried out either by Bestgene Inc, Chino Hills, CA or by Rainbow Transgenic Flies.

## Generating constructs for reporter assays and enhancer bashing and making transgenic reporter expressing stocks

Sequences corresponding to the fragments of regulatory DNA encoded in GMR35H02 and the d5778 REDfly enhancer were cloned from the BAC clone CH321-94O18( *Venken et al., 2009*) (BACPAC resources). Sequences were PCR amplified from this BAC and cloned into the pJR12 vector (a kind gift from Dr. Jens Rister) between *AscI* and *NotI* sites. Transgenes were inserted at the landing site VK00027 on the third chromosome (BL9744) by φC31 integrase mediated transgenesis (*Groth et al., 2004*) and positive transformants were screened using the *w+* marker originally present in the pJR12 plasmid. Mutated enhancers were custom synthesized as gene blocks (gBlocks, IDT DNA) (*Supplementary file 4*) and cloned as stated above. Split gBlocks were custom made for making the d5778 ey site mutant and the d5778 su(h) site mutants. These split gBlocks were then PCR spliced and cloned between NotI and AscI sites of the pJR12 vector. A custom-made gBlock was used for making the d5778 scro mutant reporter. All reporters were integrated at the same genomic site to ensure comparability across constructs and experiments.

## Generation of CRISPR enhancer deletion constructs and transgenic stocks

CRISPR gRNAs were designed by entering sequences of genomic DNA of the target +/-20 kbp into the CRISPR Optimal Target Finder web utility (*Gratz et al., 2014*) (https://www.targtfinder.flycrispr.neuro.brown.edu). Four gRNAs-two upstream and two downstream of the target region to be deleted were then selected. All four gRNAs were then cloned into the vector pCFD5 (*Port and Bullock, 2016*) (Addgene plasmid #73,914 a kind gift from Dr. Simon Bullock) using NEB-Builder HiFi DNA Assembly master mix. Constructs were then injected into fly strain containing *M{nos-cas9}ZH-2A* (BL54591) along with a 120 bp repair oligonucleotide that contained 60 bp of wild type genome sequence flanking both ends of the expected double-stranded break. After injection, the *nos-cas9* source was eliminated in the subsequent generations by crossing individual G0 progeny flies to a double balancer of genotype *ywhsFlp; Sp/CyO; Tm2/Tm6B* and selecting male G1 progeny. Individual G1 males were then crossed to the same double balancer line to create stocks. The G1 males were then genotyped by PCR to identify whether their genome had been edited, using RedExtract-n-Amp tissue PCR kit (Sigma-Aldrich). The G2 progeny of genome-edited males were then raised until homozygous stocks were established. To delete both enhancers, gRNAs for the d5778 enhancer was injected into a fly line carrying *nos-Cas9* and deletion of the u8772 enhancer and genotyped as previously. Genomic DNA from CRISPR-Cas9 edited flies was extracted using Qiagen DNeasy Blood and Tissue kit according to the manufacturer's instructions. Fragments of genomic DNA flanking the expected deletion sites were amplified by PCR using Platinum SuperFi II PCR master mix (Invitrogen). They were then sequenced by Sanger sequencing to confirm the outcome of the genome modification procedure. The deleted sequences are shown in *Supplementary file 1*.

## Generation of the heat shock inducible NICD construct

We modified the pJR12 vector used in enhancer bashing experiments to replace the GFP coding sequence with coding sequences of our interest. First, we generated a DNA fragment that matched the sequences of the pJR12 vector flanking the eGFP sequence but that did not contain the eGFP sequence itself by PCR splicing. We then cut-out the eGFP segment from the pJR12 vector and replaced it with our GFP deleted fragment using *NsiI* and *XhoI*. Next, an *FRT-stop-FRT3-FRT* segment was cloned from the CoinFlp plasmid (*Bosch et al., 2015*) (Addgene plasmid # 52889, a kind gift from Dr. Iswar Hariharan) and PCR spliced with a fragment of Notch Intracellular Domain (NICD). Sequences corresponding to the Notch Intracellular Domain were PCR amplified from Notch cDNA LD34134 (DGRC). The *FRT-stop-FRT3-FRT-NICD* fragment was then cloned into the modified pJR12 vector between the *PmeI* and *AgeI* restriction sites. The *deadpan* enhancer sequence amplified from the BAC clone CH321-86A18 (BACPAC resources) was then cloned into the modified pJR12 vector to drive the heat shock inducible NICD expressing construct in presence of a heat shock activated Flippase or *hsFlp* (this construct is abbreviated as *dpn >hsNICD*). The construct was inserted at the VK00027 landing site (BL 9744) by φC31 integrase mediated transgenesis. Positive transformants

were identified by the w+eye color marker expression and stocks were created by crossing to a double balancer of genotype *ywhsFlp; Sp/CyO; Tm2/Tm6B*.

## Generation of DamID-fusion constructs (*UAS-Dam-Ey* and *UAS-Dam-Su(H)*) and transgene expressing stocks

Su(H) coding sequence was amplified from a cDNA clone GH10914 (DGRC) by PCR. The amplified Su(H) coding sequence was cloned into the pUAST-Dam-attB vector (*Southall et al., 2013*) (a kind gift from Dr. Andrea Brand) between the *NotI* and *XhoI* sites. A gBlock of Ey coding sequence was ordered from IDT and cloned into the pUAST-LT3- UAS Dam- vector. Constructs were then incorporated at the VK00027 landing site (BL 9744) by φC31 mediated integration. Transformants were identified by expression of the w+marker gene. Final stocks were established by crossing to a double balancer line of genotype *ywhsFlp; Sp/CyO; Tm2/Tm6B*.

## Immunofluorescence staining

Antibody staining was carried out as described in *Li et al., 2013* with a few modifications. The protocol is described briefly as follows: brains from climbing third instar larvae were dissected in 1XPBS and fixed in 4% Formaldehyde solution in 1 X PBS for 30 min on ice. Brains were then incubated in primary antibody solution overnight at 4 °C, washed three times for 30 min each at 4 °C, then incubated in fluor conjugated- secondary antibody solution overnight at 4 °C and then washed again thrice at room temperature each time for 30 min. Samples were mounted in Slowfade Gold antifade reagent (Invitrogen). Images are acquired using a Zeiss LSM500 Confocal Microscope.

Antibodies used in the study are as follows: rabbit anti-Slp1, guinea-pig anti-Slp2, rabbit anti-Ey (all used at 1:500) were kind gifts from Dr. Claude Desplan, guinea-pig anti-Dpn (1:500) (a kind gift from Dr. Chris Doe). Commercially available antibodies include - sheep anti-GFP (1:500, AbD Serotec, 4745–1051), Chicken anti-beta-gal (1:500, Abcam ab9361), Rat anti-Deadpan [11D1BC7] (1:200, Abcam, ab195173) mouse anti-PCNA (1:50, Abcam ab29). These antibodies are provided by the Developmental Studies Hybridoma Bank (DSHB): mouse anti-eyeless (1:10), mouse anti-Pros (MR1A 1:20). Secondary antibodies are from Jackson or Invitrogen.

## Fluorescence in situ hybridization (FISH)

FISH was carried out using custom Stellaris probe sets (LGC Biosearch) as described in *Long et al., 2017* except that incubation with sodium borohydride was skipped. Probes were generated using the *slp1* mRNA (Flybase annotation symbol CG16738-RA) and *slp2* mRNA (Flybase annotation symbol CG2939-RA) as templates. Tiling probes for the *slp1* mRNA were conjugated to the fluorophore Quasar 670 Dye, while probes for *slp2* mRNA were conjugated to the fluorophore Quasar 570 Dye. Concurrent FISH and immunofluorescence staining were carried out as described in the same publication.

## DamID-Seq

Flies of genotype *ywhsFlp; Sp/CyO; UAS-Dam/ Tm6B, ywhsFlp; Sp/CyO; UAS-Ey-Dam/ Tm6B* or *ywhsFlp; Sp/CyO; UAS-Su(H)-Dam /Tm6B* were crossed with *UAS-Dcr2; tubGal80ts; SoxN-Gal4*. Mating crosses were incubated at 25 °C. Eggs laid within an 8-hr window were then shifted to 18 °C for 3 days until they hatched. The expressions of the Dam-fusion proteins were then induced in larval brains by shifting the larvae to 29 °C for 72 hr. Climbing third instar larvae were then dissected. Only the optic lobes were used for further sample preparation. For each sample around 100 brains were dissected.

Dam-ID libraries were made as described in *Marshall et al., 2016*. Briefly, fly brains were dissected in 1 X PBS and genomic DNA was isolated using the QIAamp DNA Micro kit (Qiagen). Genomic DNA was then digested with DpnI and purified using spin column purification (Qiagen PCR-purification kit). To the DpnI digested DNA DamID-PCR adaptors were ligated. Subsequently the adaptor-ligated DNA fragments were digested with DpnII. The DpnII digested genomic DNA fragments were then amplified by PCR to enrich for bound GATC fragments. To make sequencing libraries the PCR enriched genomic DNA sample was sonicated using a Bioruptor bath sonicator (Diagenode), purified using AMPure XP beads, end-repaired and 3'-adenylated. Illumina sequencing indexes were then ligated to these fragments. Index-labeled DNA fragments were then further enriched by PCR amplification,

checked for quality and fragment size distribution on an Agilent Bioanalyzer and by qPCR (Quality control of all libraries was carried out at the Functional Genomics Unit of Roy J. Carver Biotechnology Center, UIUC). Libraries that were deemed acceptable were then sequenced on a single SP100 lane of an Illumina NovaSeq sequencer. Read lengths were 100 bp. Two Biological replicates of *Ey-Dam*, *Su(H)-Dam* and *Dam*-only samples were sequenced, and each biological replicate is defined as one library generated from ~100 dissected brains. The number of reads obtained were as follows: Dam replicate1: 53745480 reads, Dam replicate 2: 85637355 reads, Ey-Dam replicate 1: 81121107 reads, Ey-Dam replicate 2: 82085708 reads, SuH-Dam replicate 1: 77448608 reads, replicate 2: 80015207 reads. All samples exhibited good QC scores.

## DamID-Seq data analysis

DamID-seq data was analyzed using the damidseq-pipeline (*Marshall and Brand, 2015*). Duplicate samples of Ey-Dam and Su(H)-Dam samples and Dam-only controls were aligned to the *Drosophila* reference genome UCSC dm6. Alignment rate for individual samples were as follows:Ey-Dam replicate 1%–97.04%, Ey-Dam replicate 2%–97.82%, Su(H)-Dam replicate 1%–97.91% and Su(H)-Dam replicate 2%–97.82%. The main utility- the damid-seqpipeline was used to align reads to the genome using bowtie2, bin and count reads, normalize counts and generate log-2 ratio bedgraph files for each DamID-sample and its corresponding Dam-only control. The provided gatc.track file was used for running the script. Next, the findpeaks utility was used with an F.D.R. <0.01 to identify peaks. We then used the provided peaks2genes utility to assign peaks to genes. To assess the reproducibility of our data we also ran the findpeaks script using F.D.R <0.1 to discover peaks with weaker statistical confidence and used this as input for the I.D.R. Python package (https://github.com/nboley/idr; *Boley, 2017*). 984 of 1810 (54.4.%) Ey-Dam and 972 of 1996 (48.7%) of Su(H)-Dam peaks passed an I.D.R. threshold of 0.05. All replicate tracks for Ey and for Su(H) as well as reproducible peaks identified using I.D.R. were visualized in IGV (*Robinson et al., 2011*). To generate the lists of genes bound reproducibly by Ey or Su(H), files containing lists of peaks that had passed the I.D.R. cutoff were processed using the peaks2genes utility provided along with the damidseq-pipeline. Peaks were mapped to particular genes if they were within a 1000 bp neighborhood of the gene loci.

## Image quantification and statistical analysis

No statistical method was used to predetermine sample size. Sample size /replicate numbers were decided based on the routine in the field and previous experience. All data were obtained from at least two independent experiments. The numbers of animals or clones analyzed can be found in figure legends or this section (see below). All data are highly reproducible. No outliers are excluded. For specific quantification methods and statistical analysis see below.

### Imaging and image analysis for comparison of wild-type and mutant GFP reporter intensities

All images were acquired using a Zeiss LSM 500 confocal microscope. Samples were imaged at sub-saturating illumination conditions, with stack intervals of 1 micron. Same laser settings and imaging conditions were used across all variants of the same enhancers. Image analysis was carried out using Fiji (*Schindelin et al., 2012*). Signal intensities in the GFP and the Dpn channels were measured over a small rectangular selection for all variants. For the same enhancer a 'scaling factor' was calculated for each mutant by dividing the Dpn channel intensity for the mutant reporter by the channel intensity of the corresponding wild-type reporter. The GFP channel intensities for each variant were then multiplied by this scaling factor. Finally, a ratio of the scaled GFP channel intensity to the Dpn channel intensity was calculated for each GFP reporter variant. Ratios of scaled GFP/Dpn intensities were then plotted using GraphPad Prism. For each enhancer, Ordinary One-way ANOVA and Dunnett's multiple comparisons test were carried out between reporter variants of the same enhancer. For the d5778 870 bp enhancer, five sets of data from twenty different optic lobes were acquired for each enhancer (one set- wild type and all mutant variations all imaged using the same conditions). For the u8772 220 bp enhancer three sets of data were used for quantification. For the *UAS-GFP-RNA*i; d5778 850-Su(H) binding sites mutants crossed with *Repo-Gal4*, three sets of data were used for quantification.

## Image analysis for CRISPR enhancer deletion experiments

All images were acquired using a Zeiss LSM 500 confocal microscope under sub-saturating illumination conditions. For each sample width of Slp1, Slp2, and Dpn expression domains were measured in Fiji at five different locations and averaged. The brain size of the sample was estimated by measuring the length of the brain at its widest point perpendicular to the neuroblast division axis. The width of each protein expression domain was divided by the brain size to normalize expression domain size estimates to the brain size. For each protein Slp1, Slp2, and Dpn- these ratios were then plotted in GraphPad Prism and ordinary one way ANOVA and Dunnett's multiple comparisons tests were carried out between wild-type and CRISPR enhancer deleted brains. Data from seven brains of each genotype was analyzed and quantified.

All images were processed in Adobe Photoshop and assembled using Adobe Illustrator.

## Materials and correspondence

Publicly available fly lines (those from BDSC or VDRC) should be requested directly from the corresponding stock centers: https://bdsc.indiana.edu/ or https://stockcenter.vdrc.at/control/main. Fly lines generated in this study can be requested without restriction. Correspondence and requests for materials should be addressed to Xin Li (https://mcb.illinois.edu/faculty/profile/lixin/).

## Acknowledgements

We thank the Functional Genomics Unit and the DNA-Sequencing Services Unit of the Roy J Carver Biotechnology Center at the University of Illinois at Urbana-Champaign for assistance with the quality control and sequencing of Dam-ID sequencing libraries. We thank the fly community, especially Claude Desplan, Andrea Brand, Chris Doe, Bruce Edgar, Kenneth Irvine, Miki Fujioka and James B Jaynes for generous gifts of antibodies and fly stocks. We are grateful to Bart Deplancke's team for help with the YIH pipeline to predict transcription factor binding sites within enhancers. We thank the Bloomington *Drosophila* Stock Center, the Vienna *Drosophila* RNAi Center, the Developmental Studies Hybridoma Bank, and TriP at Harvard Medical School (NIH/NIGMS R01-GM084947) for fly stocks and reagents. We thank members of the Li lab, Dr. Yu Zhang and Hailun Zhu for helpful discussions. We thank the editors and reviewers for very constructive suggestions that helped us significantly improve the mansucript during the revision. This work was supported by the National Eye Institute (Grant 1 R01 EY026965-01A1) and start-up funds from the University of Illinois at Urbana-Champaign.

## Additional information

### Funding

| Funder | Grant reference number | Author |
|---|---|---|
| National Eye Institute | Grant 1 R01 EY026965-01A1 | Xin Li |

The funders had no role in study design, data collection and interpretation, or the decision to submit the work for publication.

### Author contributions

Alokananda Ray, Conceptualization, Data curation, Formal analysis, Validation, Investigation, Visualization, Methodology, Writing - original draft, Writing - review and editing; Xin Li, Conceptualization, Resources, Data curation, Supervision, Funding acquisition, Validation, Investigation, Methodology, Writing - original draft, Project administration, Writing - review and editing

### Author ORCIDs

Xin Li (iD) http://orcid.org/0000-0002-3428-3569

### Decision letter and Author response

Decision letter https://doi.org/10.7554/eLife.75879.sa1
Author response https://doi.org/10.7554/eLife.75879.sa2

## Additional files

### Supplementary files

- Supplementary file 1. Sequences deleted in the *slp* double enhancer deletion line.
- Supplementary file 2. List of primers used for experiments described.
- Supplementary file 3. Predicted transcription factor binding sites by the web-based FIMO analysis.
- Supplementary file 4. List of gene blocks used for reporter assays and for cloning Ey coding sequence into the pUAST-Dam-attB plasmid.
- Supplementary file 5. Gene lists that have IDR peaks for both Su(H) and Ey, Ey only, and Su(H) only from the DamID-seq data.
- Supplementary file 6. DAVID analysis of gene lists that have IDR peaks for both Su(H) and Ey, Ey only, and Su(H) only from the DamID-seq data.
- Transparent reporting form

### Data availability

The raw and processed DamID-seq data have been deposited to GEO, and the accession number is GSE188643. All data generated or analysed during this study are included in the manuscript and supporting file; Source Data files have been provided for Figures 7,8,9 and Figure 2-figure supplement2.

The following dataset was generated:

| Author(s) | Year | Dataset title | Dataset URL | Database and Identifier |
|---|---|---|---|---|
| Ray A, Li X | 2021 | A Notch-dependent transcriptional mechanism controls expression of temporal patterning factors in *Drosophila* medulla | https://www.ncbi.nlm.nih.gov/geo/query/acc.cgi?acc=GSE188643 | NCBI Gene Expression Omnibus, GSE188643 |

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

# Appendix 1

## Appendix 1—key resources table

| Reagent type (species) or resource | Designation | Source or reference | Identifiers | Additional information |
|---|---|---|---|---|
| Antibody | anti SoxN (Rabbit polyclonal) | Claude Desplan | N/A | IF 1:250 |
| Antibody | anti- Ey (Rabbit polyclonal) | Claude Desplan | N/A | IF 1:500 |
| Antibody | anti-Slp1 (Rabbit polyclonal) | Claude Desplan | N/A | IF 1:500 |
| Antibody | anti-Slp2 (Guinea-pig polyclonal) | Claude Desplan | N/A | IF 1:500 |
| Antibody | anti-Scro (Guinea-pig polyclonal) | Claude Desplan | N/A | IF 1:100 |
| Antibody | Anti-D (Rabbit polyclonal) | Claude Desplan | N/A | IF 1:500 |
| Antibody | Anti-Dpn (Guinea pig polyclonal) | Chris Doe | N/A | IF 1:500 |
| Antibody | Anti-Dpn (Rat monoclonal) | Abcam | Antibody#: Ab195173 | IF 1:500 |
| Antibody | Anti-B-H1 (Rat polyclonal) | Tiffany Cook | N/A | IF 1:200; Tiffany Cook |
| Antibody | Anti-GFP (Sheep polyclonal) | AbD Serotec | 4745–1051 | IF 1:500 |
| Antibody | Anti-Repo (mouse monoclonal) | Developmental Studies Hybridoma Bank | 8D12 anti-Repo | IF 1:50 |
| Antibody | Anti-PCNA (mouse monoclonal) | Abcam | ab29 | IF 1:10 |
| Antibody | Anti-betaGalactosidase (Chicken polyclonal) | Abcam | Antibody#: Ab9361 | IF 1:500 |
| Antibody | Cy5 AffiniPure Anti-RatIgG (Donkey polyclonal) | Jackson ImmunoResearch Laboratories Inc | Catalog_#: 712-175-153 | IF!:500 |
| Antibody | Cy3 AffiniPure Anti-RatIgG (Donkey polyclonal) | Jackson ImmunoResearch Laboratories Inc | Catalog_#: 712-165-153 | IF 1:500 |
| Antibody | Cy3 AffiniPure Anti-Guinea Pig IgG (Donkey polyclonal) | Jackson ImmunoResearch Laboratories Inc | Catalog_#: 706-165-148 | IF 1:500 |
| Antibody | Alexa Fluor 647 AffiniPure Anti-Guinea Pig (Donkey polyclonal) | Jackson ImmunoResearch Laboratories Inc | Catalog_#: 706-605-148 | IF 1:500 |
| Antibody | Alexa Fluor 488 AffiniPure Anti-Guinea Pig IgG (Donkey polyclonal) | Jackson ImmunoResearch Laboratories Inc | Catalog_#: 706-545-148 | IF 1:500 |
| Antibody | Alexa Fluor 647 AffiniPure Anti-Goat IgG (Donkey polyclonal) | Jackson ImmunoResearch Laboratories Inc | Catalog_#: 705-605-147 | IF 1:500 |
| Antibody | DyLight 405 AffiniPure Anti-Mouse IgG (Donkey polyclonal) | Jackson ImmunoResearch Laboratories Inc | Catalog_#: 715-475-151 | IF 1:500 |
| Antibody | Alexa Fluor 647 AffiniPure Anti-Rabbit IgG (Donkey polyclonal) | Jackson ImmunoResearch Laboratories Inc | Catalog_#: 711-605-152 | IF 1:500 |
| Antibody | Cy5 AffiniPure Anti-Mouse IgG (Donkey polyclonal) | Jackson ImmunoResearch Laboratories Inc | Catalog_#: 715-175-151 | IF 1:500 |
| Antibody | Alexa Fluor 647 AffiniPure Anti-Chicken IgY (IgG) (Donkey polyclonal) | Jackson ImmunoResearch Laboratories Inc | Catalog_#: 703-605-155 | IF 1:500 |
| Antibody | Alexa Fluor 488 AffiniPureDonkey Anti-Mouse IgG (Donkey polyclonal) | Jackson ImmunoResearch Laboratories Inc | Catalog_#: 715-545-151 | IF 1:500 |
| Antibody | Alexa Fluor 488 AffiniPure Anti-Sheep (Donkey polyclonal) | Jackson ImmunoResearch Laboratories Inc | Catalog_#: 713-545-147 | IF 1:500 |
| Antibody | DyLight 405 AffiniPure Anti-Rat IgG (Donkey polyclonal) | Jackson ImmunoResearch Laboratories Inc | Catalog_#: 712-475-153 | IF 1:500 |
| Antibody | DyLight 405 AffiniPure Anti-Rabbit IgG (Donkey polyclonal) | Jackson ImmunoResearch Laboratories Inc | Catalog_#: 711-475-152 | IF 1:500 |

*Appendix 1 Continued on next page*

*Appendix 1 Continued*

| Reagent type (species) or resource | Designation | Source or reference | Identifiers | Additional information |
|---|---|---|---|---|
| Antibody | anti-Rabbit IgG, Alexa Fluor Â 555 conjugate (Donkey polyclonal) | Life Technologies | Catalog_#: A-31572 | IF 1:500 |
| Antibody | Anti-Opa (Rabbit polyclonal) | J. Peter Gergen | N/A | IF 1:100; J. Peter Gergen |
| Antibody | Anti-Erm (Rat polyclonal) | Claude Desplan | N/A | 1:100 |
| Recombinant DNA Reagent | pJR12 plasmid | *Rister et al., 2015* PMID:26785491 DOI: 10.1126/science.aab3417 | N/A | Jens Rister |
| Recombinant DNA Reagent | pCFD5 plasmid | Addgene | Addgene Plasmid #73,914 | Fillip Port |
| Recombinant DNA Reagent | UAS-LT3-Dam plasmid | Andrea Brand | N/A | |
| Recombinant DNA Reagent | pJR12-hsNICD plasmid | This paper | N/A | See section on 'Plasmid constructs' in 'Materials and methods' |
| Recombinant DNA Reagent | UAS-LT3-DamEy | This paper | N/A | See section on 'Plasmid constructs' in 'Materials and methods' |
| Recombinant DNA Reagent | UAS-LT3-DamSu(H) | This paper | N/A | See section on 'Plasmid constructs' in 'Materials and methods' |
| Recombinant DNA Reagent | CoinFlp plasmid | Addgene | Addgene Plasmid #52,889 | Iswar Hariharan |
| Recombinant DNA Reagent | GMR35H02 BAC | BACPAC resources | CH321-94O18 | |
| Recombinant DNA Reagent | Su(H) cDNA. | *Drosophila* Genomicsa Resource Center | GH10914 | |
| Recombinant DNA Reagent | Notch cDNA | *Drosophila* Genomics Resource Center | LD34134 | |
| Recombinant DNA Reagent | CH321-86A18 (BAC) | BACPAC resources | | BAC encoding *dpn* enhancer |
| Sequence-based reagent | 220-Ey-1d gBlock | This study | N/A | See *Supplementary file 4* for sequence |
| Sequence-based reagent | 220-Ey-2m gBlock | This study | N/A | See *Supplementary file 4* for sequence |
| Sequence-based reagent | 220-SuH-1–4 m | This study | N/A | See *Supplementary file 4* for sequence |
| Sequence-based reagent | 220-SuH-1m4m | This study | N/A | See *Supplementary file 4* for sequence |
| Sequence-based reagent | 220-Scro-1m | This study | N/A | See *Supplementary file 4* for sequence |
| Sequence-based reagent | 220-Slp1-1m2m | This study | N/A | See *Supplementary file 4* for sequence |
| Sequence-based reagent | 850-Ey-1m2d3d | This study | N/A | See *Supplementary file 4* for sequence |
| Sequence-based reagent | 850-SuH-1m2m3m6m | This study | N/A | See *Supplementary file 4* for sequence |
| Sequence-based reagent | 850-SuH-1–6 m | This study | N/A | See *Supplementary file 4* for sequence |
| Sequence-based reagent | 850-Scro-1-7d | This study | N/A | See *Supplementary file 4* for sequence |
| Sequence-based reagent | 850-Slp1-1d | This study | N/A | See *Supplementary file 4* for sequence |
| Sequence-based reagent | Ey gene block | This study | N/A | See *Supplementary file 4* for sequence |

*Appendix 1 Continued on next page*

*Appendix 1 Continued*

| Reagent type (species) or resource | Designation | Source or reference | Identifiers | Additional information |
|---|---|---|---|---|
| Sequence-based reagent | SlpR2-2FP | This study | N/A | 5'TTAGGCGCGCCAGTGC GTGTCTGCCCTTTCATTTTG 3' |
| Sequence-based reagent | SlpR2-2RP | This study | N/A | 5' ATATATGCGGCCGCCCC AGCTAGCTCCCTTCACTCTTCT 3' |
| Sequence-based reagent | SlpL1-5FP | This study | N/A | 5' TTAGGCGCGCCGAA TCGAAATGCTTCCCCGCCTCG 3' |
| Sequence-based reagent | SlpL1-5RP | This study | N/A | 5' ATATATGCGGCCGCTGAA CGTGCAACATCAAAGGCCGC 3' |
| Sequence-based reagent | SlpM1-FP | This study | N/A | 5' TTAGGCGCGCCGCGGCCT TTGATGTTGCACGTTCA 3' |
| Sequence-based reagent | SlpM1-RP | This study | N/A | 5' ATATATGCGGCCGCGGA CAGTTCGGAATGTGCCTCGA 3' |
| Sequence-based reagent | Slpf1-FP | This study | N/A | 5' TTAGGCGCGCCGAATCG AGTGGTGAGCGATAG 3' |
| Sequence-based reagent | Slpf1-RP | This study | N/A | 5' ATATATGCGGCCGCTCTG ATATTTTTCACGGCTCA 3' |
| Sequence-based reagent | Slpf2-FP | This study | N/A | 5' TTAGGCGCGCCTTCGA CCTTGTAGTGGCAAG 3' |
| Sequence-based reagent | Slpf2-RP | This study | N/A | 5' ATATATGCGGCCGCCG GAGATCGGAAGGTTAGTG 3' |
| Sequence-based reagent | Slpf3-FP | This study | N/A | 5' TTAGGCGCGCCTCTCC TTGTTGCTCCTCACA 3' |
| Sequence-based reagent | Slpf3-RP | This study | N/A | 5' ATATATGCGGCCGCTG AACGTGCAACATCAAAGG 3' |
| Sequence-based reagent | d5778FP | This study | N/A | 5' TTAGGCGCGCCTGGTC TTTTACGTTAAT CTGGGCAGCT 3' |
| Sequence-based reagent | d5778RP | This study | N/A | 5' ATATATGCGGCCGCA CATTACGCATTGCA TTCCTCCTCCTT 3' |
| Sequence-based reagent | d5778-850FP | This study | N/A | 5' TTAGGCGCGCCC ATTAACTCGAGTCT GGTTTCCGAT 3' |
| Sequence-based reagent | d5778-850RP | This study | N/A | 5' ATATATGCGGCCG CCGTACATATTCTCC AGGAGTTCGGTC 3' |
| Sequence-based reagent | pJRLPseq3 | This study | N/A | 5' AGATGGGTGAG GTGGAGTACG 3' |
| Sequence-based reagent | pJR12-TATA-seq | This study | N/A | 5' AGCTGCGCTTGT TTATTTGCTTAG 3' |
| Sequence-based reagent | SuHDamFP | This study | N/A | 5' ATATATGCGGCCGC AAATGAAGAGCTACA GCCAATTTAATTT AAACGCCGCC 3' |
| Sequence-based reagent | SuHDamRP | This study | N/A | 5' AAAATACTCGAGTC AGGATAAGCCGCTAC CATGACTATTCCATTGC 3' |
| Sequence-based reagent | DamseqFP1 | This study | N/A | 5' TGAGGGGAGAC ATAGTACTGGT 3' |
| Sequence-based reagent | DamseqFP2 | This study | N/A | 5' GAAGGTCTGGTT GAGCGCCATA 3' |
| Sequence-based reagent | pCFD5-u8772-gRFP1 | This study | N/A | 5' GCGGCCCGGGTTCGA TTCCCGGCCGATGCATC GAAATTTCCTGGTATTCG GTTTTAGAGCTAGAA ATAGCAAG 3' |
| Sequence-based reagent | pCFD5-u8772-gRRP1 | This study | N/A | 5' CGCCTCGCCCAAA TGCATTTTGCACCAG CCGGGAATCGAACCC 3' |

*Appendix 1 Continued on next page*

Appendix 1 Continued

| Reagent type (species) or resource | Designation | Source or reference | Identifiers | Additional information |
|---|---|---|---|---|
| Sequence-based reagent | pCFD5-u8772-gRFP2 | This study | N/A | 5' AAATGCATTTGGGC GAGGCGGTTTTAGAG CTAGAAATAGCAAG 3' |
| Sequence-based reagent | pCFD5-u8772-gRRP2 | This study | N/A | 5' TTAGTTTGATTGTTT CGAAGTGCACCAGCC GGGAATCGAACCC 3' |
| Sequence-based reagent | pCFD5-u8772-gRFP3 | This study | N/A | 5' CTTCGAAACAATCA AACTAAGTTTTAGAGC TAGAAATAGCAAG 3' |
| Sequence-based reagent | pCFD5-u8772-gRRP3 | This study | N/A | 5' ATTTTAACTTGCTATTT CTAGCTCTAAAACAAACT GGAAGTCATTGACCCTG CACCAGCCGGGA ATCGAACCC 3' |
| Sequence-based reagent | pCFD5-d5778-gRFP1 | This study | N/A | 5' GCGGCCCGGGTTCG ATTCCCGGCCGATGCA ATAAGTCCTTGGGTAAT ACGGTTTTAGAGCTAG AAATAGCAAG 3' |
| Sequence-based reagent | pCFD5-d5778-gRRP1 | This study | N/A | 5' AACATTTATCTAGGA CATCTTGCACCAGCC GGGAATCGAACCC 3' |
| Sequence-based reagent | pCFD5-d5778-gRFP2 | This study | N/A | 5' AGATGTCCTAGATA AATGTTGTTTTAGAG CTAGAAATAGCAAG 3' |
| Sequence-based reagent | pCFD5-d5778-gRRP2 | This study | N/A | 5' TGTTGGCAAGCGG CGCTTCATGCACCAG CCGGGAATCGAACCC 3' |
| Sequence-based reagent | pCFD5-d5778-gRFP3 | This study | N/A | 5' TGAAGCGCCGCTT GCCAACAGTTTTAGA GCTAGAAATAGCAAG 3' |
| Sequence-based reagent | pCFD5-d5778-gRRP3 | This study | N/A | 5' ATTTTAACTTGCTATTT CTAGCTCTAAAACTTTCG ATATCCCAGCTCCTTTGC ACCAGCCGGGAAT CGAACCC 3' |
| Sequence-based reagent | u8772crdelFP | This study | N/A | 5' TTGCAAATACTTTTT ATTCAAGGAATCGAC 3' |
| Sequence-based reagent | u8772crdelRP | This study | N/A | 5' AATCTCAAGTTTGGT GTTTGTAATTTTTGG 3' |
| Sequence-based reagent | d5778crdelFP | This study | N/A | 5' CTATTGAAGGGCGG ACATATTAGACAACAA TTGGATCGCTTG 3' |
| Sequence-based reagent | d5778crdelRP | This study | N/A | 5' CTGCATTCCATCC CGTCGCATCCTTGTC 3' |
| Sequence-based reagent | pJRGFPdelFP1 | This study | N/A | 5' TTAGAGATGCATCT CAAAAAAATGGTGGG CATAATAGTGTTGTTTA TATATATCAAAAATAACAAC 3' |
| Sequence-based reagent | pJRGFPdelRP1 | This study | N/A | 5' CCACCGGTCGCCA CCGACGTCAGC GGCCGGCCGC 3' |
| Sequence-based reagent | pJRGFPdelFP2 | This study | N/A | 5' GTCGCGGCCGGC CGCTGACGTCGGTG GCGACCGGTGGATC GTTTAAACAGGCC 3' |
| Sequence-based reagent | pJRGFPdelRP2 | This study | N/A | 5' CAATAACTCGAGG AGCGCCGGAGT AT AAATAGAGGCGCT TCGTCTACG 3' |
| Sequence-based reagent | pJRGFPdelseqFP | This study | N/A | 5' CCATTATAAGCTGC AATAAACAAGTTAACAAC 3' |

Appendix 1 Continued on next page

*Appendix 1 Continued*

| Reagent type (species) or resource | Designation | Source or reference | Identifiers | Additional information |
|---|---|---|---|---|
| Sequence-based reagent | pJRGFPdelseqRP | This study | N/A | 5' GTCGCTAAGCG AAAGCTAAGC 3' |
| Sequence-based reagent | U63seqfwd | This study | N/A | 5' ACGTTTTATAACT TATGCCCCTAAG 3' |
| Sequence-based reagent | pCFD5seqrev | This study | N/A | 5' GCACAATTGTCT AGAATGCATAC 3' |
| Sequence-based reagent | dpnenFP | This study | N/A | 5' TTAGGCGCGCCC TTCGCTTTTGCCTG GTCGGCTCATCGG 3' |
| Sequence-based reagent | dpnenRP | This study | N/A | 5' ATATATGCGGCCG CACGCCTCGTCCT GGCACCCTC 3' |
| Sequence-based reagent | NICDFP | This study | N/A | 5' TATTTAACCGGTTAT TATCAAATGTAGATGG CCTCGGAACCCTTG 3' |
| Sequence-based reagent | NICDRP | This study | N/A | 5' ATAATAGTTTAAACATG AGTACGCAAAGAAAG CGGGCAC 3' |
| Sequence-based reagent | hscFP1 | This study | N/A | 5' TATTTAACCGGTTAT TATCAAATGTAGATGG CCTCGGAACCCTTG 3' |
| Sequence-based reagent | hscRP1 | This study | N/A | 5' GGAAGTTCCTATTCT CTAGAAAGTATAGGAA CTTCGAATTCCAAAAT GAGTACGCAAAGAAA GCGGGCAC 3' |
| Sequence-based reagent | hscFP2 | This study | N/A | 5' GTGCCCGCTTTCTTT GCGTACTCATTTTGGAA TTCGAAGTTCCTATACT TTCTAGAGAATAGG AACTTCC 3' |
| Sequence-based reagent | hscRP2 | This study | N/A | 5' ATAATAGTTTAAAC GAAGTTCCTATTCTC TAGAAAGTATAGGAA CTTCCCCGC 3' |
| Genetic reagent (*D. melanogaster*) | *UAS-ey -RNAi* | Bloomington *Drosophila* Stock Centre | BDSC 32486; Symbol: CG1464; Flybase ID:FBgn0005558 | |
| Genetic reagent (*D. melanogaster*) | *UAS-Su(H)-RNAi* | Vienna *Drosophila* Stock Centre | VDRC 103597 Symbol: CG3497; Flybase ID: FBgn0004837 | |
| Genetic reagent (*D. melanogaster*) | *UAS-N-RNAi* | Bloomington *Drosophila* Stock Centre | BDSC 7078; Symbol: CG3936; Flybase ID: FBgn0004647 | |
| Genetic reagent (*D. melanogaster*) | *UAS-Dl-RNAi* | Vienna *Drosophila* Stock Centre | VDRC 32788; Symbol: CG3619; Flybase ID: FBgn0000463 | |
| Genetic reagent (*D. melanogaster*) | *UAS-PCNA-RNAi* | Vienna *Drosophila* Stock Centre | VDRC 51253; Symbol: CG9193; Flybase ID: FBgn0005655 | |
| Genetic reagent (*D. melanogaster*) | *UAS-mam-DN* | Justin Kumar | Symbol: CG8118; Flybase ID: FBgn0002643 | Justin Kumar |
| Genetic reagent (*D. melanogaster*) | *UAS-stg-RNAi* | Vienna *Drosophila* Stock Centre | VDRC 17760; Symbol: CG1395 Flybase ID: FBgn0003525 | |

*Appendix 1 Continued on next page*

*Appendix 1 Continued*

| Reagent type (species) or resource | Designation | Source or reference | Identifiers | Additional information |
|---|---|---|---|---|
| Genetic reagent (*D. melanogaster*) | UAS-dap | Bloomington *Drosophila* Stock Centre | BDSC 83338; Symbol: CG1772 Flybase ID: FBgn0010316 | |
| Genetic reagent (*D. melanogaster*) | UAS-scro-RNAi | Bloomington *Drosophila* Stock Centre | BDSC 33890; Symbol: CG17594 Flybase ID: FBgn0287186 | |
| Genetic reagent (*D. melanogaster*) | UAS-Ey | Bloomington *Drosophila* Stock Centre | BDSC56560 | |
| Genetic reagent (*D. melanogaster*) | UAS-Scro | FlyORF | F000666 | |
| Genetic reagent (*D. melanogaster*) | GMR35H02-Gal4 | Bloomington *Drosophila* Stock Centre (**Pfeiffer et al., 2008**) | BDSC 49923 | |
| Genetic reagent (*D. melanogaster*) | GMR41H10-Gal4 (SoxN-Gal4) | Bloomington *Drosophila* Stock Centre (**Pfeiffer et al., 2008**) | N/A | No longer available from BDSC |
| Genetic reagent (*D. melanogaster*) | UAS-GFP-nls | Bloomington *Drosophila* Stock Centre | BDSC 4776 | |
| Genetic reagent (*D. melanogaster*) | UAS-EGFP-RNAi | Bloomington *Drosophila* Stock Centre | BDSC 9931 | |
| Genetic reagent (*D. melanogaster*) | ayGal4 "y w hsFLP; act >y +> Gal4 UAS-GFP / CyO" | **Ito et al., 1997** PMID:9043058 DOI: 10.1242/dev.124.4.761 | N/A | |
| Genetic reagent (*D. melanogaster*) | "y w hs FLP; act >y +> Gal4 UAS GFP / CyO; UASDCR2/ TM6B" | **Zhu et al., 2022** PMID:35273186 DOI: 10.1038/s41467-022-28915-3 | N/A | |
| Genetic reagent (*D. melanogaster*) | ayGal4 UASlacZ | Bloomington *Drosophila* Stock Centre | BDSC 4410 | |
| Genetic reagent (*D. melanogaster*) | "y,w, hsFLP, UASCD8GFP; FRT40A tubGal80; tubGal4/ TM6B" | Liqun Luo | N/A | |
| Genetic reagent (*D. melanogaster*) | "y,w,; UbiRFPnls FRT40A/CyO" | Bloomington *Drosophila* Stock Centre | BDSC 34500 | |
| Genetic reagent (*D. melanogaster*) | FRT40A slp$^{S37A}$ /SM6-TM6B | **Sato and Tomlinson, 2007** PMID:17215299 DOI: 10.1242/dev.02786 | N/A | Andrew Tomlinson |
| Genetic reagent (*D. melanogaster*) | E(spl)mγGFP | **Almeida and Bray, 2005** PMID:16275038 DOI: 10.1016 /j.mod.2005.08.004 | N/A | |
| Genetic reagent (*D. melanogaster*) | "VsxGal4;Dpn-LacZ/CyO; UAS-Dcr2/TM6B" | **Erclik et al., 2017** PMID:28077877 DOI: 10.1038/nature20794 | N/A | |
| Genetic reagent (*D. melanogaster*) | dpn-Gal4 (GMR13C02) | Bloomington *Drosophila* Stock Centre | BDSC47859 | |
| Genetic reagent (*D. melanogaster*) | repo-Gal4 | Bloomington *Drosophila* Stock Centre | BDSC7415 | |
| Genetic reagent (*D. melanogaster*) | "y,w,hsFlp; Sp/CyO; u8772 220 GFP/Tm6B" | This study | N/A | See section on 'Fly stocks and genetics' in 'Materials and Methods' |
| Genetic reagent (*D. melanogaster*) | "y,w,hsFlp; Sp/CyO; d5778 850 GFP/Tm6B" | This study | N/A | See section on 'Fly stocks and genetics' in 'Materials and Methods' |

*Appendix 1 Continued on next page*

Appendix 1 Continued

| Reagent type (species) or resource | Designation | Source or reference | Identifiers | Additional information |
|---|---|---|---|---|
| Genetic reagent (*D. melanogaster*) | "y,w, hsFlp; Sp/CyO; 220-Ey-1d/Tm6B" | This study | N/A | See section on 'Fly stocks and genetics' in 'Materials and Methods' |
| Genetic reagent (*D. melanogaster*) | "y,w, hsFlp; Sp/CyO; 220-Ey-2m/Tm6B" | This study | N/A | See section on 'Fly stocks and genetics' in 'Materials and Methods' |
| Genetic reagent (*D. melanogaster*) | "y,w, hsFlp; Sp/CyO; 220-SuH-1–4 m/Tm6B" | This study | N/A | See section on 'Fly stocks and genetics' in 'Materials and Methods' |
| Genetic reagent (*D. melanogaster*) | "y,w, hsFlp; Sp/CyO; 220-SuH-1m4m/Tm6B" | This study | N/A | See section on 'Fly stocks and genetics' in 'Materials and Methods' |
| Genetic reagent (*D. melanogaster*) | "y,w, hsFlp; Sp/CyO; 220-Scro-1m/Tm6B" | This study | N/A | See section on 'Fly stocks and genetics' in 'Materials and Methods' |
| Genetic reagent (*D. melanogaster*) | "y,w, hsFlp; Sp/CyO; 220-Slp1-1m2m/Tm6B" | This study | N/A | See section on 'Fly stocks and genetics' in 'Materials and Methods' |
| Genetic reagent (*D. melanogaster*) | "y,w, hsFlp; Sp/CyO; 850-Ey-1m2d3d/Tm6B" | This study | N/A | See section on 'Fly stocks and genetics' in 'Materials and Methods' |
| Genetic reagent (*D. melanogaster*) | "y,w, hsFlp; Sp/CyO; 850-SuH-1m3m6m/Tm6B" | This study | N/A | See section on 'Fly stocks and genetics' in 'Materials and Methods' |
| Genetic reagent (*D. melanogaster*) | "y,w, hsFlp; Sp/CyO; 850-SuH-1m2m3m6m/Tm6B" | This study | N/A | See section on 'Fly stocks and genetics' in 'Materials and Methods' |
| Genetic reagent (*D. melanogaster*) | "y,w, hsFlp; Sp/CyO; 850-SuH-1–6 m/Tm6B" | This study | N/A | See section on 'Fly stocks and genetics' in 'Materials and Methods' |
| Genetic reagent (*D. melanogaster*) | "y,w, hsFlp; Sp/CyO; 850-Scro-1-7d/Tm6B" | This study | N/A | See section on 'Fly stocks and genetics' in 'Materials and Methods' |
| Genetic reagent (*D. melanogaster*) | "y,w, hsFlp; Sp/CyO; 850-Slp1-1d/Tm6B" | This study | N/A | See section on 'Fly stocks and genetics' in 'Materials and Methods' |
| Genetic reagent (*D. melanogaster*) | "y,w, hsFlp; Sp/CyO; UAS-DamEy/Tm6B" | This study | N/A | See section on 'Fly stocks and genetics' in 'Materials and Methods' |
| Genetic reagent (*D. melanogaster*) | "y,w, hsFlp; Sp/CyO; UAS-SuHDam/Tm6B" | This study | N/A | See section on 'Fly stocks and genetics' in 'Materials and Methods' |
| Genetic reagent (*D. melanogaster*) | "y,w,hsFlp; Sp/CyO; UAS-Dam/Tm6B" | This study | N/A | See section on 'Fly stocks and genetics' in 'Materials and Methods' |
| Genetic reagent (*D. melanogaster*) | "Dcr2; tubG80ts; SoxNG4" | This study | N/A | See section on 'Fly stocks and genetics' in 'Materials and Methods' |
| Genetic reagent (*D. melanogaster*) | "Dcr2; Su(H)RNAi/CyO; u8772-220-GFP/ Tm6B" | This study | N/A | See section on 'Fly stocks and genetics' in 'Materials and Methods' |
| Genetic reagent (*D. melanogaster*) | "Dcr2; Su(H)RNAi/CyO; d5778-850-GFP/Tm6B" | This study | N/A | See section on 'Fly stocks and genetics' in 'Materials and Methods' |
| Genetic reagent (*D. melanogaster*) | "y,w,hsFlp; Sp/CyO; UAS-eyRNAi/Tm6B" | This study | N/A | See section on 'Fly stocks and genetics' in 'Materials and Methods' |
| Genetic reagent (*D. melanogaster*) | "y,w,hsFlp; Sp/CyO; UAS-eyRNAi/Tm6B" | This study | N/A | See section on 'Fly stocks and genetics' in 'Materials and Methods' |

*Appendix 1 Continued on next page*

*Appendix 1 Continued*

| Reagent type (species) or resource | Designation | Source or reference | Identifiers | Additional information |
|---|---|---|---|---|
| Genetic reagent (*D. melanogaster*) | "y,w,hsFlp; ayGal4 UASlacZ/ CyO; UAS-eyRNAi/Tm6B" | This study | N/A | See section on 'Fly stocks and genetics' in 'Materials and Methods' |
| Genetic reagent (*D. melanogaster*) | "y,w,hsFlp; ayGal4 UASlacZ/ CyO; hsNICD/Tm6B" | This study | N/A | See section on 'Fly stocks and genetics' in 'Materials and Methods' |
| Genetic reagent (*D. melanogaster*) | "y,w,hsFlp; Ubi RFPnls FRT40A; u8772-220-GFP/Sm6-Tm6B " | This study | N/A | See section on 'Fly stocks and genetics' in 'Materials and Methods' |
| Genetic reagent (*D. melanogaster*) | "y,w,hsFlp; Ubi RFPnls FRT40A; d5778-850-GFP/Sm6-Tm6B" | This study | N/A | See section on 'Fly stocks and genetics' in 'Materials and Methods' |
| Genetic reagent (*D. melanogaster*) | "y,w,hsFlp; UAS Dc2; dpn-Gal4 /Sm6-Tm6B" | This study | N/A | See section on 'Fly stocks and genetics' in 'Materials and Methods' |
| Genetic reagent (*D. melanogaster*) | "y,w,hsFlp; UAS Dc2; repo-Gal4 /Sm6-Tm6B" | This study | N/A | See section on 'Fly stocks and genetics' in 'Materials and Methods' |
| Genetic reagent (*D. melanogaster*) | "y,w,hsFlp; UAS GFP RNAi; d5778-850-GFP /Sm6-Tm6B" | This study | N/A | See section on 'Fly stocks' and 'Making of transgenic fly stocks' in 'Materials and Methods' |
| Genetic reagent (*D. melanogaster*) | "y,w,hsFlp; UAS GFP RNAi; 850-SuH-1m3m6m/Sm6-Tm6B" | This study | N/A | See section on 'Fly stocks' and 'Making of transgenic fly stocks' in 'Materials and Methods' |
| Genetic reagent (*D. melanogaster*) | "y,w,hsFlp; UAS GFP RNAi; 850-SuH-1m2m3m6m/Sm6-Tm6B" | This study | N/A | See section on 'Fly stocks' and 'Making of transgenic fly stocks' in 'Materials and Methods' |
| Genetic reagent (*D. melanogaster*) | "y,w,hsFlp; UAS GFP RNAi; 850-SuH-1–6 m/Sm6-Tm6B" | This study | N/A | See section on 'Fly stocks' and 'Making of transgenic fly stocks' in 'Materials and Methods' |
| Genetic reagent (*D. melanogaster*) | "y,w,hsFlp; ayGal4 UAS PCNA RNAi; hsNICD/ Sm6-Tm6B" | This study | N/A | See section on 'Fly stocks and genetics' and 'Making of transgenic fly stocks' in 'Materials and methods' |
| Genetic reagent (*D. melanogaster*) | "VsxGal4; E(spl) mγGFP/ CyO; UAS Dc2 /Tm6B" | This study | N/A | See section on 'Fly stocks and genetics' in 'Materials and methods' |
| Genetic reagent (*D. melanogaster*) | y, w, nos-Cas9 | Bloomington *Drosophila* Stock Centre | BL54591 | See section on 'Fly stocks' and 'Making of transgenic fly stocks' in 'Materials and methods' |
| Genetic reagent (*D. melanogaster*) | "y, w; PBAC{y[+]-attP-9A} VK00027" | Bloomington *Drosophila* Stock Centre | BL9744 | See section on 'Fly stocks' and 'Making of transgenic fly stocks' in 'Materials and methods' |
| Sequence-based reagent | Stellaris-Quasar 570-conjugated *slp2* mRNA probes | This study | Symbol: CG2939-RA; Flybase ID: FBtr0077500 | See section 'Fluorescence in situ hybridization' in 'Materials and methods' |
| Sequence-based reagent | Stellaris-Quasar 670-conjugated *slp1* mRNA probes | This study | Symbol: CG16738-RA; Flybase ID: FBtr0077499 | See section 'Fluorescence in situ hybridization' in 'Materials and methods' |
| Peptide, recombinant protein | AscI | New England Biolabs | Catalog_#: R0558S | See section 'Plasmid constructs ' in 'Materials and methods' |
| Peptide, recombinant protein | NotI-HF | New England Biolabs | Catalog_#: R3189S | See section 'Plasmid constructs 'in 'Materials and methods' |
| Peptide, recombinant protein | XhoI | New England Biolabs | Catalog_#: R0146S | See section 'Plasmid constructs 'in 'Materials and methods' |
| Peptide, recombinant protein | AgeI-HF | New England Biolabs | Catalog_#: R3552S | See section 'Plasmid constructs ' in 'Materials and methods' |

*Appendix 1 Continued on next page*

*Appendix 1 Continued*

| Reagent type (species) or resource | Designation | Source or reference | Identifiers | Additional information |
|---|---|---|---|---|
| Peptide, recombinant protein | PmeI | New England Biolabs | Catalog_#: R0560S | See section 'Plasmid constructs ' in 'Materials and methods' |
| Peptide, recombinant protein | ZraI | New England Biolabs | Catalog_#: R0659S | See section 'Plasmid constructs ' in 'Materials and methods' |
| Commercial assay or kit | NEB Builder HiFi DNA Assembly kit | New England Biolabs | E2621L | See section 'Plasmid constructs 'in 'Materials and methods' |
| Commercial assay or kit | Expand High Fidelity PCR System, dNTPack | Roche | 04738268001 | See section 'Plasmid constructs 'in 'Materials and methods' |
| Commercial assay or kit | Platinum SuperFi II PCR mastermix | Invitrogen | 12368010 | See section 'Plasmid constructs 'in 'Materials and methods' |
| Commercial assay or kit | REDExtract-N-Amp PCR ReadyMix | Sigma-Aldrich | R4775-1.2ML | See section 'Plasmid constructs ' in 'Materials and methods' |
| Commercial assay or kit | DNeasy Blood and Tissue Kit | Qiagen | 69,504 | See section 'Plasmid constructs ' in 'Materials and methods' |
| Peptide, recombinant protein | DpnI | New England Biolabs | Catalog_#: R0176S | See section 'DamID-seq' in 'Materials and methods' |
| Peptide, recombinant protein | DpnII | New England Biolabs | Catalog_#: R0543S | See section 'DamID-seq' in 'Materials and methods' |
| Peptide, recombinant protein | AlwI | New England Biolabs | Catalog_#: R0513S | See section 'DamID-seq' in 'Materials and methods' |
| Peptide, recombinant protein | Quick Ligation kit | New England Biolabs | Catalog_#: M2200S | See section 'DamID-seq' in 'Materials and methods' |
| Peptide, recombinant protein | T4 DNA ligase | New England Biolabs | Catalog_#: M0202S | See section 'DamID-seq' in 'Materials and methods' |
| Peptide, recombinant protein | T4 DNA polymerase | New England Biolabs | Catalog_#: M0203S | See section 'DamID-seq' in 'Materials and methods' |
| Peptide, recombinant protein | Klenow fragment | New England Biolabs | Catalog_#: M0210S | See section 'DamID-seq' in 'Materials and methods' |
| Peptide, recombinant protein | Klenow 3'->5' exo-enzyme | New England Biolabs | Catalog_#: M0201S | See section 'DamID-seq' in 'Materials and methods' |
| Peptide, recombinant protein | NEBNext High-Fidelity 2 X PCR master mix | New England Biolabs | Catalog_#: M0541S | See section 'DamID-seq' in 'Materials and methods' |
| Peptide, recombinant protein | Advantage 2 cDNA polymerase | Clontech | Catalog_#: A63880 | See section 'DamID-seq' in 'Materials and methods' |
| Peptide, recombinant protein | RNase A | Roche | Catalog_#: 11119915001 | See section 'DamID-seq' in 'Materials and methods' |
| Commercial assay or kit | Qubit dsDNA HS assay kit | Invitrogen | Catalog_#: Q32851 | See section 'DamID-seq' in 'Materials and methods' |
| Commercial assay or kit | Agencourt AMPure XP beads | Beckman-Coulter | Catalog_#: A63880 | See section 'DamID-seq' in 'Materials and methods' |
| Sequence-based reagent | AdRt oligo | *Marshall and Brand, 2015* PMID:27490632 DOI: 10.1038/nprot.2016.084 | N/A | 5' CTAATACGACTCACTATAGGGCAGCGTGGTCGCGGCCGAGGA 3' |

*Appendix 1 Continued on next page*

*Appendix 1 Continued*

| Reagent type (species) or resource | Designation | Source or reference | Identifiers | Additional information |
|---|---|---|---|---|
| Sequence-based reagent | AdRb oligo | **Marshall and Brand, 2015** PMID:27490632 DOI: 10.1038/nprot.2016.084 | N/A | 5' TCCTCGGCCG 3' |
| Sequence-based reagent | DamID_PCR oligo | **Marshall and Brand, 2015** PMID:27490632 DOI: 10.1038/nprot.2016.084 | N/A | 5' GGTCGCGG CCGAGGATC 3' |
| Sequence-based reagent | NGS_PCR1 | **Marshall and Brand, 2015** PMID:27490632 DOI: 10.1038/nprot.2016.084 | N/A | 5' AATGATACGGC GACCACCGA*G 3' *=phosphorothioate linkage |
| Sequence-based reagent | NGS_PCR2 | **Marshall and Brand, 2015** PMID:27490632 DOI: 10.1038/nprot.2016.084 | N/A | 5' CAAGCAGAAGA CGGCATACGA*G 3' *=phosphorothioate linkage |
| Sequence-based reagent | NGS_adaptors | **Marshall and Brand, 2015** PMID:27490632 DOI: 10.1038/nprot.2016.084 | N/A | |
| Software, algorithm | FIMO based web-application | Bart Deplancke | Site: https://biss.epfl.ch | |
| Software, algorithm | FIMO | **Grant et al., 2011** PMID:21330290 DOI: 10.1093/bioinformatics/btr064 | Site: https://meme-suite.org/meme Versions 4.11.1–5.4.1 | Accessed between 2016 to 2022; See section 'Bioinformatic analysis' in 'Materials and methods'. |
| Software, algorithm | The MEME suite | **Bailey et al., 2015** PMID:25953851 DOI: 10.1093/nar/gkv416 | Site: https://meme-suite.org/meme Versions 4.11.1–5.4.1 | Accessed between 2016 to 2022; See section 'Bioinformatic analysis' in 'Materials and methods'. |
| Software, algorithm | TOMTOM | **Gupta et al., 2007** PMID:17324271 DOI: 10.1186/gb-2007-8-2-r24 | Site: https://meme-suite.org/meme Versions 4.11.1–5.4.1 | Accessed between 2016 to 2022; See section 'Bioinformatic analysis' in 'Materials and methods'. |
| Software, algorithm | JASPAR | **Sandelin et al., 2004** PMID:14681366 doi: 10.1093/nar/gkh012 | Site: https://jaspar.genereg.net/ releases 6, 7, 8,9. | Accessed between 2016 to 2022; See section 'Bioinformatic analysis' in 'Materials and methods'. |
| Software, algorithm | Fly Factor Survey | **Zhu et al., 2011** PMID:21097781 DOI: 10.1093/nar/gkq858 | Site: https://mccb.umassmed.edu/ffs | Accessed between 2016 to 2022; See section 'Bioinformatic analysis' in 'Materials and Methods'. |
| Software, algorithm | Sequence Manipulation Suite Version 2 | **Stothard, 2000** PMID:10868275 DOI: 10.2144/00286ir01 | Site: https://www.bioinformatics.org/sms2 | Accessed between 2016 to 2022; See section 'Bioinformatic analysis' in 'Materials and Methods'. |
| Software, algorithm | Primer3web version 4.1.0 | Site: https://bioinfo.ut.ee/primer3 | Primer3web version 4.1.0 | |
| Software, algorithm | damidseq_pipeline | **Marshall and Brand, 2015** PMID:26112292 DOI: 10.1093/bioinformatics/btv386 | Site: http://owenjm.github.io/damidseq_pipeline Version: v1.4.6 | See section 'DamID-seq data analysis' in 'Materials and Methods'. |
| Software, algorithm | Irreproducibility Discover Rate (IDR) | Nathan Boley | Site: https://github.com/nboley/idr Version: 2.0.3 | See section 'DamID-seq data analysis' in 'Materials and Methods'. |
| Software, algorithm | Integrated Genomics Viewer (IGV) | Robinson, J.T., et al. (2012) PMID:21221095 doi: 10.1038/nbt.1754 | IGV_Linux_2.11.0 | See section 'DamID-seq data analysis' in 'Materials and Methods'. |
| Software, algorithm | DAVID | Site: https://david.ncifcrf.gov | Releases 6.8 and Dec 2021 | See section 'DamID-seq data analysis' in 'Materials and Methods'. |
| Software, algorithm | FIJI | **Schindelin et al., 2012** PMID:22743772 doi: 10.1038/nmeth.2019 | FIJI version 2.3.051 | See section 'Image analysis' in 'Materials and Methods'. |
| Software, algorithm | GraphPad Prism | GraphPad | GraphPad Prism version 9.2.0 | See section 'DamID-seq data analysis' in 'Materials and Methods'. |
| Software, algorithm | Adobe Photoshop | Adobe | Adobe Photoshop 2022 version: 23.4.1 | |
| Software, algorithm | Adobe Illustrator | Adobe | Adobe Illustrator 2022 Version 26.3.1 | |

