## [Editor Report]

Neural stem cells express cascades of transcription factors that are important for generating the diversity of neurons in the brain of flies and mammals. Yet nothing is known about whether the transcription factor cascades are built from direct gene regulation, e.g. factor A binding to enhancers in gene B to activate its expression. Here, Ray and Li show that one temporal factor, Slp1/2, is regulated transcriptionally via two molecularly defined enhancers that directly bind two other transcription factors in the cascade as well as integrating Notch signaling. This is a major step forward for the field and provides a model for subsequent studies on other temporal transcription factor cascades.

---

## [Decision Letter]

**Decision letter after peer review:**

Thank you for submitting your article "A Notch-dependent transcriptional mechanism controls expression of temporal patterning factors in *Drosophila* medulla" for consideration by *eLife*. Your article has been reviewed by 3 peer reviewers, including Chris Q Doe as the Reviewing Editor and Reviewer #1, and the evaluation has been overseen by a Reviewing Editor and Utpal Banerjee as the Senior Editor. The following individual involved in review of your submission has agreed to reveal their identity: Cédric Maurange (Reviewer #2).

The reviewers have discussed their reviews with one another, and the Reviewing Editor has drafted this to help you prepare a revised submission. Overall, the manuscript is a beautiful body of work using multiple methods to address the question of how one TTF (Slp1/2) is temporally regulated. There are two major comments that require experiments (hopefully they have been anticipated and are near completion) and numerous minor comments for clarity.

We encourage you to address the two essential points and all of the minor points prior to resubmission. We look forward to the speedy publication of your work.

Essential revisions:

(1) Deletion of both enhancers is necessary to show that they are required for activation of Slp1/2.

(2) Currently the role of N signaling in direct regulation of Slp1/2 is unclear. Does Notch directly and specifically regulates Slp1/2 expression instead of indirectly through earlier factors? I would like to see proof that Su(H) binding sites in the Slp1/2 locus are indeed required for Slp1/2 expression.

*Reviewer #1 (Recommendations for the authors):*

This work is from a new lab working on the mechanism of temporal transcription factor (TTF) gene regulation in optic lobe neuroblasts in *Drosophila*. About a dozen candidate TTFs are known to be sequentially expressed in OL neuroblasts, in part due to beautiful work by the π during her postdoc in the Desplan lab. Although some cross-regulatory interactions in this cascade have been defined, nothing is known about which are direct and nothing is known about the role of Notch or cell cycle progression in the TTF cascade.

This manuscript provides evidence for multiple direct inputs promoting the timely expression of the TTF Slp1/2: Eyeless, Scro, and Su(H) the Notch pathway effector. All acting through two partially redundant enhancers of ~220 and ~800 bp. Overall the paper provides a wonderful diversity of approaches that all build support for their conclusion. The text is clear and logical, and the figures mostly support the conclusions drawn. The paper obviously represents a great deal of work, during covid times, by the first author, which deserves appreciation.

I have tried to frame my comments so that they can be used to improve the manuscript without additional experiments, or at least with only simple experiments. All requests for new experiments should be considered as requests but not requirements. I note also the scholarly and comprehensive introduction -- nicely done.

Figure 1D should have a scale bar to indicate the length of genomic DNA shown. This would also serve to show that everything is at the same scale.

The single enhancer deletions show little or no phenotype, which makes it impossible to conclude that they are necessary for timely activation of Slp1/2; if possible, deleting both enhancers would be an excellent addition to the story: no phenotype would reveal the existence of another enhancer with this function; loss of Slp1/2 expression would show that these are the only required enhancers.

The binding site mutation data needs to be explained better. Are the constructs for wild type and the binding site mutants precisely the same except for the deleted DNA? Same length? Same genomic insertion site? And also, the precise nature of the DNA changes should be shown in the figure.

The location of the relevant TF binding sites should be shown in Figure 3, along with other predicted sites for other TFs – the latter might identify new regulators of the temporal cascade.

How are the clonal data in Figures 2 and 3 quantified? How many n's were assayed?

Figure 6A needs to be enlarged, I could barely read it. Although I may be the only person in the world left who reads the paper as a hard copy! It also needs to show the enhancers more clearly somehow… it took me three tries at the figure before finding them.

Figure 6A does not appear to show enrichment of the DamID peaks over the enhancers; the peaks seemed to be much wider. What am I missing here?

Figure 7 was the most difficult figure to process. The indicated phenotypes where hard to visualize. A reporter gene needs to be added to show the Vsx domain; the brackets were hard to link to the phenotype (e.g. there seemed to be phenotypes outside the bracketed region).

*Reviewer #2 (Recommendations for the authors):*

– The whole paragraph describing how the two enhancers u8772 and d5778 are identified is really not clear. In particular, it is not mentioned in the text how many and which candidate regulatory segments were cloned in the GFP-reporter vector and tested, and on what basis they were identified. The authors should better introduce the previous work that was made in particular by Fijioka and Jaynes on the dissection of the regulatory regions in the slp locus, and how much they rely on their work to identify some regulatory regions active in medulla NBs. They should also better explain the basis underlying REDfly-mediated identification of enhancers. The authors should also make a scheme depicting which regions they have screened for activity in medulla NBs and through what type of constructs and transgenic assays. Finally, do the authors think they have exhaustively covered the locus?

– The description of the bioinformatic analysis is unclear. The authors mention that they use different strategies to search for binding motifs. Their first strategy using TOMTOM and BISS appears to fail identifying the important sites after validation using mutated transgenic constructs. However the data are not shown. A second strategy using JASPAR, Fly Factor Survey and the Sequence Manipulation Suite appeared more fruitful. It would be useful to know what is the precise outcome from the two approaches. Did the first strategy only identified a subset of the predicted binding sites ? If yes which one were missing and identified via the second strategy?

Please also make, in a main figure, a scheme depicting the putative binding sites of the different transcription factors within the two enhancers and with the identified binding motifs.

– I am not familiar with the resolution that this technology should provide but the DamID profiles of SuH and Ey seem very similar in Figure 6, perhaps too similar. Can you show profiles on genes known to be targeted by either of the transcription factors as controls?

– Not clear whether the delay in Slp activation upon inactivation of Notch signaling is due to a global delay of all temporal transitions, or a specific delay in the Ey->slp transition. What about the Hth->Ey transition NBs upon inactivation of Notch Signaling? Are the late D+ and Tll+ temporal fates induced?

– Do you think Notch signaling is required, or facilitates all temporal transitions, or is only involved in the Ey->Slp transition? Please discuss this.

*Reviewer #3 (Recommendations for the authors):*

1. The authors should show that Slp1/2 mRNA transcription colocalize well with Slp1/2 expression in combined high magnification image to strengthen the conclusion the Slp1/2 expression is controlled at the transcription level.

2. The data shown in Figure S1F does not demonstrate that the two identified enhancer elements actually regulate Slp2 expression. While the authors show data to suggest that these enhancers can drive an expression pattern that is overlapping to Slp2, they do not show that the enhancers have any significant effect on the expression pattern of Slp2. The authors should consider performing deletion of both enhancers to see if this can significantly reduce Slp2 expression or should limit their conclusions to Slp1 with regards to these enhancer elements.

3. The data shown in Figure 1E and 1F are not sufficient to confidently describe the expression pattern of these enhancer elements. If the authors claim that d5778 is expressed in neuroblasts is true, they should be able to reduce neuroblast-specific GFP expression by driving gfpRNAi using a neuroblast-specific gal4 line but not by driving gfpRNAi using a glia-specific gal4 line.

4. In Figure 2, the data suggested that Su(H)RNAi delayed expression of Slp2 expression, and in figure 3, Su(H)RNAi causes a reduction in enhancer activity. The authors should stain for Slp1/Slp2 and the GFP enhancer in the same brains where Su(H)RNAi or eyRNAi are driven. Currently, the data presented does not suggest that the enhancer expression is directly linked to the expression of Slp1/2.

5. The authors claim that they identify Su(H) peaks around scro or hbn, however, the data shown in Figure S2B does not indicate any significant peaks for Su(H), only for eyeless. The authors either need to repeat the experiment to generate a strong enough consensus for Su(H) peaks in order to demonstrate significance or remove this claim.

6. The authors' DamID data suggests that eyeless is bound by Su(H). Could the effects seen by Su(H)RNAi be due to a reduction in eyeless signaling? The authors should consider the possibility that Su(H) acts on Slp1/2 through eyeless.

7. In regards, to Figure 4, the authors claim that mutating Slp binding sites did not significantly reduce reporter expression, but in the figure the graph shown suggests that the Slp binding site mutant had a significantly reduced expression, while the image shown suggests that patterning of the Slp binding site mutants is still intact. The authors should reconcile these inconsistencies.

8. The authors claim in Figure 5D and 5F, that the upon mutating Su(H) sites in enhancer, that GFP signal is still intact in the glia and likely intensified. If this is the case, the authors should be able to demonstrate this glial expression changes in the supplement. Additionally, the images in Figure 5D show what appears to be a complete decoupling in enhancer patterning. It is not clearly shown if this is due to the GFP signal in the glia, or if this loss of patterning is from another cause.

9. For Figure 7E and 7F the authors conclude the overexpression of NICD in the PCNA RNAi background rescues Slp2 expression loss, but the authors do not include a control showing what occurs if NICD is overexpressed in a WT background. The authors also did not offer any potential explanation for what role the cell cycle is playing

10. Please examine if the temporal transcription factors upstream of Slp1/2 can be activated by Notch.

11. In Figure 7E and 7F the authors use AyGal4 to minimally affect Ey expression, but the clones shown make it difficult to see how much Ey expression is affected. To more clearly illustrate Ey role, the authors should show if NICD can rescue Slp1/2 expression pattern in an Ey Loss of function background. This would help to clarify if Notch signaling is playing only a direct role or if Ey can mediate Notch activation of Slp1/2.

12. The authors do not include a model summarizing their findings, especially in regards to what role Notch and cell cycle progression play in temporal patterning.

---

## [Author Response]

Essential revisions:(1) Deletion of both enhancers is necessary to show that they are required for activation of Slp1/2.

We have successfully generated a line in which both enhancers are deleted (*slp^DED^*). We placed this double enhancer deletion line (*slp^DED^*) over a deficiency line (*slp^S37A^* that lacks the *slp1* transcription unit and all intervening sequences up to 16bp downstream of the *slp2* transcriptional start site) and examined Slp1/2 expression in larval brains. To our delight, in such brains, both Slp1 and Slp2 expression are lost specifically in medulla neuroblasts and their progeny, but their expression in lamina neurons and glia are not affected. As controls, heterozygous *slp^S37A^* or *slp^DED^* brains show wild-type Slp1 and Slp 2 expression patterns. This result (included in the new Figure 2) confirms that these two enhancers acting together are necessary and specific enhancers for Slp1 and Slp2 expression in medulla neuroblasts and progeny.

(2) Currently the role of N signaling in direct regulation of Slp1/2 is unclear. Does Notch directly and specifically regulates Slp1/2 expression instead of indirectly through earlier factors? I would like to see proof that Su(H) binding sites in the Slp1/2 locus are indeed required for Slp1/2 expression.

Thanks for raising this great question. We examined whether loss of Notch affects earlier factors, and indeed the upstream TTF Ey is also delayed when we knock down Notch or Su(H). Medulla neuroblasts from the youngest to the oldest are aligned on the lateral to medial spatial axis. Ey is normally turned on in the 2^nd^-3^rd^ neuroblast from the lateral edge of the medulla. In N-RNAi clones, Ey is turned on in the 5^th^-8^th^ neuroblast (6.50±1.05, n=6) (Figure 5A1-8), suggesting that Notch signaling is required for the timely turning-on of Ey. However, this does not preclude an additional direct role for Notch signaling in the Ey to Slp transition. To examine this, we counted the number of Ey+ neuroblast in which the transition to Slp stage occurs. We observed that in wild type neuroblasts, the transition to Slp stage occurs in the 3^rd^ -4^th^ Ey+ neuroblast, while in N-RNAi clones, the transition to Slp occurs in the 6^th^-9^th^ Ey+ neuroblast (7.67±1.21, n=6) (Figure 5A1-8). In *Su(H)^47^* mutant clones, the transition occurs in the 6^th^-8^th^ Ey+ neuroblasts (6.80±0.84) (Figure 5B-C’’). To make sure the delayed Slp transition is not mediated through Ey in *Su(H)^47^* mutant clones, we supplied Ey using a *dpn*-Gal4 and UAS-Ey in *Su(H)^47^* mutant clones. In such clones, Ey is over-expressed, but the transition to Slp stage is still delayed to the 6^th^-7^th^ Ey+ neuroblast (6.40±0.54, n=6) (Figure 5D-E’’). The difference from wild type is significant by t-test (see Figure 5 legend). These results suggest that Notch signaling is also required to facilitate the Ey to Slp transition in addition to an earlier role on Ey. We also examined the expression of Opa, the TTF upstream of Ey. Opa has two stripes of expression in medulla NBs. When we knock-down Notch pathway components, the 1^st^ stripe of Opa is expanded, while the 2^nd^ stripe is delayed. Furthermore, the expression of later factors D and BarH1 are severely delayed or mostly lost when we knocked down Su(H) (Figure 5G-H’). Overall, these results suggest that Notch may facilitate each temporal transition. Specifically for the Ey to Slp transition, we demonstrated a direct role for Notch signaling to promote Slp expression through the u8772 220bp enhancer. Mutating two predicted Su(H) binding sites in the u8772 220bp enhancer dramatically reduced the reporter expression, while mutating all four predicted Su(H) binding sites abolished the reporter expression. It is a different case for the d5778 850bp enhancer, mutation of four or six out of six predicted Su(H) binding sites did not decrease the reporter expression, suggesting Notch signaling does not activate the d5778 850bp enhancer through these binding sites. Taken together, these data suggest that Notch signaling may activate Slp expression only through the u8772 220bp enhancer. This explains why the d5778 850bp enhancer drives reporter expression later than the 220bp enhancer and the endogenous Slp. Further, with loss of Notch signaling, endogenous Slp expression is only delayed but not completely lost, and this can also be explained by our finding that the d5778 850 bp enhancer directs a delayed expression of Slp and this expression is not dependent on Notch signaling.

Reviewer #1 (Recommendations for the authors):This work is from a new lab working on the mechanism of temporal transcription factor (TTF) gene regulation in optic lobe neuroblasts in *Drosophila*. About a dozen candidate TTFs are known to be sequentially expressed in OL neuroblasts, in part due to beautiful work by the π during her postdoc in the Desplan lab. Although some cross-regulatory interactions in this cascade have been defined, nothing is known about which are direct and nothing is known about the role of Notch or cell cycle progression in the TTF cascade.This manuscript provides evidence for multiple direct inputs promoting the timely expression of the TTF Slp1/2: Eyeless, Scro, and Su(H) the Notch pathway effector. All acting through two partially redundant enhancers of ~220 and ~800 bp. Overall the paper provides a wonderful diversity of approaches that all build support for their conclusion. The text is clear and logical, and the figures mostly support the conclusions drawn. The paper obviously represents a great deal of work, during covid times, by the first author, which deserves appreciation.I have tried to frame my comments so that they can be used to improve the manuscript without additional experiments, or at least with only simple experiments. All requests for new experiments should be considered as requests but not requirements. I note also the scholarly and comprehensive introduction -- nicely done.Figure 1D should have a scale bar to indicate the length of genomic DNA shown. This would also serve to show that everything is at the same scale.

A scale bar is added, and this panel is now Figure 2A. In this panel, we also added the fragments that were screened and cloned.

The single enhancer deletions show little or no phenotype, which makes it impossible to conclude that they are necessary for timely activation of Slp1/2; if possible, deleting both enhancers would be an excellent addition to the story: no phenotype would reveal the existence of another enhancer with this function; loss of Slp1/2 expression would show that these are the only required enhancers.

Our double enhancer deletion result demonstrated that these two enhancers together are necessary and specific enhancers for Slp1 and Slp2 in medulla neuroblasts and progeny. Please see our response to “Essential Revisions #1”.

The binding site mutation data needs to be explained better. Are the constructs for wild type and the binding site mutants precisely the same except for the deleted DNA? Same length? Same genomic insertion site? And also, the precise nature of the DNA changes should be shown in the figure.

Thanks for pointing these out. We have now included all of the details for the binding site mutation data. In Table 2, we included the binding sites and the exact deletion or mutation to the binding sites in different constructs. Briefly, yes, the constructs for wild type and the binding site mutants are precisely the same except for the deleted or mutated DNA indicated in the table, and all constructs are inserted in the same genomic landing site.

The location of the relevant TF binding sites should be shown in Figure 3, along with other predicted sites for other TFs – the latter might identify new regulators of the temporal cascade.

We have now included a figure (Figure 3) showing the location of the predicted binding sites for Ey, Su(H), Scro/Vnd, and Slp in the two enhancers. Supplementary table 2 lists all of the predicted binding sites for transcription factors from FIMO analysis. However, we have checked our scRNA-seq database, most of other predicted transcription factors are either not expressed or expressed broadly, thus we did not focus on them for further analysis.

How are the clonal data in Figures 2 and 3 quantified? How many n's were assayed?

Thanks for raising this question. Figure 2 and 3 are currently Figure 4 and 6. The number of clones examined for each experiment are now listed in the figure legends. In general, more than 5 clones are quantified. For example, we quantified 12 N-RNAi clones, by counting the number of neuroblast from the lateral edge in which Slp is first turned on. While in wild-type brains Slp2 expression is seen in the 4th-5th neuroblast from the lateral edge of the medulla, inside N-RNAi clones Slp2 expression is first noted in the 10th -15th neuroblast (n=12 clones).

Figure 6A needs to be enlarged, I could barely read it. Although I may be the only person in the world left who reads the paper as a hard copy! It also needs to show the enhancers more clearly somehow… it took me three tries at the figure before finding them.

Sorry about that. We have reformatted the Dam-ID figures to improve visibility.

Figure 6A does not appear to show enrichment of the DamID peaks over the enhancers; the peaks seemed to be much wider. What am I missing here?

For the 220bp reporter, we do see IDR reproducible peaks for both Ey and Su(H) right over the enhancer. For the d5778 850bp enhancer, we also see prominent Ey binding peaks but less Su(H) binding peaks, consistent with our new observation that Notch signaling activates Slp expression through the 220bp enhancer but not the 850bp enhancer. Binding peaks for Ey and Su(H) are also observed surrounding the enhancer and promoter of *slp* genes, possibly because of the relatively low resolution of Dam-ID compared to ChIP-seq, and/or possible enhancer/promoter looping that brings the fusion protein close to the promoter region. It is also possible that Ey and Su(H) may bind to other enhancers in the region that are responsible for driving expression of *slp* genes in other contexts.

Figure 7 was the most difficult figure to process. The indicated phenotypes where hard to visualize. A reporter gene needs to be added to show the Vsx domain; the brackets were hard to link to the phenotype (e.g. there seemed to be phenotypes outside the bracketed region).

We have now included PCNA staining to show where the affected region is. In the affected region, the nuclear staining of PCNA is lost.

Reviewer #2 (Recommendations for the authors):– The whole paragraph describing how the two enhancers u8772 and d5778 are identified is really not clear. In particular, it is not mentioned in the text how many and which candidate regulatory segments were cloned in the GFP-reporter vector and tested, and on what basis they were identified. The authors should better introduce the previous work that was made in particular by Fijioka and Jaynes on the dissection of the regulatory regions in the slp locus, and how much they rely on their work to identify some regulatory regions active in medulla NBs. They should also better explain the basis underlying REDfly-mediated identification of enhancers. The authors should also make a scheme depicting which regions they have screened for activity in medulla NBs and through what type of constructs and transgenic assays. Finally, do the authors think they have exhaustively covered the locus?

Thanks for this great suggestion. We have now included a Table (Table 1) showing all of the fragments that we have screened or cloned. The coordinates of the fragments on chromosome 2L are also included. We have added detailed description of how we screened the lines generated by Fijioka and Jaynes on the dissection of the regulatory regions in the slp locus, and included all of our data on these lines as a supplementary figure (Figure 2—figure supplement 3). We also included how we did two rounds of enhancer bashing to identify the minimal 220bp enhancer, and how we narrowed down the second enhancer to 850bp. We have generated a schematic depicting all of the fragments we screened and cloned (Figure 2A). We have covered most of the locus. Furthermore, we showed deleting both enhancers by CRISPR completely eliminated Slp1 and Slp2 expression in medulla neuroblasts, suggesting that there is no other enhancer to drive Slp1and 2 expression in medulla neuroblasts.

– The description of the bioinformatic analysis is unclear. The authors mention that they use different strategies to search for binding motifs. Their first strategy using TOMTOM and BISS appears to fail identifying the important sites after validation using mutated transgenic constructs. However the data are not shown. A second strategy using JASPAR, Fly Factor Survey and the Sequence Manipulation Suite appeared more fruitful. It would be useful to know what is the precise outcome from the two approaches. Did the first strategy only identified a subset of the predicted binding sites ? If yes which one were missing and identified via the second strategy?Please also make, in a main figure, a scheme depicting the putative binding sites of the different transcription factors within the two enhancers and with the identified binding motifs.

Yes, the first strategy only identified a subset of the predicted binding sites. We have included a figure (Figure 3) and a table (Table 2) showing the location of the predicted binding sites for different transcription factors within the two enhancers, and the identified binding motifs. In our Table 2, we listed all of the binding sites identified by the second approach, and the sites also identified by the first strategy are indicated with a star*. We also included data for our first round of mutation on the 220bp enhancer: deletion of the FIMO-predicted binding site of Su(H) did not affect the reporter expression (Figure 7—figure supplement 1).

– I am not familiar with the resolution that this technology should provide but the DamID profiles of SuH and Ey seem very similar in Figure 6, perhaps too similar. Can you show profiles on genes known to be targeted by either of the transcription factors as controls?

For DamID-sequencing, the locations of readings also depend on the open chromatin region and the presence of GATC sequences. Therefore in the genomic regions where both transcription factors bind, their reading profiles may appear very similar. Furthermore, we need to look at the IDR reproducible peaks for the real binding events.

Our analysis did identify a list of genes that were bound reproducibly both by Ey and Su(H) (Supplementary Table 4). We think it is reasonable to expect that the TTF Ey and the Notch signaling pathway, both being critical regulators in the neuroblasts, may cooperatively regulate a large number of genes in neuroblasts. Using DAVID, we conducted a functional annotation clustering analysis of this gene list. Genes related to the Notch pathway and participating in cell cycles were strikingly enriched within this list of co-bound genes (Supplementary Table 5), thus raising the possibility that in medulla neuroblasts, TTFs such as Ey can modulate the responsiveness of Notch target genes in the medulla to mitogenic Notch signaling as has been seen in Type II neuroblasts (Farnsworth et. al 2015).

Our analysis also identified a set of genes bound uniquely by Ey. For example, only Ey I.D.R peaks were observed in the genomic region of *hmx* gene, and the profiles are now shown in (Figure 10—figure supplement1B). Hmx is a transcription factor expressed in Notch-off neurons born from the Ey stage neuroblasts, and consistently our results demonstrates specific binding for Ey but not Su(H).

A third set of genes are bound uniquely by Su(H) reproducibly (Supplementary Table 4). For example, an I.D.R peak only for Su(H) was found in the dll genomic region (Figure 10—figure supplement1C). Since it has been shown that Notch-on neurons born from the Tll stage neuroblasts express Dll, it will be interesting to test whether Notch signaling activates Dll expression directly through binding to this peak region.

– Not clear whether the delay in Slp activation upon inactivation of Notch signaling is due to a global delay of all temporal transitions, or a specific delay in the Ey->slp transition. What about the Hth->Ey transition NBs upon inactivation of Notch Signaling? Are the late D+ and Tll+ temporal fates induced?

As detailed in our response to “Essential Revisions #2”, loss of Notch signaling does affect the onset timing of earlier TTFs like Opa and Ey. However, by counting the number of Ey+ neuroblast in which the transition to Slp occurs, we found that the Ey->slp transition is further delayed by loss of Notch signaling. The expression of later factors D and BarH1 (a TTF that we recently identified to be between D and Tll, and required for Tll expression) are severely delayed or mostly lost when we knocked down Su(H). Thus our results suggest that Notch signaling may facilitate each temporal transition. Specifically for the Ey to Slp transition, we demonstrated a direct role for Notch signaling to promote Slp expression through the u8772 220bp enhancer. Mutating two predicted Su(H) binding sites in the u8772 220bp enhancer significantly reduced the reporter expression, while mutating all four predicted Su(H) binding sites abolished the reporter expression.

– Do you think Notch signaling is required, or facilitates all temporal transitions, or is only involved in the Ey->Slp transition? Please discuss this.

As discussed above, our data suggest that Notch signaling may facilitate each temporal transition. However, whether Notch signaling acts directly to regulate transcription of other TTFs remain to be investigated. We have added this discussion to the text.

Reviewer #3 (Recommendations for the authors):1. The authors should show that Slp1/2 mRNA transcription colocalize well with Slp1/2 expression in combined high magnification image to strengthen the conclusion the Slp1/2 expression is controlled at the transcription level.

We have now provided data showing that Slp2 mRNA colocalizes well with Slp2 protein in medulla neuroblasts (Figure 1D). Slp1 antibody, unfortunately doesn’t work well after the *in-situ* hybridization procedures. However, since Slp1 and Slp2 proteins colocalize (Figure 1B), and Slp1 and Slp2 mRNA also colocalize well, we think it is safe to conclude that Slp1/2 expression is controlled at the transcription level.

2. The data shown in Figure S1F does not demonstrate that the two identified enhancer elements actually regulate Slp2 expression. While the authors show data to suggest that these enhancers can drive an expression pattern that is overlapping to Slp2, they do not show that the enhancers have any significant effect on the expression pattern of Slp2. The authors should consider performing deletion of both enhancers to see if this can significantly reduce Slp2 expression or should limit their conclusions to Slp1 with regards to these enhancer elements.

We have now included our double enhancer deletion result which demonstrated that these two enhancers together are necessary and specific enhancers for both Slp1 and Slp2 in medulla neuroblasts and progeny. Please see our response to “Essential Revisions #1”.

3. The data shown in Figure 1E and 1F are not sufficient to confidently describe the expression pattern of these enhancer elements. If the authors claim that d5778 is expressed in neuroblasts is true, they should be able to reduce neuroblast-specific GFP expression by driving gfpRNAi using a neuroblast-specific gal4 line but not by driving gfpRNAi using a glia-specific gal4 line.

Thanks for this great suggestion. We used a neuroblast-specific Gal4 (dpn-Gal4) to drive UAS-GFP-RNAi in neuroblasts, and this eliminated GFP expression in the medulla region. In contrast, GFP expression in neuroblasts is not affected when we use repo-Gal4 to drive UAS-GFP-RNAi. We provide these data in the Figure 2—figure supplement 4.

4. In Figure 2, the data suggested that Su(H)RNAi delayed expression of Slp2 expression, and in figure 3, Su(H)RNAi causes a reduction in enhancer activity. The authors should stain for Slp1/Slp2 and the GFP enhancer in the same brains where Su(H)RNAi or eyRNAi are driven. Currently, the data presented does not suggest that the enhancer expression is directly linked to the expression of Slp1/2.

We did have Slp2 staining in the same experiment, but didn’t show the Slp2 channel in the original figure. Now we have included the Slp2 channel (current Figure 6). We can see that both the reporter expression and the Slp2 expression are delayed, and they closely correlate with each other. These data show that the enhancer expression is directly linked to the expression of Slp1/2.

5. The authors claim that they identify Su(H) peaks around scro or hbn, however, the data shown in Figure S2B does not indicate any significant peaks for Su(H), only for eyeless. The authors either need to repeat the experiment to generate a strong enough consensus for Su(H) peaks in order to demonstrate significance or remove this claim.

We have removed this claim.

6. The authors' DamID data suggests that eyeless is bound by Su(H). Could the effects seen by Su(H)RNAi be due to a reduction in eyeless signaling? The authors should consider the possibility that Su(H) acts on Slp1/2 through eyeless.

We examined whether loss of Notch or Su(H) affects Ey expression. Indeed Ey is also delayed when we knock down Notch or Su(H). Ey is normally turned on in the 2^nd^-3^rd^ neuroblast. In N-RNAi clones, Ey is turned on in the 6^th^-8^th^ neuroblast (6.50±1.05, n=6) (Figure 5A1-8), suggesting that Notch signaling is required for the timely turning-on of Ey. However, this does not preclude an additional direct role for Notch signaling in the Ey to Slp transition. To examine this, we counted the number of Ey+ neuroblast in which the transition to Slp occurs. We observed that in wild type neuroblasts, the transition to Slp occurs on the 3^rd^ -4^th^ (3.83±0.41, n=6) Ey+ neuroblast, while in N-RNAi clones, the transition to Slp occurs on the 6^th^-9^th^ Ey+ neuroblast (7.67±1.21, n=6) (Figure 5A1-8). In *Su(H)^47^* mutant clones, the transition occurs on the 6^th^-8^th^ Ey+ neuroblasts (6.80±0.84) (Figure 5B-C’’). To make sure the delayed Slp expression is not mediated through Ey in *Su(H)^47^* mutant clones, we supplied Ey using a *dpn*-Gal4 and UAS-Ey in *Su(H)^47^* mutant clones. In such clones, Ey is over-expressed, but the transition to Slp stage is still delayed to the 6^th^-7^th^ Ey+ neuroblast (6.40±0.54, n=5) (Figure 5D-E’’). The difference is significant by t-test ( see figure 5 legends). These results suggest that Notch signaling also facilitate the Ey to Slp transition in addition to an earlier role on Ey.

7. In regards, to Figure 4, the authors claim that mutating Slp binding sites did not significantly reduce reporter expression, but in the figure the graph shown suggests that the Slp binding site mutant had a significantly reduced expression, while the image shown suggests that patterning of the Slp binding site mutants is still intact. The authors should reconcile these inconsistencies.

Mutating Slp binding site reduced the expression level significantly, but didn’t affect the expression pattern. We have clarified this in the text.

8. The authors claim in Figure 5D and 5F, that the upon mutating Su(H) sites in enhancer, that GFP signal is still intact in the glia and likely intensified. If this is the case, the authors should be able to demonstrate this glial expression changes in the supplement. Additionally, the images in Figure 5D show what appears to be a complete decoupling in enhancer patterning. It is not clearly shown if this is due to the GFP signal in the glia, or if this loss of patterning is from another cause.

Mutating Su(H) sites in d5778 850bp caused greatly intensified glia expression preventing us to assess the neuroblast expression. We now provide a co-staining with glia marker Repo to show the intensified GFP expression in glia (Figure 9 —figure supplement1). When we used repo-Gal4 to drive GFP-RNAi in glia to remove the glia expression as suggested (Figure 9), we actually can see the neuroblast expression pattern again. This shows that the apparent change in enhancer patterning is due to very strong GFP signal in the glia.

9. For Figure 7E and 7F the authors conclude the overexpression of NICD in the PCNA RNAi background rescues Slp2 expression loss, but the authors do not include a control showing what occurs if NICD is overexpressed in a WT background. The authors also did not offer any potential explanation for what role the cell cycle is playing

The NICD construct we use to rescue PCNA-RNAi is *dpn>FRT-stop-FRT3-FRT-NICD* transgene (which drives the expression of NICD from the *dpn* enhancer in presence of heat shock activated Flippase). We have now included panels showing Ey and Slp expression with the same NICD construct under same heat shocking conditions in wild type background. The initiation of Slp is not significantly affected in the neuroblasts, possibly because normal cycling neuroblast already have active Notch signaling.

We have provided data and discussion showing that the cell cycle progression is required to maintain active Notch signaling in neuroblasts, which is required to promote temporal transitions.

10. Please examine if the temporal transcription factors upstream of Slp1/2 can be activated by Notch.

We demonstrate that loss of Notch or Su(H) does cause delay in the expression of Ey, which is the TTF upstream of Slp1/2. However, as detailed in our answer to question #6, Notch signaling also has an additional direct role in the Ey to Slp transition.

11. In Figure 7E and 7F the authors use AyGal4 to minimally affect Ey expression, but the clones shown make it difficult to see how much Ey expression is affected. To more clearly illustrate Ey role, the authors should show if NICD can rescue Slp1/2 expression pattern in an Ey Loss of function background. This would help to clarify if Notch signaling is playing only a direct role or if Ey can mediate Notch activation of Slp1/2.

In our current Figure 5, we showed that supplying NICD did not rescue Slp2 expression in Ey-RNAi clones. However, this only suggests that NICD requires Ey to activate Slp expression, and does not preclude a direct role of NICD. To test this, we over-expressed Ey in *Su(H)* mutant clones, and we showed that although Ey is over-expressed, the transition to the Slp stage is still delayed (please see details in our answer to question 6). These results suggest that both Ey and Notch signaling are directly required for the timing of Slp activation.

12. The authors do not include a model summarizing their findings, especially in regards to what role Notch and cell cycle progression play in temporal patterning.

We have now included a model figure (Figure 12).